# LNUCB-TA: LINEAR-NONLINEAR HYBRID BANDIT LEARNING WITH TEMPORAL ATTENTION

## ABSTRACT

Existing contextual multi-armed bandit (MAB) algorithms struggle to simultaneously capture long-term trends as well as local patterns across all arms, leading to suboptimal performance in complex environments with rapidly changing reward structures. Additionally, they typically employ static exploration rates, which do not adapt to dynamic conditions. To address these issues, we present LNUCB-TA, a hybrid bandit model that introduces a novel nonlinear component (adaptive $k$-Nearest Neighbors ($k$-NN)) designed to reduce time complexity, and an innovative global-and-local attention-based exploration mechanism. Our method incorporates a unique synthesis of linear and nonlinear estimation techniques, where the nonlinear component dynamically adjusts $k$ based on reward variance, thereby effectively capturing spatiotemporal patterns in the data. This is critical for reducing the likelihood of selecting suboptimal arms and accurately estimating rewards while reducing computational time. Also, our proposed attention-based mechanism prioritizes arms based on their historical performance and frequency of selection, thereby balancing exploration and exploitation in real-time without the need for fine-tuning exploration parameters. Incorporating both global attention (based on overall performance across all arms) and local attention (focusing on individual arm performance), the algorithm efficiently adapts to temporal and spatial complexities in the available context. Empirical evaluation demonstrates that LNUCB-TA significantly outperforms state-of-the-art contextual MAB algorithms, including purely linear, nonlinear, and vanilla combination of linear and nonlinear bandits based on cumulative and mean rewards, convergence performance, and demonstrates consistency of results across different exploration rates. Theoretical analysis further proves the robustness of LNUCB-TA with a sub-linear regret bound.

## 1 INTRODUCTION

The multi-armed bandit (MAB) problem brings to light a fundamental challenge in decision-making dynamics, emphasizing the need to strike balance between exploration and exploitation (Russac et al., 2019; Audibert et al., 2009; Hillel et al., 2013). In Reinforcement Learning (RL), this challenge manifests as a continuous decision-making process (Zhu et al., 2022). Specifically, the RL agents must navigate the trade-off between uncovering new opportunities to better utilize their environment versus leveraging proven strategies to realize immediate benefits (Reeve et al., 2018; Bouneffouf et al., 2020; Sani et al., 2012). Balancing this trade-off is critical for developing adaptive strategies to improve outcomes across various domains such as online advertising (Schwartz et al., 2017), recommendation systems (Li et al., 2010; Ding et al., 2021), and clinical trials (Villar et al., 2015; Aziz et al., 2021). This dilemma becomes pronounced in environments marked by uncertainty, *e.g.* digital marketing (Shi et al., 2023). Particularly, algorithms aim to maximize user engagement by deciding advertisements displays to different segments, *i.e.*, weighing the benefits of exploring diverse advertisements against exploiting those with proven success.

**Foundational approaches.** Given the extensive literature on MAB, our study specifically concentrates on Upper Confidence Bound (UCB) variants and linear estimation methods. Foundational methods such as the UCB algorithm optimize decision-making by constructing confidence bounds around estimated rewards and selecting the action with the highest upper bound (Auer et al., 2002a). This technique is further refined in the Kullback-Leibler Upper Confidence Bound (KL-UCB) algo-

rithm, which enhances the accuracy of these intervals using the Kullback-Leibler divergence (Garivier & Cappé, 2011). Despite their efficacy, both UCB and KL-UCB often overlook the crucial role of contextual information, where each action can be tailored to the specific observable environmental factors, or 'contexts' to maximize the obtained rewards (Bubeck et al., 2012).

Extending these concepts to address contextual dynamics, the Linear Upper Confidence Bound (LinUCB) algorithm assumes a linear relationship between contextual features and expected rewards (Chu et al., 2011; Dimakopoulou et al., 2019). LinUCB constructs confidence bounds around these estimated rewards and selects actions based on the upper bounds of these estimates (Li et al., 2010). Linear Thompson Sampling (LinThompson) also operates under the assumption that expected rewards are linearly related to contextual features, utilizing Thompson Sampling (TS) to balance exploration and exploitation(Agrawal & Goyal, 2013). Despite its strategic approach, LinThompson can fall short by often estimating influence probabilities directly, which can lead to locally optimal solutions due to insufficient exploration. To address this, the LinThompsonUCB algorithm combines linear estimation with TS's probabilistic approach and UCB confidence intervals to enhance exploration and performance. (Zhang, 2019). However, while effective, the reliance of LinUCB, LinThompson, and LinThompsonUCB on linear assumptions can limit their performance in more complex environments. To address this limitation, the $k$-Nearest Neighbour UCB ($k$-NN UCB) and $k$-Nearest Neighbour KL-UCB ($k$-NN KL-UCB) methods utilize the locality of feature space to enhance action selection (Reeve et al., 2018). These models leverage contextual information by considering environmental features, thereby improving accuracy.

**Existing gaps and intuition.** Despite advancements in MAB algorithms, existing algorithms predominantly fail to incorporate adaptive strategies for reward estimation as a function of the context. Linear models, constrained by static parameter updates, often fail in scenarios with inherently nonlinear relationships between contextual features and rewards, leading to outdated estimations and slower convergence (Russac et al., 2019; Dimakopoulou et al., 2019; Zhang, 2019). While nonlinear approaches like $k$-NN-based models (Reeve et al., 2018) offer flexibility, they often struggle with computational efficiency and adaptability in dynamic environments. Moreover, these models usually overlook crucial long-term trends, which can lead to overfitting in sparse scenarios, degraded generalization, and increased variance in reward estimations (Eleftheriadis et al., 2024). These limitations restrict existing algorithms' ability to capture both long-term trends and immediate local patterns effectively, leading to inconsistent performance across various scenarios.

In addition, conventional methods rely on static exploration rates, leading to inefficient convergence and suboptimal decision-making (Bubeck et al., 2012). Specifically, high exploration rates cause algorithms to frequently test suboptimal options, slowing progress and increasing regret (Audibert et al., 2009). Conversely, low exploration rates leads to premature conclusions on less optimal solutions, foregoing potentially better options (Odeyomi, 2020). To address this, studies have proposed fine-tuning, experimentation, and dynamic exploration rates (Carlsson et al., 2021; Russac et al., 2019; Alon et al., 2015). However, these approaches often fall short in fully capturing the intricate, evolving patterns of rewards in non-stationary environments, such as recommendation systems or clinical trials (Villar et al., 2015; Liu et al., 2024b; De Curtò et al., 2023). Existing solutions typically rely on pre-defined heuristics or manual tuning, which can be suboptimal when rewards shift unexpectedly, complicating the search for an optimal setting (Bouneffouf et al., 2020; Russac et al., 2019). A key challenge of the existing approaches is to effectively adapt exploration rates as reward distributions change over time. As a result, context-awareness becomes critical to successfully manage exploration (Liu et al., 2024b).

**Contribution.** In this work, we have developed LNUCB-TA, which introduces a novel nonlinear strategy through an adaptive $k$-NN that dynamically adapts based on reward characteristics and shifts, effectively solving the time complexity issues commonly associated with nonlinear models. It also presents an attention-based exploration factor to move beyond the constraints of existing exploration rates. This model culminates in a unique synthesis of linear and nonlinear hybrid contextual MAB algorithms, comprehensively addressing the need for adaptive strategies in reward estimation to simultaneously capture long-term trends as well as local patterns across all arms. As shown in Table 1, LNUCB-TA incorporates a linear component for a global approximation of the reward function and a unique nonlinear component for capturing local patterns. The proposed nonlinear component employs a data driven (variance-guided), non-parametric criterion for $k$ selection based on reward

Table 1: Key attributes in our approach compared to existing MAB algorithms. "Yes" indicates the presence of the feature, "No" indicates the absence of the feature, and "N/A" indicates not applicable.

| Algorithm | Linear Modeling | Local History Modeling | Attention Mechanism | $k$ Selection Method |
|---|---|---|---|---|
| UCB | No | No | No | N/A |
| KL-UCB | No | No | No | N/A |
| $k$-NN UCB | No | Yes | No | Function optimization |
| $k$-NN KL-UCB | No | Yes | No | Function optimization |
| LinThompson | Yes | No | No | N/A |
| LinThompsonUCB | Yes | No | No | N/A |
| LinUCB | Yes | No | No | N/A |
| **LNUCB-TA** | **Yes** | **Yes** | **Yes** | **Variance guided, nonparametric** |

histories to reduce time complexity. Complementing this, the attention-based mechanism, inspired by the global-and-local attention (GALA) concept (Linsley et al., 2018), dynamically adjusts the exploration strategy by utilizing past interactions and rewards. This temporal attention approach adaptively prioritizes arms based on their historical rewards and selection frequency, eliminating the need for fine-tuning and precisely balancing exploration and exploitation in real-time.

**Motivating examples.** One application of the proposed hybrid model is in online advertisement recommendation, aiming to maximize user engagement through demographics, browsing history, and time-specific data (Zeng et al., 2016). The linear component captures broad trends, such as higher click-through rates for fashion advertisements among users aged 18 to 25, while the adaptive $k$-NN component refines this by recognizing local patterns. For instance, users within the 18 to 25 age group who frequently visit sports websites might prefer sports equipment advertisements. Furthermore, the novel exploration mechanism dynamically balances exploring new advertisement types and exploiting known preferences, thus optimizing real-time recommendations by leveraging both global trends and individual user behaviors.

Another application is in the exploration of partially observed social networks to maximize node discovery within a set query budget (Madhawa & Murata, 2019b), where our proposed hybrid model proves beneficial. The linear model identifies nodes with high-degree centrality as valuable targets based on their potential to connect to many others. The adaptive $k$-NN model enhances this strategy by pinpointing densely connected sub-communities within these high-centrality nodes, likely revealing new nodes when queried. Meanwhile, the attention mechanism dynamically shifts the exploration and exploitation based on the real-time performance of each node, enhancing the efficiency of network exploration by focusing on nodes that show promising connectivity trends while still exploring lesser-known parts of the network.

**Organization.** The rest of the paper is structured as follows. Section 2 covers the rigorous mathematical setup of the problem. Section 3 presents the LNUCB-TA algorithm. The theoretical analysis of the algorithm is presented in Section 4. Section 5 provides the experimental results. Conclusions are discussed in Section 6. Detailed proofs of the theoretical results, additional findings, limitations, future research directions, and implementation guidelines are included in the Appendix.

## 2 HYBRID CONTEXTUAL MAB LEARNING

**Problem definition.** We consider a hybrid contextual MAB problem within a metric space $(\mathcal{X} \times \mathcal{Z}, \rho)$, where $\mathcal{X} \times \mathcal{Z}$ represents the joint space of context features and reward history. Time is indexed discretely as $t = 1, 2, \ldots, T$, where $T$ is the total number of time steps. Each context $x_t \in \mathcal{X}$ at time $t$ corresponds to a set of possible actions, or "arms," indexed by $a$ within the set $\mathcal{A} = \{1, \ldots, A\}$, where $A$ is the total number of arms. The reward corresponding to each arm $a$ for a given context $x_t$ at time $t$ is denoted as a random variable $Y_t^a$, constrained within the interval $[-1, 1]$. The vector $Y_t = (Y_t^a)_{a \in \mathcal{A}} \in \mathbb{R}^A$ comprises the stochastic rewards for all arms at time $t$, and the random variable

$Y_t^a$ is defined conditionally on the context and the history of previous rewards. Upon observation, the realized reward for arm $a$ is given by $\hat{Y}_t^a = o_t^a(x_t, z_t) + \xi_t^a$, where $\xi_t^a$ is the noise term, capturing stochastic errors not explained by the model predictions for arm $a$. Here, the expected reward for arm $a$ at time $t$ is given by the function $o_t^a : \mathcal{X} \times \mathcal{Z} \to [-1, 1]$, defined as:

$$o_t^a(x_t^a, z_t^a) = \mathbb{E}[Y_t^a \mid X_t = x_t, Z_t = z_t] = l_t^a(x_t^a) + f_{k,t}^a(x_t^a, z_t^a) = \mu_t^a \cdot x_t^a + \text{k-NN}_{k,t}^a(x_t^a, z_t^a), \quad (1)$$

where $\mathbb{E}$ denotes the expectation, and $z_t^a = \{\hat{Y}_s^a : s < t, a \in \mathcal{A}\}$ represents the observed historical rewards for arm $a$ up to time $t$ with $z_t \in \mathcal{Z}$, and the feature vector $X_t$ is drawn independently and identically distributed (i.i.d.) from a fixed marginal distribution $\mathbb{D}$ over $X$. The linear model's prediction for arm $a$ at the context $x_t^a$, which represents the specific feature vector for arm $a$ at time $t$, is given by $l_t^a(x_t^a) = \mu_t^a \cdot x_t^a$. The $k$-NN model's estimation using $k$ number of nearest neighbors for each arm is based on the corresponding historical observed rewards for the selected neighbors up to time $t$. The value of $k_t^a$ is determined dynamically based on the variance of the reward history for each arm at time step $t$, ensuring that the model adapts to changes in the distribution of rewards over time. The $k$-NN estimation is defined as $f_{k,t}^a(x_t^a, z_t^a) = \frac{1}{k_t^a} \sum_{s \in N_{k_t^a}(x_t^a)} \hat{Y}_s^a$, where $\hat{Y}_s^a$ represents the observed reward for arm $a$ at time step $s$ (with $s < t$). The set of neighbors $N_{k_t^a}(x_t^a)$ denotes the indices of the $k_t^a$-nearest neighbors to $x_t^a$, selected based on the Euclidean distance within the contextual feature space. Thus, $f_{k,t}^a$ uses only the observed rewards from $z_t^a$ for neighbors that are closest in terms of context similarity. Furthermore, for any context $x_t^a \in \mathcal{X}$ and a radius $r > 0$, $\text{BALL}_t^a(x_t^a, r)$ denotes the open metric ball centered at $x_t^a$ with radius $r$ for arm $a$. This metric ball is pivotal for analyzing distances and neighborhood relations within the joint space $\mathcal{X} \times \mathcal{Z}$.

**Decision policy.** The decision-making process within the hybrid contextual MAB framework is guided by a policy $\pi = \{\pi_t\}_{t \in [T]}$, where each policy function $\pi_t : \mathcal{X} \times \mathcal{Z} \to [A]$ maps the observed context and reward history to an arm. This mapping is based on integration of linear estimation and nonlinear estimation utilizing the historical data $\mathcal{H}_{t-1} = \{(X_s, \pi_s, Y_s^{\pi_s})\}_{s \in [t-1]}$, which consists of previously observed contexts, the arms chosen, and the corresponding rewards, respectively.

**Exploration-exploitation trade-off.** In our problem, the exploration-exploitation trade-off is managed through a dynamic, attention-based exploration factor. This approach adapts the exploration parameter $\alpha$ in real-time based on both global performance ($g$) and specific reward patterns of individual arms ($n_t^a$), ensuring a more balanced and effective strategy. The exploration parameter $\alpha$ is updated dynamically according to the formula with weight factor $\kappa$ as:

$$\alpha_{N_t^a} = \frac{\alpha_0}{N_t^a + 1} \cdot (\kappa g + (1 - \kappa) n_t^a), \quad (2)$$

where $n_t^a = \frac{1}{N_t^a} \sum_{\hat{Y}_s^a \in z_t^a} \hat{Y}_s^a = \frac{1}{N_t^a} \sum_{s=1}^{t-1} \hat{Y}_s^a$ represents the average reward history of arm $a$ up to time $t$ (reward patterns of an individual arm), with $N_t^a = |\hat{Y}_{1:t-1}^a|$ as the number of pulls of arm $a$ up to time $t$. If $N_t^a = 0$, $n_t^a$ is set to zero.

**Objective.** The primary aim is to maximize the cumulative reward over $T$ time steps, represented by $\sum_{t \in [T]} Y_t^{\pi_t}$, and to minimize the regret relative to an oracle policy $\pi^* = \{\pi_t^*\}_{t \in [T]}$, where $\pi_t^* = \arg\max_{a \in [A]} o_t^a(x_t, z_t)$. In LNUCB-TA, the optimal decision $(\pi_t^a)^*$ through the optimal context $(x_t^a)^*$ for each arm would be the decision that maximizes the expected combined reward based on the linear model predictions and the adjustments made by the $k$-NN model, using the best available historical data up to time step $t$ defined as:

$$(x_t^a)^* \in \arg\max_{x \in D} \left( (\mu^a)^* \cdot (x_t^a) + f_{k,t}^a(x_t^a, z_t^a) \right), \quad (3)$$

where $(\pi_t^a)^*$ refers to the best reward obtained for arm $a$ based on its history over $t$ steps, which leads to the theoretical optimal action $\pi_t^*$, and $D$ represents the decision space. Although we compute an optimal action for each arm, the model ultimately selects only one arm to play per time step, choosing the one with the highest expected reward, $(\mu^a)^*$ is the best estimate of the parameter vector across arm $a$, assuming an oracle setting, or the true underlying model known retrospectively.

**Regret analysis.** The regret, $R_T(\pi)$, is a measure of the performance difference and is defined as:

$$R_T(\pi) = \sum_{t \in [T]} (Y_t^{\pi_t^*} - Y_t^{\pi_t}). \tag{4}$$

In our proposed model, for a single arm $a$, the regret at time $t$ can be defined as:

$$\text{regret}_t^a = \Delta_t^a \left( g_t^a \left( (x_t^a)^*, (z_t^a)^* \right) - o_t^a \left( x_t^a, z_t^a \right) \right), \tag{5}$$

where $g_t^a \left( (x_t^a)^*, (z_t^a)^* \right)$ is the optimal expected reward for arm $a$ at the optimal context $(x_t^a)^*$, which is the feature vector that would yield the highest reward for arm $a$, leading to optimal $(z_t^a)^*$, and $\Delta_t^a$ is the indicator function that equals 1 if arm $a$ is selected at time $t$ and 0 otherwise. The function $o_t^a(x_t^a, z_t^a)$ represents the expected reward under the decision made by the policy $\pi_t^a$ at context $x_t^a$ with reward history of $z_t^a$. As a result, the total cumulative regret for LNUCB-TA over a time horizon $T$ across all arms is calculated as:

$$R_T = \sum_{a=1}^{A} \sum_{t=0}^{T} \Delta_t^a \left( g^a \left( (x_t^a)^*, (z_t^a)^* \right) - o_t^a \left( x_t^a, z_t^a \right) \right)$$

$$= \sum_{a=1}^{A} \sum_{t=0}^{T} \Delta_t^a \left( l_t^a \left( (x_t^a)^* \right) + f_{k,t}^a \left( (x_t^a)^*, (z_t^a)^* \right) - \left( l_t^a(x_t^a) + f_{k,t}^a \left( x_t^a, z_t^a \right) \right) \right) \tag{6}$$

$$= \sum_{a=1}^{A} \sum_{t=0}^{T} \Delta_t^a \left( (\mu_t^a)^* \cdot (x_t^a)^* + k\text{-NN}_{k,t}^a \left( (x_t^a)^*, (z_t^a)^* \right) - \left( \mu_t^a \cdot x_t^a + k\text{-NN}_{k,t}^a \left( x_t^a, z_t^a \right) \right) \right).$$

## 3 METHODOLOGY

### 3.1 OVERALL CONCEPT

**Intuition.** We propose the LNUCB-TA model, which introduces two significant innovations to previously proposed contextual UCB algorithms. Both of these advancements enhance the adaptability and accuracy in dynamic environments. The proposed method, shown in Algorithm 1, is initiated using the structural framework of the LinUCB algorithm, which employs a linear model to estimate the rewards for each arm $a$ based on contextual features indicated as $l_t^a = (x_t^a)^\top \mu_t^a$. This basic linear framework is then augmented using a nonlinear component through the use of the $k$-Nearest Neighbors method. This enhancement integrates insights from the history of both the reward and context, and effectively captures the recent profile of the features (Algorithm 2).

In addition to refining reward estimations, our approach introduces an attention-based exploration factor, $\alpha_{N_t^a}$, which tunes the exploration-exploitation balance dynamically (Algorithm 3). This provides the dynamic upper confidence bound as:

$$UCB_t^a = (\alpha_{N_t^a}) \cdot \sqrt{(x_t^a)^\top (\Sigma_t^a)^{-1} x_t^a} \tag{7}$$

---

**Algorithm 1** LNUCB-TA

---

1: **Input:** $\lambda, \beta, \alpha_0, \kappa$          ▷ Model parameters
2: **for** $t = 0, 1, 2, \ldots$ **do**
3:      **for** each arm $a$ in $A$ **do**
4:          Compute $l_t^a = (x_t^a)^\top \mu_t^a$          ▷ Linear estimation
5:          Compute $k$-NN score (reward adjustment)          ▷ Nonlinear estimation
6:          Compute $UCB_t^a$ based on attention-based exploration rate          ▷ Dynamic UCB
7:      **end for**
8:      Select arm $a_t = \arg\max_{a \in A} \left( l_t^a + k\text{-NN adjustment} + UCB_t^a \right)$
9:      Update $\text{BALL}_{t+1}^a$ and model parameters          ▷ Uncertainty region
10: **end for**

---

**Method.** LNUCB-TA model, shown in Algorithm 1, not only maintains the structure of the original LinUCB framework but also seamlessly integrates adaptive nonlinear adjustments and real-time refinements in confidence bounds and exploration rates. These enhance the model's adaptability and accuracy in complex environments. Through this careful augmentation, we extend the LinUCB's capability while preserving its theoretical underpinnings, ensuring that our contributions are both innovative and robustly grounded in established methodologies. In the following section, the two novel components are discussed in more detail.

## 3.2 NONLINEAR ESTIMATION USING FEATURE AND REWARD HISTORY

**Intuition.** The adaptive $k$-NN ensures that the model adjusts its reliance on the reward history of each arm based on the stability of the rewards. It seamlessly integrates more insights from $k$-NN as additional data becomes available and defaults to a more conservative approach when data is sparse. This unique method effectively captures local patterns with improved time efficiency, without the need for extensive function optimization, thereby enhancing adaptability and responsiveness in dynamic environments.

**Method.** The adaptive $k$-NN strategy employed in the model, detailed in Algorithm 2, takes both the reward history and the feature vector of each arm as inputs. This method is applied conditionally, specifically when the length of the feature vector $x_t^a$ (where $x_t^a$ represents the contextual features of arm $a$ at time $t$) is greater than or equal to the number of neighbors $k_t^a$ (where $k_t^a$ is the dynamically determined number of nearest neighbors for arm $a$ at time $t$). This ensures sufficient historical data is available for accurate neighbor selection and reward estimation.

---

**Algorithm 2** Adaptive $k$-NN integration for LNUCB-TA

---

1: **Input:** Decision space $D$, Historical data $\mathcal{H}$, $\theta_{\min}$ and $\theta_{\max}$ to determine the number of neighbors
2: Observe context $X_t$, Reward history $Z_t$
3: **for each** arm $a$ in $\mathcal{A}$ at time $t$ **do**
4:     Compute variance of rewards $\text{Var}(z_t^a)$                ▷ Reward variance
5:     $k_t^a = \theta_{\min} + (\theta_{\max} - \theta_{\min}) \times \text{Var}(z_t^a)$
6:     **if** $\text{len}(x_t^a) \geq k_t^a$ **then**
7:         $f_{k,t}^a(x_t^a, z_t^a) = k\text{-NN}_{k,t}^a(x_t^a, z_t^a)$           ▷ $k$-NN-score
8:         Estimated reward $= l_t^a(x_t^a) + f_{k,t}^a(x_t^a, z_t^a)$     ▷ Reward estimation
9:         Model update $=$ Estimated reward $+ \text{UCB}_t^a$
10:         Select arm $a$ with the highest updated model prediction
11:     **end if**
12: **end for**

---

In Algorithm 2, the variance in rewards for each arm at time $t$, $\text{Var}(z_t^a)$, drives the adaptive selection of $k$, which influences the depth of historical data utilized for the $k$-NN based prediction. The $k$ value dynamically adjusts between predefined minimum ($\theta_{\min}$) and maximum ($\theta_{\max}$) thresholds. The selection of $\theta_{\min}$ and $\theta_{\max}$ is determined through hyper-parameter tuning as they define the range for $k$, based on the observed variability of rewards, where

- **Low Variance:** Indicates stable reward patterns, suggesting that fewer historical data points are sufficient for accurate predictions. This stability allows the model to maintain a smaller $k$, closer to the minimum threshold, optimizing computational efficiency while maintaining predictive accuracy.

- **High Variance:** Reflects irregular or unpredictable reward patterns, necessitating a larger $k$ to incorporate a broader historical context. This expanded view helps to mitigate the impact of variability, enhancing the robustness of reward predictions.

Furthermore, unlike existing nonlinear approaches that use a static $k$ or involve searching over the preceding time steps $k \in [1, t-1]$ (Park et al., 2014; Reeve et al., 2018), our proposed model utilizes a data driven approach for selecting $k$. The algorithm achieves a time complexity of $O(t)$, which can reach $O(1)$ per update in the optimal case, significantly decreasing time complexity compared to the function optimization techniques used in $k$-NN UCB and $k$-NN KL-UCB.

### 3.3 TEMPORAL ATTENTION

**Intuition.** Our model replaces static exploration parameters with an attention-based mechanism, which allows for dynamic adjustment of exploration efforts based on time-dependent changes (temporal) and distinct reward patterns across different arms or contexts (spatial). The proposed method analyzes global performance across all arms, specific reward patterns of individual arms, and the frequency of arm selections, dynamically adjusting $\alpha$ for each arm at each time step. This innovation leads to consistent results, independent of the initial choice of the exploration rate.

---

**Algorithm 3** Temporal attention-based exploration rate for LNUCB-TA

---

1: **Input:** $\alpha_0$, $N_t^a$ (number of times arm $a$ played up to $t$), $g$ as global average of rewards, $n_t^a$ as mean average of each arm, $\kappa$ as weight factor
2: **for each** arm $a$ in $\mathcal{A}$ at time $t$ **do**
3: $\quad n_t^a = \frac{1}{N_t^a} \sum_{\hat{Y}_s^a \in z_t^a} \hat{Y}_s^a$ $\qquad\qquad\qquad\qquad$ ▷ Local attention for arm $a$
4: $\quad \alpha_{N_t^a} = \frac{\alpha_0}{(N_t^a + 1)} \cdot (\kappa g + (1 - \kappa) n_t^a)$ $\qquad$ ▷ Attention based exploration factor
5: $\quad$ Update $\hat{UCB}_t^a$
6: **end for**

---

**Method.** As shown in Algorithm 3, the attention-based $\alpha_{N_t^a}$ dynamically decreases as the frequency of arm selection increases, signifying a shift from exploration to exploitation, which reflects a reduction in uncertainty about the performance of each arm. Parallelly, increase in local reward for specific arms further tailor the exploration factor, enabling focused investigation of arms showing promising trends. This mechanism adeptly balances exploration and exploitation by adapting to both overall performance and individual arm dynamics, thus providing significantly more consistent results where traditional MAB models falter.

## 4 THEORETICAL ANALYSIS

**Theorem 1** (Regret bound). *Suppose the noise $|\xi_t^a|$ is bounded by $\sigma$ ($|\xi_t^a| \leq \sigma$), the true parameter vector $(\mu^a)^*$ has a norm bounded by $W$ ($\|(\mu^a)^*\| \leq W$), and the context vectors $x$ are bounded such that $\|x\| \leq B$ for all $x \in D$, and let $\lambda = \frac{\sigma^2}{W^2}$. Then $\beta_t^a$ can be defined as:*

$$\beta_t^a := \sigma^2 \left( 2 + 4d \log \left( 1 + \frac{TB^2W^2}{d} + \frac{\sum_{a=1}^{A} T^a (u_{t,k}^a)^2}{d} \right) + 8 \log \left( \frac{4}{\delta} \right) \right), \tag{8}$$

*with probability greater than $1 - \delta$, for all $t \geq 0$,*

$$R_T \leq b\sigma \sqrt{T \left( d \log \left( 1 + \frac{TB^2W^2}{d\sigma^2} + \frac{\sum_{a=1}^{A} T^a (u_{t,k}^a)^2}{d\sigma^2} \right) + \log \left( \frac{4}{\delta} \right) \right)}. \tag{9}$$

*where $\sigma^2$ represents the total variance accounting for both the linear component and the additional variance from the $k$-NN model, $\delta$ is the probability with which the confidence bounds are held, $b$ is an absolute constant, and $\sum_{a=1}^{A} T^a (u_{t,k}^a)^2$ represents the sum of the squared uncertainties for each arm $a$, capturing the influence of $k$-NN's neighborhood-based uncertainty for each specific arm. This sum is scaled by the number of times each arm $a$ is played $T^a$, where $\sum_{a=1}^{A} T^a \leq T$ as not all arms may utilize the $k$-NN adjustment at every time step. This sum represents an upper bound, capturing the maximum possible contribution from the $k$-NN component. Given these conditions, the simplified regret bound for LNUCB-TA gives $R_T = \mathcal{O}(\sqrt{dT \log T})$, which by absorbing logarithmic factors into $\tilde{\mathcal{O}}$, we can state*

$$R_T = \tilde{\mathcal{O}}(\sqrt{dT}). \tag{10}$$

This bound demonstrates that LNUCB-TA achieves sub-linear regret, highlighting its diminishing regret growth rate over time, contrasting with linear regret, where regret scales linearly with time steps. To prove this Theorem, we need to establish two critical propositions as outlined below:

**Proposition 1** (Uniform confidence bound). *Let $\delta > 0$. We have*

$$\Pr\left(\forall t, (\mu^a)^* \in BALL_t^a\right) \geq 1 - \delta. \tag{11}$$

*The second key proposition in analyzing LNUCB-TA involves demonstrating that, provided the aforementioned high-probability event occurs, the growth of the regret can be effectively controlled. Let us define the instantaneous regret as*

$$regret_t^a = (\mu^a)^* \cdot (x_t^a)^* + k\text{-}NN_{k,t}^a\left((x_t^a)^*, (z_t^a)^*\right) - \left((\mu^a)^* \cdot x_t^a + k\text{-}NN_{k,t}^a(x_t^a, z_t^a)\right). \tag{12}$$

*The following proposition provides an upper bound on the sum of the squares of the instantaneous regret.*

**Proposition 2** (Sum of squares regret bound). *Suppose $\|x\| \leq B$ for all $x \in D$, as we can suppose $(\mu^a)^* \in BALL_t^a$ for all $t$. Then, the sum of the squares of instantaneous regret for each arm $a$ over time is bounded as*

$$\sum_{t=0}^{T-1} (regret_t^a)^2 \leq 8\beta_t^a d \log\left(1 + \frac{TB^2}{d\lambda} + \frac{\sum_{a=1}^{A} T^a (u_{t,k}^a)^2}{d\lambda}\right). \tag{13}$$

*The cumulative squared regret bound is given by*

$$R_T = \sum_{a=1}^{A} \sum_{t=0}^{T-1} regret_t^a \leq \sqrt{T \sum_{t=0}^{T-1} (regret_t^a)^2}$$
$$\leq \sqrt{8T\beta_t^a d \log\left(1 + \frac{TB^2}{d\lambda} + \frac{\sum_{a=1}^{A} T^a (u_{t,k}^a)^2}{d\lambda}\right)}. \tag{14}$$

**Theorem 2** (Temporal exploration-exploitation balance). *Given a set of arms $\{1, 2, \ldots, A\}$ in a MAB problem, where each arm $a$ has a set of observed rewards denoted by $Y_t = (Y_t^a)_{a \in \mathcal{A}}$ in $\mathbb{R}^A$, and $N_t^a$ is the number of times arm $a$ has been selected up to time $t$. An attention mechanism can be designed, which dynamically updates the exploration parameter $\alpha$ according to the formula*

$$\alpha_{N_t^a} = \frac{\alpha_0}{N_t^a + 1} \cdot \left(\kappa g + (1 - \kappa) n_t^a\right), \tag{15}$$

*where $g$ represents the global attention derived from the average rewards across all arms, $n_t^a$ represents local attention derived from the average reward of arm $a$ at time $t$, and $\kappa$ is a weighting factor that balances global and local attention components.*

## 5   RESULTS

We have evaluated LNUCB-TA on a benchmark news recommendation dataset with 10,000 entries, each with 102 features. The first feature indicates one of ten news articles, the second represents user engagement (click/no click), and the remaining features provide contextual information (Li et al., 2010; 2011). Both the estimated reward and its variability serve as critical metrics in our analysis. Additional validations using other datasets and a broader comparison of metrics are provided in the Appendix B, offering a comprehensive view of the model's applicability across different scenarios.

Figure 1 provides a comparative analysis over 800 steps, showcasing cumulative and mean rewards of LNUCB-TA against 11 state-of-the-art (SOTA) MAB models, including enhanced Epsilon Greedy, BetaThompson, and Lin Thompson models with our adaptive $k$-NN method and a temporal attention mechanism. Each model was tested under six exploration settings to determine optimal performance, ensuring a rigorous comparison. The mean reward graph in Figure 1(b) provides further insights into the efficiency of the models at each step. The LNUCB-TA model has demonstrated rapid convergence to higher mean rewards, maintaining leading performance throughout the trials. Notably, while models such as $k$-NN KL-UCB and LinUCB show competitive performance initially, they do not sustain high rewards as consistently as LNUCB-TA. Additionally, the enhancements introduced through Algorithm 2 and an attention mechanism to traditional models have also resulted in performance improvements (please refer to Appendix B). However, these improvements do not reach the level achieved by LNUCB-TA.

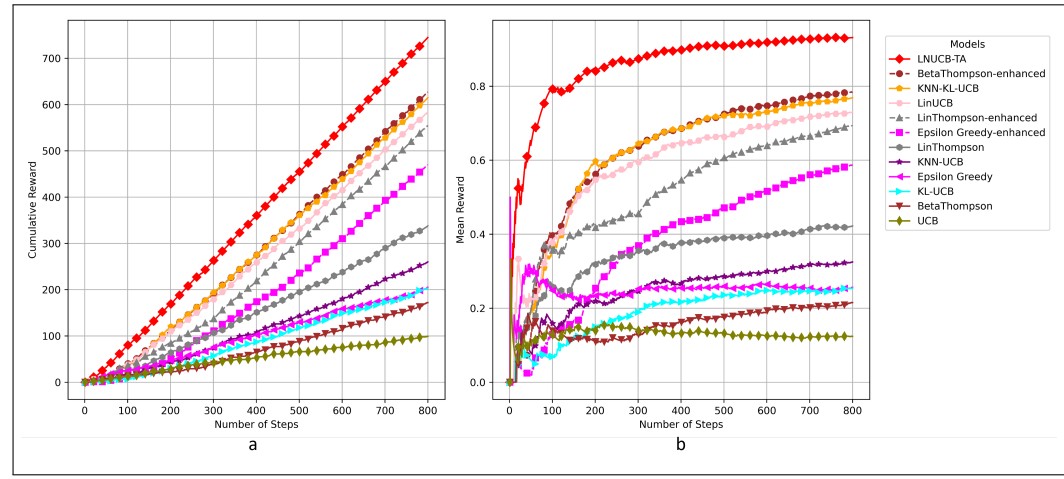

Figure 1: (a) cumulative rewards over 800 steps for **LNUCB-TA** and other SOTA models, demonstrating LNUCB-TA's superior performance. (b) mean rewards per time step, highlighting LNUCB-TA's rapid convergence and consistent high performance.

Table 2: Comparative analysis of **LNUCB-TA** against conventional linear, nonlinear, and vanilla combination model. It contrasts LNUCB-TA's superior cumulative and mean rewards with those of solely linear (LinUCB), non-linear ($k$-NN UCB), and basic linear-non-linear combinations (((Lin+$k$-NN)-UCB) across various exploration rates, demonstrating enhanced stability and effectiveness in dynamic decision-making environments.

| Model | Exploration Rate ($\alpha/\rho$) | Cumulative Reward | Mean Reward | Run Time (s) | Std Dev of Mean Reward |
|---|---|---|---|---|---|
| (Lin+$k$-NN)-UCB | 0.1 | 662 | 0.83 | 715.02 | 0.35 |
| (Lin+$k$-NN)-UCB | 1 | 617 | 0.77 | 733.72 | 0.35 |
| (Lin+$k$-NN)-UCB | 10 | 160 | 0.20 | 758.82 | 0.35 |
| LinUCB | 0.1 | 567 | 0.71 | 8.09 | 0.30 |
| LinUCB | 1 | 424 | 0.53 | 8.73 | 0.30 |
| LinUCB | 10 | 98 | 0.12 | 5.97 | 0.30 |
| $k$-NN UCB | 0.1 | 195 | 0.24 | 459.71 | 0.05 |
| $k$-NN UCB | 1 | 192 | 0.24 | 434.08 | 0.05 |
| $k$-NN UCB | 10 | 260 | 0.33 | 457.07 | 0.05 |
| **LNUCB-TA** | 0.1 | **741** | **0.93** | 324.5 | **0.01** |
| **LNUCB-TA** | 1 | **752** | **0.94** | 293.83 | **0.01** |
| **LNUCB-TA** | 10 | **752** | **0.94** | 297.28 | **0.01** |

Complementing Figure 1, Table 2 contrasts the performance of LNUCB-TA with solely linear models (LinUCB), solely nonlinear models ($k$-NN UCB), and a basic linear-nonlinear combination (((Lin+$k$-NN)-UCB) across various exploration rates. This table demonstrates that at lower exploration rates (0.1 and 1), linear models outperform nonlinear models, whereas at a higher exploration rate (10), nonlinear models excel. The basic combination generally surpasses both linear and nonlinear models at exploration rates of 0.1 and 1 but performs worse than nonlinear models at an exploration rate of 10. However, our hybrid model, LNUCB-TA, consistently outperforms all these models at every exploration rate, demonstrating superior reward accumulation and greater operational efficiency. It also requires less time compared to the vanilla combinations, highlighting the refined efficacy and efficiency of LNUCB-TA in dynamically adjusting to complex environments.

**Ablation study.** We have assessed the impact of integrating our novel components through various model variants, as shown in Figure 2. Model a represents the base LinUCB model. Model b, which incorporates the temporal attention mechanism, significantly enhances reward consistency, reducing the standard deviation from 0.32 (Model a) to 0.02. This indicates that dynamic adjustment of the exploration parameter, informed by historical data relevance, effectively stabilizes reward outcomes. Model c, which implements the adaptive $k$-NN approach, increases average mean rewards from 0.37 to 0.62 by optimizing the number of neighbors based on observed reward variance, capturing more nuanced patterns and improving prediction accuracy. While Model b ensures robustness against environmental fluctuations, Model c, despite its higher average reward, exhibits greater variability. Model d (LNUCB-TA), integrating both temporal attention and adaptive $k$-NN, achieves the highest average mean reward (0.90) and median reward (0.91), with the greatest consistency among all models tested. This demonstrates that combining these enhancements effectively balances exploration and exploitation, setting a new standard for adaptability and precision in dynamic MAB environments.

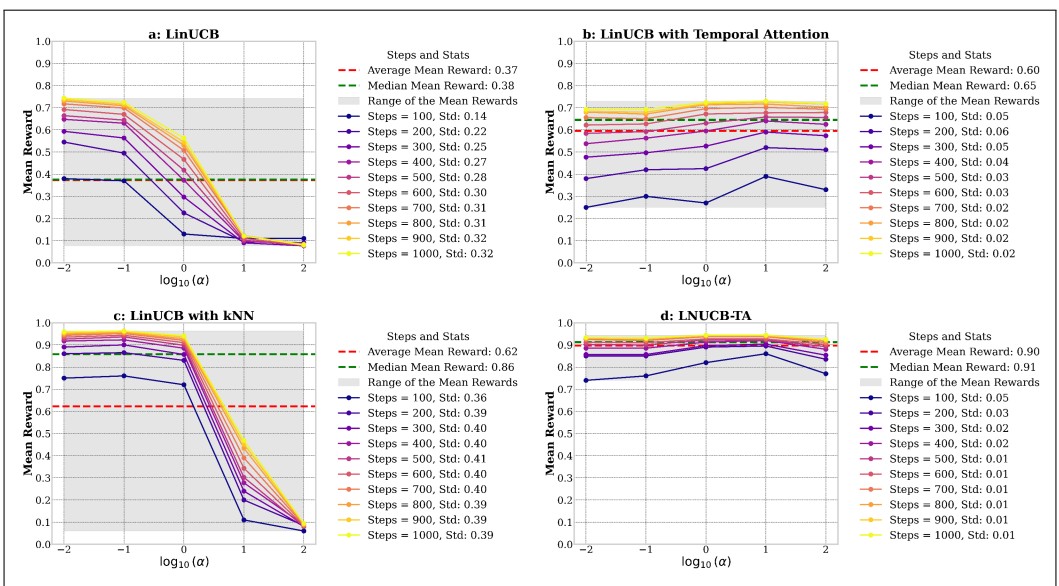

Figure 2: Impact of integrating the novel components. Model a is the base LinUCB model. Model b incorporates the temporal attention mechanism, significantly enhancing consistency. Model c implements the adaptive $k$-NN approach, increasing average mean rewards. Model d (**LNUCB-TA**) integrates both temporal attention and adaptive $k$-NN, achieving the highest average and median rewards with the greatest consistency.

# 6 CONCLUSION

In this paper, we address a hybrid contextual MAB problem within the joint space of context features and reward history by introducing a distinctive synthesis of linear and nonlinear algorithms, named LNUCB-TA, which features an innovative nonlinear estimation along with an attention-based exploration mechanism. Our proposed nonlinear component, adaptive $k$-NN, enhances reward predictions by continuously adapting to changes in reward history and feature vectors. The temporal attention mechanism further refines this process by dynamically balancing exploration and exploitation, adjusting exploration factors in real-time based on data variations. These enhancements have shown the potential to improve performance of different MAB model regardless of the underlying model, providing a robust framework for complex decision-making tasks. We also prove that the regret of LNUCB-TA is optimal up to $R_T = O(\sqrt{dT \log(T)})$, demonstrating a sub-linear regret.

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

# A APPENDIX

## A.1 PROOFS

**Proof sketch.** This section provides a structured and detailed exposition of the proofs for Theorems 1 and 2. For Theorem 1, the proof is comprehensive and requires the establishment of the two critical propositions 1 and 2. We begin with an overview of the model's parameters and introduce definitions crucial for understanding the proofs. Next, we list the key assumptions that underpin the theorem and its supporting propositions. Following this, we detail and prove the supporting lemmas that provide the necessary groundwork for the propositions. Using these lemmas, we rigorously prove each proposition, which directly supports the final proof of Theorem 1. Finally, after proving the sub-linear regret bound (Theorem 1), we prove Theorem 2 by relying on fundamental principles of the GALA concept.

**Model overview.** As discussed in Section 3 of the paper, we have a hybrid contextual MAB problem, where the expected reward for each arm $a$ at context $x_t$ is modeled through a linear and a nonlinear component defined as equation (1). This formulation seeks to effectively combine linear insights with the local history learned from the $k$-NN approach, adjusting for historical reward data $z_t$, which comprises past rewards related to arm $a$ according to Algorithm 2. Additionally, the regret associated with each arm $a$ at time $t$ quantifies the difference between the reward that could have been achieved by selecting the optimal action and the reward actually received by equation (5). As a result, total regret is calculated as equation (6). This measure of regret reflects the performance difference and highlights the effectiveness of the decision policy in approximating the optimal action choices over time.

**Corollary 1** (Uncertainty region). *The essence of LNUCB-TA revolves around the concept of "optimism in the face of uncertainty" (Liu et al., 2024a; Kamiura & Sano, 2017; Lykouris et al., 2021; Li et al., 2010; Russo & Van Roy, 2013). Following (Chu et al., 2011, Section 8.3), the center of an uncertainty region, $BALL_t^a$ is $\hat{\mu}_t^a$, which is the solution of the following ridge regression problem:*

$$
\begin{aligned}
\hat{\mu}_t^a &= \arg\min_\theta \left\| (X_t^a)^T \theta - (Y_t^a - f_{k,t}^a(x_t^a, z_t^a)) \right\|_2^2 + \lambda \|\theta\|_2^2 \\
&= ((X_t^a)^T X_t^a + \lambda I)^{-1} (X_t^a)^T (Y_t^a - f_{k,t}^a(x_t^a, z_t^a)) \\
&= (\Sigma_t^a)^{-1} \sum_{t=0}^{t-1} X_t^a (Y_t^a - f_{k,t}^a(x_t^a, z_t^a)),
\end{aligned}
\tag{16}
$$

*where $\theta$ is the parameter vector being optimized, $\lambda$ is the regularization parameter, and $\Sigma_t^a = (X_t^a)^T X_t^a + \lambda I$ is the covariance matrix (Lattimore & Szepesvári, 2020, equation 20.1) updated to time $t$ for arm $a$, reflecting the context feature information and the regularization term.*

**Definition 1.** For LNUCB-TA, the shape of the region $BALL_t^a$ following corollary 1 is defined through the feature covariance $\Sigma_t^a$. Precisely, the uncertainty region, or *confidence ball*, is defined as:

$$
BALL_t^a = \{\mu \mid (\mu - \hat{\mu}_t^a)^T \Sigma_t^a (\mu - \hat{\mu}_t^a) \le \beta_t^a\}.
\tag{17}
$$

**Corollary 2** (Uncertainty of nonlinear estimation). *Following (Reeve et al., 2018, Section 3.1), for each context $x_t^a$ in $\mathcal{X}$ and each arm $a$, at a given time step $t \in [n]$ and with access to the reward history up to $t$, represented as $\mathcal{Z}_t$, we define an enumeration of indices from $[t-1]$ as $\{\tau_{t,q}^a(x_t^a)\}_{q \in [t-1]}$ for each arm $a$ as*

$$
\rho((x_t^a, z_t^a), (X_{\tau_{t,q}^a(x_t^a)}, Z_{\tau_{t,q}^a(x_t^a)})) \le \rho((x_t^a, z_t^a), (X_{\tau_{t,q+1}^a(x_t^a)}, Z_{\tau_{t,q+1}^a(x_t^a)})).
\tag{18}
$$

*This enumeration is ordered such that for each $q \le t - 2$, where $q$ is a numeric $\mathbb{N}$, $X_{\tau_{t,q}^a(x_t^a)}$ and $Z_{\tau_{t,q}^a(x_t^a)}$ are the historical contexts and rewards associated with arm $a$ at index $\tau_{t,q}^a(x_t^a)$. Given $k \in [t-1]$, $\Gamma_{t,k}^a(x_t^a)$ is defined as*

$$
\Gamma_{t,k}^a(x_t^a) = \{\tau_{t,q}^a(x_t^a) : q \in [k]\} \subseteq [t-1].
\tag{19}
$$

*This set includes indices of the $k$ closest historical data points to the current feature vector $x_t^a$ for arm $a$, selected based on their proximity in the combined feature and reward space as measured by $\rho$. The maximum distance or uncertainty measure for arm $a$ at time $t$, $u_{t,k}^a(x_t^a)$, satisfies*

$$
u_{t,k}^a = \max\{\rho((x_t^a, z_t^a), (X_s, Z_s^a)) : s \in \Gamma_{t,k}^a(x_t^a)\} = \rho((x_t^a, z_t^a), (X_{\tau_{t,k}^a(x_t^a)}, Z_{\tau_{t,k}^a(x_t^a)})).
\tag{20}
$$

*This measure assesses the greatest distance between the current feature vector and reward data $(x_t^a, z_t^a)$ and those of the historical data within the nearest neighbors.*

**Corollary 3** (Arm-specific regular sets and measures). *Using (Reeve et al., 2018, Definition 1), we can state that in the extended metric space $(\mathcal{X} \times \mathcal{Z}, \rho)$, where $\mathcal{X} \times \mathcal{Z}$ represents the joint space of context features and reward history for arm $a$, and $\rho$ is the metric, a subset $A \subset \mathcal{X} \times \mathcal{Z}$ is a $(c_0, r_0^a)$ regular set if for all $(x^a, z^a) \in A$ and all $r \in (0, r_0^a)$,*

$$v^a(A \cap BALL^a((x^a, z^a); r)) \geq c_0 \cdot v^a(BALL^a((x^a, z^a); r)). \tag{21}$$

*A measure $\nu^a$ with $\mathrm{supp}(\nu^a) \subset \mathcal{X} \times \mathcal{Z}$ is a $(c_0, r_0^a, \nu_{min}^a, \nu_{max}^a)$ regular measure with respect to $v^a$ if $\mathrm{supp}(\nu^a)$ is a $(c_0, r_0^a)$-regular set with respect to $v^a$ and $\nu^a$ is absolutely continuous with respect to $v^a$ with Radon-Nikodym derivative (Folland, 1999, Theorem 3.8) as*

$$v^a(x^a, z^a) = \frac{d\nu^a(x^a, z^a)}{dv^a(x^a, z^a)}, \tag{22}$$

*ensuring*

$$\nu_{min}^a \leq v^a(x^a, z^a) \leq \nu_{max}^a. \tag{23}$$

**Assumption 1** (Arm-specific dimension assumption). Applying (Rigollet & Zeevi, 2010, Section 2.2), we can assume that for each arm $a \in \{1, \ldots, A\}$, there exist constants $C_d$, $d$, and $R_X^a > 0$ such that for all $(x^a, z^a) \in \mathrm{supp}(\nu^a)$ and $r \in (0, R_X^a)$, it holds

$$\nu^a(\mathrm{BALL}^a((x^a, z^a); r)) \geq C_d \cdot r^d. \tag{24}$$

Here, $r$ represents the radius of the ball in the joint space of features and reward history for arm $a$, indicating the scale of the local neighborhood around $(x^a, z^a)$ considered for the measure. The $\mathrm{BALL}^a((x^a, z^a); r)$ highlights the dependency on both context and past rewards within this radius. To prove this assumption, we shall follow a corollary followed from (Eftekhari & Wakin, 2015, Lemma 12).

**Corollary 4** (Arm-specific dimension). *For each arm $a$, let $M \subseteq \mathbb{R}^D$ be a $C^\infty$-smooth compact sub-manifold of uniform dimension $d$ (Lee, 2006) with a defined reach $\tau^a$ (Federer, 1959), quantified based on (Niyogi et al., 2008) as*

$$\tau^a := \sup\left\{r > 0 : \forall j \in \mathbb{R}^D, \inf_{q \in M}\{\|j - q\|_2\} < r \implies \exists! \, p \in M, \|j - p\|_2 = \inf_{q \in M}\{\|j - q\|_2\}\right\}. \tag{25}$$

*This reach reflects the maximum radius such that for every point $j$ within this distance from the manifold $M$, there is a nearest point on the manifold, ensuring stable local geometric properties, supported by (Boissonnat et al., 2018, Lemma 7.2). If $\nu^a$ is a $(c_0, R_0^a, \nu_{min}^a, \nu_{max}^a)$ regular measure with respect to $V_M$, then $\nu^a$ satisfies assumption 1 with constants $R_X^a = \min\{\tau^a/4, R_0^a\}$, $d$, and $C_d = \nu_{min}^a \cdot c_0 \cdot v_d^a \cdot 2^{-d}$, where $v_d^a$ is the Lebesgue measure of the unit ball in $\mathbb{R}^d$.*

*Proof.* For each arm $a$, consider any point $(x^a, z^a) \in \mathrm{supp}(\nu^a)$ and radius $r \in (0, R_X^a)$. Applying the (Niyogi et al., 2008, Lemma 5.3), for arm $a$, the volume within the $\mathrm{BALL}_r((x^a, z^a))$ can be estimated by

$$V_M(\mathrm{BALL}_r((x^a, z^a))) \geq \left(1 - \frac{r^2}{4(\tau^a)^2}\right)^{\frac{d}{2}} \cdot v_d \cdot r^d. \tag{26}$$

This equation reflects the geometrical properties of the manifold within a local neighborhood around $(x^a, z^a)$, given the manifold's reach and dimensionality.

Moreover, since $\nu^a$ is $(c_0, R_0^a, \nu_{min}^a, \nu_{max}^a)$-regular using corollary 3, it holds

$$\nu^a(\mathrm{BALL}_r((x^a, z^a))) \geq \nu_{min}^a \cdot c_0 \cdot V_M(\mathrm{BALL}_r((x^a, z^a))) \tag{27}$$

Combining this with the volume estimation provided by corollary 4, we get

$$\nu^a(\mathrm{BALL}_r((x^a, z^a))) \geq \nu_{min}^a \cdot c_0 \cdot \left(1 - \frac{r^2}{4(\tau^a)^2}\right)^{\frac{d}{2}} \cdot v_d \cdot r^d, \tag{28}$$

$$\nu^a(\mathrm{BALL}_r((x^a, z^a))) \geq \nu_{min}^a \cdot c_0 \cdot v_d \cdot 2^{-d} \cdot r^d, \tag{29}$$

.

This calculation demonstrates that the measure $\nu^a$ within the ball $\text{BALL}_r((x^a, z^a))$ exceeds a lower bound that scales with $r^d$, the dimensionally-scaled radius of influence. This establishes the local density and regularity of $\nu^a$ around each point in its support, confirming the validity of the arm-specific dimension assumption for the manifold $M$.

**Assumption 2** (Bounded rewards assumption). For all time steps $t \in [n]$ and for each arm $a \in [A]$, the rewards $Y_t^a$ observed after integrating both linear and $k$-Nearest Neighbors ($k$-NN) adjustments are bounded within an interval assumed as

$$-1 \leq Y_t^a \leq 1. \tag{30}$$

**Assumption 3** (Confidence in parameter estimation). For all time steps $t \in [n]$ and for each arm $a \in [A]$, we shall assume that the true parameter vector $\mu^*$ resides within a confidence ball centered around the estimated parameter $\mu_t^a$. This confidence ball, denoted as $\text{BALL}_{(t,a)}$, is defined based on the estimation error and the uncertainty in the measurements up to time $t$, incorporating adjustments for both linear and nonlinear adjustments.

**Lemma 1** (Width of confidence Ball for LNUCB-TA). *Let $x \in D$. As $\mu$ belongs to $BALL_t^a$ for each arm $a$ and $x \in D$ according to assumption 3, then*

$$|(\mu - \hat{\mu}_t^a)^T x| \leq \sqrt{\beta_t^a x^T (\Sigma_t^a)^{-1} x}. \tag{31}$$

*This lemma follows (Agarwal et al., 2019, Lemma 6.8).*

*Proof.* Starting with the absolute value of the dot product of $(\mu - \hat{\mu}_t^a)$ and $x$, we get

$$|(\mu - \hat{\mu}_t^a)^T x|. \tag{32}$$

By utilizing the Cauchy-Schwarz inequality (Strang, 2022, Section 1.2), which states that for all vectors $u$ and $v$ in an inner product space, we have

$$|\langle u, v \rangle|^2 \leq \langle u, u \rangle \cdot \langle v, v \rangle, \tag{33}$$

where $\langle \cdot, \cdot \rangle$ is the inner product. Every inner product gives rise to a Euclidean $l_2$ norm, called the canonical or induced norm, where the norm of a vector $u$ is defined by

$$\|u\| := \sqrt{\langle u, u \rangle}. \tag{34}$$

By taking the square root of both sides of equation(34), the Cauchy-Schwarz inequality can be written in terms of the norm

$$|\langle u, v \rangle| \leq \|u\| \|v\|. \tag{35}$$

Moreover, the two sides are equal if and only if $u$ and $v$ are linearly dependent. Applying this inequality to $u = (\Sigma_t^a)^{1/2}(\mu - \hat{\mu}_t^a)$ and $v = (\Sigma_t^a)^{-1/2} x_t^a$, we get

$$|(\mu - \hat{\mu}_t^a)^T x| = |((\Sigma_t^a)^{1/2}(\mu - \hat{\mu}_t^a))^T (\Sigma_t^a)^{-1/2} x| \leq \|(\Sigma_t^a)^{1/2}(\mu - \hat{\mu}_{t,a})\| \cdot \|(\Sigma_t^a)^{-1/2} x\|$$
$$= \sqrt{(\mu - \hat{\mu}_t^a)^T \Sigma_t^a (\mu - \hat{\mu}_t^a)} \cdot \sqrt{x^T (\Sigma_t^a)^{-1} x}. \tag{36}$$

Since $\mu$ is assumed to be within the confidence set $\text{BALL}_t^a$ as assumption 3, we have

$$(\mu - \hat{\mu}_t^a)^T \Sigma_t^a (\mu - \hat{\mu}_t^a) \leq \beta_t^a, \tag{37}$$

plugging this back into equation (36), we can obtain

$$|(\mu - \hat{\mu}_t^a)^T x| \leq \sqrt{\beta_t^a} \cdot \sqrt{x^T (\Sigma_t^a)^{-1} x} = \sqrt{\beta_t^a x^T (\Sigma_t^a)^{-1} x}. \tag{38}$$

**Lemma 2** (Normalized width for LNUCB-TA)). *Fix $t \leq T$. As $(\mu^a)^* \in BALL_t^a$ based on assumption 3, we define*

$$w_t^a = \sqrt{(x_t^a)^T (\Sigma_t^a)^{-1} x_t^a}, \tag{39}$$

*which is the "normalized width" at time $t$ for arm $a$ in the direction of the chosen decision, then*

$$regret_t^a \leq 2 \min\left(\sqrt{\beta_t^a} w_t^a, 1\right) \leq 2\sqrt{\beta_T^a} \min(w_t^a, 1). \tag{40}$$

This lemma is inspired by the theoretical analysis of nonlinear bandits presented in (Dong et al., 2021), where the sample complexity for finding an approximate local maximum is discussed, leveraging the model complexity rather than the action dimension. Additionally, the approach to handling confidence bounds in linear bandits (Agrawal & Goyal, 2013; Li et al., 2010), provides a foundational understanding for the linear components of this work.

*Proof.* Let $\tilde{\mu} \in \mathrm{BALL}_t^a$, we define instantaneous regret as

$$
\begin{aligned}
\mathrm{regret}_t^a = (\mu^a)^T (x^a)^* - (\mu^a)^T x_t^a &\leq (\tilde{\mu} - (\mu^a)^*)^\top x_t^a \\
&= (\tilde{\mu} - \hat{\mu}_t^a)^\top x_t^a + (\hat{\mu}_t^a - (\mu^a)^*)^\top x_t^a.
\end{aligned}
\tag{41}
$$

For the sum of two inner products, the triangle inequality (Axler, 2015, Section 4.5) gives

$$
|(\tilde{\mu} - \hat{\mu}_t^a)^\top x_t^a + (\hat{\mu}_t^a - (\mu^a)^*)^\top x_t^a| \leq |(\tilde{\mu} - \hat{\mu}_t^a)^\top x_t^a| + |(\hat{\mu}_t^a - (\mu^a)^*)^\top x_t^a|,
\tag{42}
$$

and by using the given bound for $|(\mu - \hat{\mu}_t^a)^\top x|$ in lemma 1, we can obtain

$$
|(\tilde{\mu} - \hat{\mu}_t^a)^\top x_t^a| \leq \sqrt{\beta_t^a (x_t^a)^T (\Sigma_t^a)^{-1} x_t^a} = \sqrt{\beta_t^a} w_t^a,
\tag{43}
$$

$$
|(\hat{\mu}_t^a - (\mu^a)^*)^\top x_t^a| \leq \sqrt{\beta_t^a (x_t^a)^T (\Sigma_t^a)^{-1} x_t^a} = \sqrt{\beta_t^a} w_t^a.
\tag{44}
$$

Thus,

$$
|(\tilde{\mu} - \hat{\mu}_t^a)^\top x_t^a + (\hat{\mu}_t^a - (\mu^a)^*)^\top x_t^a| \leq 2\sqrt{\beta_t^a} w_t^a,
\tag{45}
$$

and since $-1 \leq Y_t^a \leq 1$ (assumption 2), the regret is at most 2, then

$$
\mathrm{regret}_t^a \leq 2\sqrt{\beta_t^a} w_t^a \leq \min(2\sqrt{\beta_t^a} w_t^a, 2).
\tag{46}
$$

Expressing it with 2 outside the minimum function for clarity and to align with the bound mentioned in assumption 2, satisfies

$$
\mathrm{regret}_t^a \leq 2\min(\sqrt{\beta_t^a} w_t^a, 1),
\tag{47}
$$

and as $\beta_t^a$ is non-decreasing over time (common in learning systems where confidence typically increases with more data), $\beta_T^a \geq \beta_t^a$ for any $t \leq T$. Thus, applying this monotonicity property of $\beta_t^a$,

$$
2\sqrt{\beta_t^a} \min(w_t^a, 1) \leq 2\sqrt{\beta_T^a} \min(w_t^a, 1),
\tag{48}
$$

which completes the proof.

**Lemma 3** (Determinant expansion). *We have*

$$
det(\Sigma_T^a) = det(\Sigma_0^a) \prod_{t=0}^{T-1} (1 + (w_t^a)^2 + \gamma e_{t,k}^a),
\tag{49}
$$

*where $e_{t,k}^a = \left( u_{t,k}^a \right)^2$. This lemma is structured based on (Agarwal et al., 2019, Lemma 6.8) and (Perrault et al., 2020, Theorem 1).*

*Proof.* By definition of $\Sigma_{t+1}^a$, we get

$$
\Sigma_{t+1}^a = \Sigma_t^a + x_t^a (x_t^a)^\top + \gamma e_{t,k}^a I,
\tag{50}
$$

where $\gamma$ helps to scale the identity matrix $I$ multiplied by the variance term $e_{t,k}^a$, which quantifies the uncertainty contributed by the $k$-NN predictions at each time step for arm $a$. Considering the determinant, we have

$$
det(\Sigma_{t+1}^a) = det(\Sigma_t^a + x_t^a (x_t^a)^\top + \gamma e_{t,k}^a I).
\tag{51}
$$

Then, a special case of "matrix determinant lemma" attributed to (Harville, 1998, Corollary 18.2.10), originated from (Sherman & Morrison, 1950) is utilized as equation (52)

$$
det(A + uv^\top) = det(A)(1 + v^\top A^{-1} u).
\tag{52}
$$

Applying the concept of equation (52) in equation (51), we can obtain

$$\det(\Sigma_{t+1}^a) = \det(\Sigma_t^a)\det\left(I + (\Sigma_t^a)^{-1/2}x_t^a(x_t^a)^\top(\Sigma_t^a)^{-1/2} + \gamma e_{t,k}^a(\Sigma_t^a)^{-1/2}I(\Sigma_t^a)^{-1/2}\right). \quad (53)$$

Then, by decomposing the calculation further, considering $v_t = (\Sigma_t^a)^{-1/2}x_t^a$ and $u_t = \gamma e_{t,k}^a I$,

$$\det(I + v_t v_t^\top + \gamma e_{t,k}^a(\Sigma_t^a)^{-1/2}I(\Sigma_t^a)^{-1/2}) = \det(I + v_t v_t^\top + \gamma e_{t,k}^a I). \quad (54)$$

Since $I$ is the identity matrix and commutes with any matrix, using the property that $\Sigma_t^{a-1/2}I\Sigma_t^{a-1/2} = I$ due to normalization, and where $v_t^a = (\Sigma_t^a)^{-1/2}x_t^a$ based on the proof of lemma 2. Now we can observe $(v_t^a)^\top v_t^a = (w_t^a)^2$ and

$$(I + v_t^a(v_t^a)^\top)v_t^a = v_t^a + v_t^a((v_t^a)^\top v_t^a) = (1 + (w_t^a)^2)v_t^a. \quad (55)$$

For this reason $(1 + (w_t^a)^2)$ is an eigenvalue of $I + v_t^a(v_t^a)^\top$. Since $v_t^a(v_t^a)^\top$ is a rank one matrix, all other eigenvalues of $I + v_t^a(v_t^a)^\top$ equal 1. Hence, $\det(I + v_t^a(v_t^a)^\top) = (1 + (w_t^a)^2)$, is implies

$$\det(I + v_t^a(v_t^a)^\top + \gamma e_{t,k}^a I) = \det(I + (w_t^a)^2 + \gamma e_{t,k}^a), \quad (56)$$

which gets

$$\det(\Sigma_{t+1}^a) = (1 + (w_t^a)^2 + \gamma e_{t,k}^a)\det(\Sigma_t^a). \quad (57)$$

Finally, iterating equation (57) from $t = 0$ to $T - 1$ gives

$$\det(\Sigma_T^a) = \det(\Sigma_0^a)\prod_{t=0}^{T-1}(1 + (w_t^a)^2 + \gamma e_{t,k}^a). \quad (58)$$

**Lemma 4** (Potential function bound). *Consider the sequence $x_0^a, \ldots, x_{T-1}^a$ such that $\|x_t^a\|_2 \leq B$ for all $t < T$, the potential function bound is given by*

$$
\begin{aligned}
\log\left(\frac{det(\Sigma_{T-1}^a)}{det(\Sigma_0^a)}\right) &= \log\left(det\left(I + \frac{1}{\lambda}\left(\sum_{t=0}^{T-1}x_t^a(x_t^a)^\top + \sum_{a=1}^{A}\gamma e_{t,k}^a I\right)\right)\right) \\
&= \log\left(det\left(I + \frac{1}{\lambda}\left(\sum_{t=0}^{T-1}x_t^a(x_t^a)^\top + \sum_{a=1}^{A}\gamma(u_{t,k}^a)^2 I\right)\right)\right) \\
&\leq d\log\left(1 + \frac{1}{d\lambda}\left(TB^2 + \sum_{a=1}^{A}T^a(u_{t,k}^a)^2\right)\right).
\end{aligned}
\quad (59)
$$

*Proof.* For $\Sigma_{T-1}^a$, we have

$$\Sigma_{T-1}^a = \Sigma_0^a + \sum_{t=0}^{T-1}x_t^a(x_t^a)^\top + \sum_{a=1}^{A}\gamma(u_{t,k}^a)^2 I. \quad (60)$$

Then, we use the identity that relates the determinant of a sum to the product of eigenvalues

$$\log\left(\frac{\det(\Sigma_{T-1}^a)}{\det(\Sigma_0^a)}\right) = \log\left(\det\left(I + (\Sigma_0^a)^{-1}\left(\sum_{t=0}^{T-1}x_t^a(x_t^a)^\top + \sum_{a=1}^{A}\gamma(u_{t,k}^a)^2 I\right)\right)\right), \quad (61)$$

which simplifies to

$$\log\left(\det\left(I + \frac{1}{\lambda}\left(\sum_{t=0}^{T-1}x_t^a(x_t^a)^\top + \sum_{a=1}^{A}\gamma(u_{t,k}^a)^2 I\right)\right)\right). \quad (62)$$

Let $\sigma_1, \ldots, \sigma_d$ be the eigenvalues of $\sum_{t=0}^{T-1}x_t^a(x_t^a)^\top + \sum_{a=1}^{A}\gamma(u_{t,k}^a)^2 I$. Applying the Arithmetic Mean-Geometric Mean (AM-GM) Inequality (Cvetkovski, 2012, Theorem 2.1), we can obtain

$$\text{Trace}\left(\sum_{t=0}^{T-1}x_t^a(x_t^a)^\top + \sum_{a=1}^{A}\gamma(u_{t,k}^a)^2 I\right) = \sum_{t=0}^{T-1}\|x_t^a\|^2 + A\gamma(u_{t,k}^a)^2. \quad (63)$$

Then, we shall assume $\sum_{t=0}^{T-1} \|x_t^a\|^2 \le TB^2$, and by summing the regularizing terms, we get

$$\sum_{i=1}^{d} \sigma_i \le TB^2 + \sum_{a=1}^{A} \gamma(u_{t,k}^a)^2. \tag{64}$$

Finally, using the equation (63),

$$\log\left(\det\left(I + \frac{1}{\lambda}\left(\sum_{t=0}^{T-1} x_t^a(x_t^a)^\top + \sum_{a=1}^{A} \gamma(u_{t,k}^a)^2 I\right)\right)\right) \tag{65}$$

$$= \log\left(\prod_{i=1}^{d}\left(1 + \frac{\sigma_i}{\lambda}\right)\right) \tag{66}$$

$$= \sum_{i=1}^{d} \log\left(1 + \frac{\sigma_i}{\lambda}\right) \le d\log\left(1 + \frac{1}{d\lambda}\left(TB^2 + \sum_{a=1}^{A} \gamma(u_{t,k}^a)^2\right)\right). \tag{67}$$

This inequality uses the AM-GM inequality in the form $\log\left(\prod_{i=1}^{d}\left(1 + \frac{\sigma_i}{\lambda}\right)\right) \le d\log\left(1 + \frac{\text{Trace}}{d\lambda}\right)$.

**Lemma 5** (Linear Operator). *Let $\Sigma_0^a$ be an initial covariance matrix, $x_t^a$ a feature vector for arm $a$ at time $t$, and $\gamma$ a scaling constant, and $u_{t,k}^a$ is defined as stated in the description. The operator*

$$\Sigma_{T-1}^a = \Sigma_0^a + \sum_{t=0}^{T-1} x_t^a(x_t^a)^\top + \gamma\sum_{a=1}^{A}(u_{t,k}^a)^2 I \tag{68}$$

*is a linear operator from $\mathbb{R}^d$ to $\mathbb{R}^d$, where $d$ is the dimension of the feature vectors.*

*Proof.* A linear operator in the context of linear algebra is a mapping $L : V \to W$ between two vector spaces $V$ and $W$ that satisfies the linearity conditions (Rudin et al., 1964):

- **Additivity:** $L(u + v) = L(u) + L(v)$ for any vectors $u, v \in V$.
- **Homogeneity:** $L(\alpha u) = \alpha L(u)$ for any scalar $\alpha$ and vector $u \in V$.

**Additivity:** For any vectors $u, v \in \mathbb{R}^d$,

$$\Sigma_{T-1}^a(u + v) = \Sigma_0^a(u + v) + \sum_{t=0}^{T-1} x_t^a(x_t^a)^\top(u + v) + \gamma\sum_{a=1}^{A}(u_{t,k}^a)^2 I(u + v) \tag{69}$$

$$= \Sigma_0^a(u) + \Sigma_0^a(v) + \sum_{t=0}^{T-1} x_t^a((x_t^a)^\top u + (x_t^a)^\top v) + \gamma\sum_{a=1}^{A}(u_{t,k}^a)^2(Iu + Iv) \tag{70}$$

$$= \Sigma_0^a(u) + \sum_{t=0}^{T-1} x_t^a(x_t^a)^\top u + \gamma\sum_{a=1}^{A}(u_{t,k}^a)^2 Iu + \Sigma_0^a(v) + \sum_{t=0}^{T-1} x_t^a(x_t^a)^\top v + \gamma\sum_{a=1}^{A}(u_{t,k}^a)^2 Iv \tag{71}$$

$$= \Sigma_{T-1}^a(u) + \Sigma_{T-1}^a(v). \tag{72}$$

**Homogeneity:** For any scalar $\alpha$ and vector $u \in \mathbb{R}^d$,

$$\Sigma_{T-1}^a(\alpha u) = \Sigma_0^a(\alpha u) + \sum_{t=0}^{T-1} x_t^a(x_t^a)^\top(\alpha u) + \gamma\sum_{a=1}^{A}(u_{t,k}^a)^2 I(\alpha u) \tag{73}$$

$$= \alpha\Sigma_0^a(u) + \alpha\sum_{t=0}^{T-1} x_t^a(x_t^a)^\top u + \alpha\gamma\sum_{a=1}^{A}(u_{t,k}^a)^2 Iu \tag{74}$$

$$= \alpha(\Sigma_0^a(u) + \sum_{t=0}^{T-1} x_t^a(x_t^a)^\top u + \gamma\sum_{a=1}^{A}(u_{t,k}^a)^2 Iu) \tag{75}$$

$$= \alpha\Sigma_{T-1}^a(u). \tag{76}$$

Since $\Sigma_{T-1}^a$ satisfies both additivity and homogeneity, it is a linear operator. Hence, the lemma 5 is proved.

**Corollary 5** (Self-normalized bound). *For each arm $a$, the reward is generated as*

$$Y_t^a = l_t^a + f_{k,t}^a(x_t^a, z_t^a) + \xi_t^a = \mu_t^a \cdot x_t^a + k\text{-NN}_{k,t}^a(x_t^a, z_t^a) + \xi_t^a. \tag{77}$$

*Here, $\xi_t^a$ is the noise term associated with arm $a$, which captures the inherent randomness in the rewards after accounting for both the linear model's predictions and the $k$-NN adjustments. This term remains conditionally $\delta$-sub-Gaussian.*

*Given the linear operator proved in lemma 5, the self-normalized bound, structured by (Abbasi-Yadkori et al., 2011, Theorem 1) and (Auer et al., 2002b), with the probability at least $1 - \delta$ is followed by*

$$\left\| \sum_{t=1}^{T} X_t^a \xi_t^a \right\|_{(\Sigma_t^a)^{-1}}^2 \le \sigma^2 \log \left( \frac{\det(\Sigma_t^a) \det(\Sigma_0^a)^{-1}}{\delta^2} \right), \tag{78}$$

*where $\xi_t^a$ encapsulates both inherent randomness and any deviation from $k$-NN estimates.*

### A.1.1 PROOF OF THEOREM 1

**Proof of proposition 1.** Consider the defined reward for each arm $a$ in equation (77) in corollary 5, the deviation of the estimated parameter $\mu_t^a$ from the true parameter $(\mu^a)^*$ is calculated as

$$\mu_t^a - (\mu^a)^* = \Sigma_t^{a-1} \left( \sum_{t=0}^{t-1} x_t^a \left( (\mu^a)^* + \xi_t^a + k\text{-NN}_{k,t}^a(x_t^a, z_t^a) \right) x_t^a - \lambda \Sigma_t^{a-1} \left( (\mu^a)^* \right) \right). \tag{79}$$

By utilizing lemma 2, we can obtain

$$\sqrt{(\mu_t^a - (\mu^a)^*)^\top \Sigma_t^a (\mu_t^a - (\mu^a)^*)} = \| \Sigma_t^{a\,1/2} (\mu_t^a - (\mu^a)^*) \| \tag{80}$$

$$\le \| \lambda \Sigma_t^{a-1/2} (\mu^a)^* \| + \| \Sigma_t^{a-1/2} \sum_{t=0}^{t-1} \xi_t^a x_t^a \| \tag{81}$$

$$\le \sqrt{\lambda} \| (\mu^a)^* \| + \sqrt{2\sigma^2 \log \left( \frac{\det(\Sigma_t^a) \det(\Sigma_0)^{-1}}{\delta} \right)}. \tag{82}$$

Using the triangle inequality and considering $\Sigma_t^{a-1}$ as always positive definite, implying $(\Sigma_t^a)^{-1} \ge \frac{1}{\lambda} I$. Our goal is to lower bound $\Pr(\forall t; (\mu^a)^* \in \text{BALL}_t^a)$. At $t = 0$, by our initial choice, $\text{BALL}_0^a$ contains $(\mu^a)^*$, hence $\Pr((\mu^a)^* \notin \text{BALL}_0^a) = 0$. For $t \ge 1$, we designate the failure probability for the $t$-th event as

$$\delta_t = \left( \frac{3}{\pi^2} \right) \frac{1}{t^2} \cdot 2\delta. \tag{83}$$

Using the preceding results and a union bound, gives us an upper bound on the cumulative failure probability as

$$1 - \Pr(\forall t; (\mu^a)^* \in \text{BALL}_t^a) = \Pr(\exists t; (\mu^a)^* \notin \text{BALL}_t^a) \le \sum_{t=1}^{\infty} \left( \frac{1}{t^2} - \frac{3}{2^t} \right) 2\delta = \frac{1}{2} \cdot 2\delta = \delta. \tag{84}$$

**Proof of Proposition 2.** Considering assumption 3 for all time steps $t$ and arms $a$, we start by expressing the sum of squared regrets

$$\sum_{t=0}^{T-1} (\text{regret}_t^a)^2 \le \sum_{t=0}^{T-1} 4\beta_t^a \min((w_t^a)^2, 1) \tag{85}$$

$$\le 4\beta_T^a \sum_{t=0}^{T-1} \min((w_t^a)^2, 1) \le \max\{8, \frac{4}{\log 2}\} \beta_T^a \sum_{t=0}^{T-1} \log(1 + (w_t^a)^2 + \gamma(u_{t,k}^a)^2) \tag{86}$$

$$\leq 8\beta_T^a \log\left(\frac{\det(\Sigma_{T-1}^a)}{\det(\Sigma_0^a)}\right) = 8\beta_T^a d \log\left(1 + \frac{TB^2}{d\lambda} + \frac{\sum_{a=1}^A \sum_{t=0}^{T-1}(u_{t,k}^a)^2}{d\lambda}\right) \tag{87}$$

The first inequality follows from lemma 2. The second is from since $\beta_t^a$ is an increasing function of $t$, $\beta_t^a \leq \beta_{t+1}^a$ for all $t$ where $0 \leq t < T-1$ and $\sum_{t=0}^{T-1}\beta_t^a = \beta_T^a$. The third follows that for $0 \leq y \leq 1$, the inequality $y \geq \log(1+y) \geq \frac{y}{1+y} \geq \frac{y}{2}$ holds, and specifically for $(w_t^a)^2$ within these bounds, we have

$$(w_t^a)^2 + \gamma(u_{t,k}^a)^2 \leq 2\log(1 + (w_t^a)^2 + \gamma(u_{t,k}^a)^2). \tag{88}$$

When $(w_t^a)^2 > 1$, the relationship shifts to

$$4\beta_T^a = \frac{4}{\log 2}\beta_T^a \log 2 \leq \frac{4}{\log 2}\beta_T^a \log(1 + (w_t^a)^2 + \gamma(u_{t,k}^a)^2). \tag{89}$$

The equation (88) follow lemma 3, and equation (89) follows lemma 4.

With the proof of the two propositions, we can conclude the Theorem 1, showing the regret bound as

$$R_T \leq b\sigma\sqrt{T\left(d\log\left(1 + \frac{TB^2W^2}{d\sigma^2} + \frac{\sum_{a=1}^A T^a(u_{t,k}^a)^2}{d\sigma^2}\right) + \log\left(\frac{4}{\delta}\right)\right)}. \tag{90}$$

To prove the sub-linear regret bound, we need to analyze and simplify the dominant terms within the regret bound.

**Dominant term analysis.** To identify the dominant term, we carefully analyze how each term scales with $T$:

**Term 1:** $\frac{TB^2W^2}{d\sigma^2}$, which grows linearly with $T$.

**Term 2:** $\frac{\sum_{a=1}^A T^a(u_{t,k}^a)^2}{d\sigma^2}$, which scales with $\sum_{a=1}^A T^a$, which is at most $T$, as not all arms may utilize the $k$-NN adjustment at every time step. This sum represents an upper bound, capturing the maximum possible contribution from the $k$-NN component.

**Simplifying the logarithmic term.** Considering both terms inside the logarithm, we have

$$\log\left(1 + \frac{TB^2W^2}{d\sigma^2} + \frac{\sum_{a=1}^A T^a(u_{t,k}^a)^2}{d\sigma^2}\right), \tag{91}$$

which for large $T$, we can approximate the logarithm as

$$\log\left(1 + \frac{TB^2W^2}{d\sigma^2} + \frac{\sum_{a=1}^A T^a(u_{t,k}^a)^2}{d\sigma^2}\right) \approx \log\left(\frac{T(B^2W^2 + \sum_{a=1}^A(u_{t,k}^a)^2)}{d\sigma^2}\right). \tag{92}$$

**Refined bound.** Given that both terms grow with $T$, for large $T$, we have

$$\log\left(1 + \frac{T(B^2W^2 + \sum_{a=1}^A(u_{t,k}^a)^2)}{d\sigma^2}\right). \tag{93}$$

So, the regret bound becomes

$$R_T \leq b\sigma\sqrt{T\left(d\log\left(\frac{T(B^2W^2 + \sum_{a=1}^A(u_{t,k}^a)^2)}{d\sigma^2}\right)\right)}. \tag{94}$$

And for large $T$, we have

$$R_T = O\left(\sigma\sqrt{dT\log T}\right). \tag{95}$$

Without assuming any term is negligible, the regret of LNUCB-TA is optimal up to

$$R_T = O(\sqrt{dT\log T}). \tag{96}$$

And by absorbing logarithmic factors into $\tilde{O}$, we can state

$$R_T = \tilde{O}(\sqrt{dT}). \tag{97}$$

This result establishes the optimality and efficiency of LNUCB-TA in achieving sub-linear regret, proving Theorem 1.

### A.1.2 PROOF THEOREM 2

We begin by considering the exploration parameter $\alpha_{N_t^a}$, which is dynamically updated as:

$$\alpha_{N_t^a} = \frac{\alpha_0}{N_t^a + 1} \cdot (\kappa g + (1 - \kappa) n_t^a), \tag{98}$$

where $g$ represents the global attention and $n_t^a$ is the local attention for arm $a$ up to time $t$. Specifically, the global attention $g$ is defined as:

$$g = \frac{1}{A} \sum_{a=1}^{A} \overline{Y}^a, \tag{99}$$

with $A$ being the number of arms and $\overline{Y}^a$ the average reward of arm $a$. The local attention $n_t^a$ is given by:

$$n_t^a = \frac{1}{N_t^a} \sum_{s=1}^{t-1} \hat{Y}_s^a, \tag{100}$$

where $N_t^a$ is the number of times arm $a$ has been selected up to time $t$, and $\hat{Y}_s^a$ is the reward observed from arm $a$ at time $s$. Our goal is to compute $\frac{d\alpha_{N_t^a}}{dN_t^a}$, representing the rate of change of the exploration parameter as $N_t^a$ increases, i.e., how the system shifts from exploration to exploitation as more pulls are made on arm $a$.

First, applying the product rule to differentiate $\alpha_{N_t^a}$ with respect to $N_t^a$, we have:

$$\frac{d\alpha_{N_t^a}}{dN_t^a} = \frac{d}{dN_t^a} \left( \frac{\alpha_0}{N_t^a + 1} \cdot (\kappa g + (1 - \kappa) n_t^a) \right). \tag{101}$$

This can be expanded as:

$$\frac{d\alpha_{N_t^a}}{dN_t^a} = \frac{\alpha_0}{N_t^a + 1} \cdot \frac{d}{dN_t^a} (\kappa g + (1 - \kappa) n_t^a) + (\kappa g + (1 - \kappa) n_t^a) \cdot \frac{d}{dN_t^a} \left( \frac{\alpha_0}{N_t^a + 1} \right). \tag{102}$$

Next, we compute the derivatives of each term separately. Since $g$ is the global attention and does not depend on $N_t^a$, its derivative is zero, and we only need to differentiate $n_t^a$. Using the quotient rule, we compute the derivative of $n_t^a$ as follows:

$$n_t^a = \frac{1}{N_t^a} \sum_{s=1}^{t-1} \hat{Y}_s^a, \tag{103}$$

hence,

$$\frac{dn_t^a}{dN_t^a} = -\frac{1}{(N_t^a)^2} \sum_{s=1}^{t-1} \hat{Y}_s^a. \tag{104}$$

Substituting this into the derivative of the first term:

$$\frac{d}{dN_t^a}\left(\kappa g + (1-\kappa)n_t^a\right) = (1-\kappa)\cdot\left(-\frac{1}{(N_t^a)^2}\sum_{s=1}^{t-1}\hat{Y}_s^a\right). \tag{105}$$

For the second term, we differentiate $\frac{\alpha_0}{N_t^a+1}$ with respect to $N_t^a$:

$$\frac{d}{dN_t^a}\left(\frac{\alpha_0}{N_t^a+1}\right) = -\frac{\alpha_0}{(N_t^a+1)^2}. \tag{106}$$

Now, substituting these results back into the expression for $\frac{d\alpha_{N_t^a}}{dN_t^a}$, we obtain:

$$\frac{d\alpha_{N_t^a}}{dN_t^a} = \frac{\alpha_0}{N_t^a+1}\cdot(1-\kappa)\cdot\left(-\frac{1}{(N_t^a)^2}\sum_{s=1}^{t-1}\hat{Y}_s^a\right) - \frac{\alpha_0}{(N_t^a+1)^2}\cdot(\kappa g + (1-\kappa)n_t^a). \tag{107}$$

Expanding $n_t^a$ in the second term gives:

$$n_t^a = \frac{1}{N_t^a}\sum_{s=1}^{t-1}\hat{Y}_s^a, \tag{108}$$

so we substitute this into the second term to obtain:

$$\frac{d\alpha_{N_t^a}}{dN_t^a} = \frac{\alpha_0}{N_t^a+1}\cdot(1-\kappa)\cdot\left(-\frac{1}{(N_t^a)^2}\sum_{s=1}^{t-1}\hat{Y}_s^a\right) - \frac{\alpha_0}{(N_t^a+1)^2}\cdot\left(\kappa g + (1-\kappa)\cdot\frac{1}{N_t^a}\sum_{s=1}^{t-1}\hat{Y}_s^a\right). \tag{109}$$

We can further expand both terms. The first term becomes:

$$\frac{\alpha_0}{N_t^a+1}\cdot(1-\kappa)\cdot\left(-\frac{1}{(N_t^a)^2}\sum_{s=1}^{t-1}\hat{Y}_s^a\right) = -\frac{\alpha_0(1-\kappa)}{(N_t^a+1)\cdot(N_t^a)^2}\sum_{s=1}^{t-1}\hat{Y}_s^a. \tag{110}$$

The second term expands as:

$$-\frac{\alpha_0}{(N_t^a+1)^2}\cdot\left(\kappa g + (1-\kappa)\cdot\frac{1}{N_t^a}\sum_{s=1}^{t-1}\hat{Y}_s^a\right). \tag{111}$$

This can be split into two parts:

$$-\frac{\alpha_0\kappa g}{(N_t^a+1)^2} - \frac{\alpha_0(1-\kappa)}{(N_t^a+1)^2\cdot N_t^a}\sum_{s=1}^{t-1}\hat{Y}_s^a. \tag{112}$$

Finally, the complete expanded expression for $\frac{d\alpha_{N_t^a}}{dN_t^a}$ is:

$$\frac{d\alpha_{N_t^a}}{dN_t^a} = -\frac{\alpha_0(1-\kappa)}{(N_t^a+1)\cdot(N_t^a)^2}\sum_{s=1}^{t-1}\hat{Y}_s^a - \frac{\alpha_0\kappa g}{(N_t^a+1)^2} - \frac{\alpha_0(1-\kappa)}{(N_t^a+1)^2\cdot N_t^a}\sum_{s=1}^{t-1}\hat{Y}_s^a. \tag{113}$$

This result shows how the exploration parameter $\alpha_{N_t^a}$ decreases as $N_t^a$ increases, driven by both local attention ($n_t^a$) and global attention ($g$). The terms decay quadratically with $N_t^a$, highlighting the shift from exploration to more focused exploitation as more observational data is gathered and the rewards from each arm become better understood.

## B  ADDITIONAL RESULTS

In this Section, more quantitative results are provided.

**Analysis of models with different parameters.**  The experimental results, shown in Figures 3 and 4 and summarized in Table 3, highlight the performance of various MAB algorithms across different parameter settings. In this section, we set $\kappa = 0.5$, $\theta_{\min} = 1$, and $\theta_{\max} = 5$. The maximum value of $k$ for $k$-NN KL-UCB and $k$-NN UCB is considered to be 5 to ensure a fair comparison among the models. The BetaThompson model, which was tested with six combinations of $(\alpha, \beta)$ parameters, achieved its best performance with parameters $(4, 4)$, resulting in a mean reward of 0.22 and a cumulative reward of 176. Similarly, the Epsilon Greedy algorithm, evaluated with six different $\epsilon$ values, achieved the highest mean reward of 0.26 and a cumulative reward of 208 at $\epsilon = 0.2$. KL-UCB, another prominent algorithm, demonstrated its best performance at $c = 0.1$, with a mean reward of 0.25 and a cumulative reward of 200. $k$-NN KL-UCB and $k$-NN UCB, incorporating $k$-Nearest Neighbors, showed optimal results at $c = 5$ and $\rho = 10$, respectively, with mean rewards of 0.76 and 0.34. Notably, LinThompson and LinUCB algorithms, which leverage linear estimations, achieved mean rewards of 0.42 and 0.73, with cumulative rewards of 336 and 584. The UCB algorithm, when tested with six different $\rho$ values, performed best at $\rho = 10$, resulting in a mean reward of 0.14 and a cumulative reward of 112.

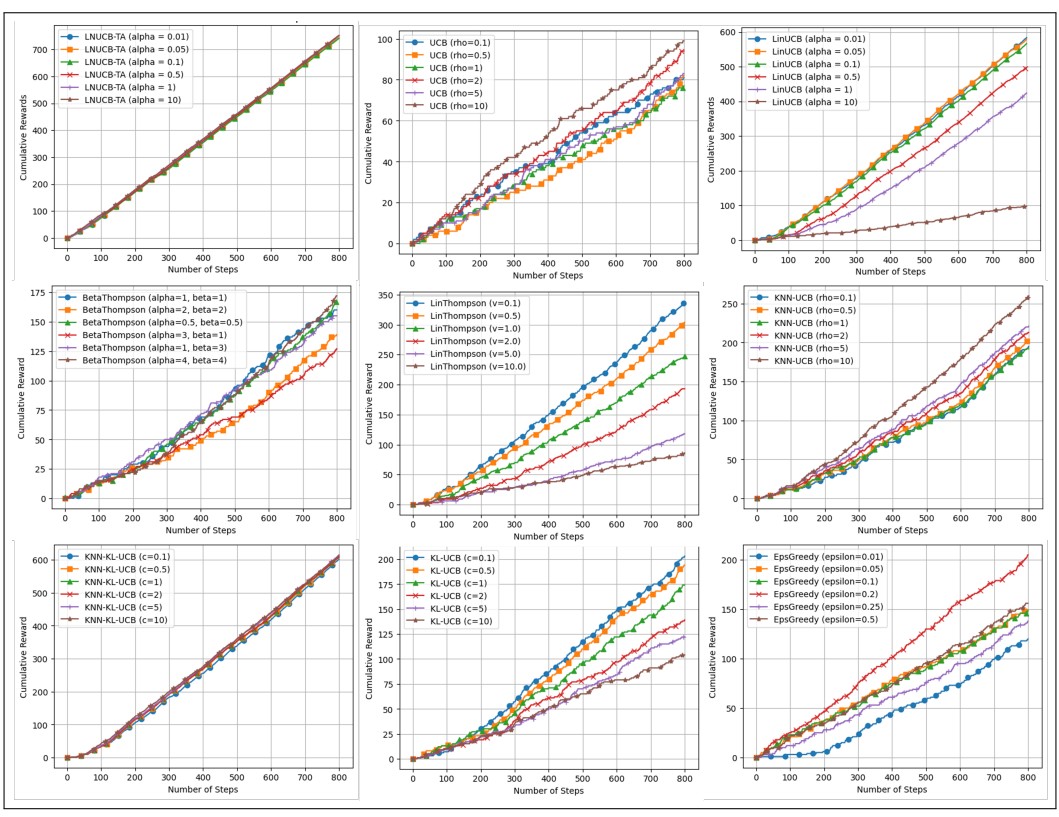

Figure 3: Performance comparison of models based on cumulative reward across six distinct parameter settings. The LNUCB-TA model demonstrates superior performance and more stable results compared to other models.

As indicated in Table 3, our novel LNUCB-TA model significantly outperformed all the aforementioned algorithms, achieving a mean reward of 0.94 and a cumulative reward of 753. The improvement by LNUCB-TA over other models is substantial, with the highest relative improvement observed over UCB (572%), followed by BetaThompson (327%), Epsilon Greedy (262%), KL-UCB (276%), $k$-NN UCB (176%), LinThompson (124%), LinUCB (28%), and $k$-NN KL-UCB (23%). The signif-

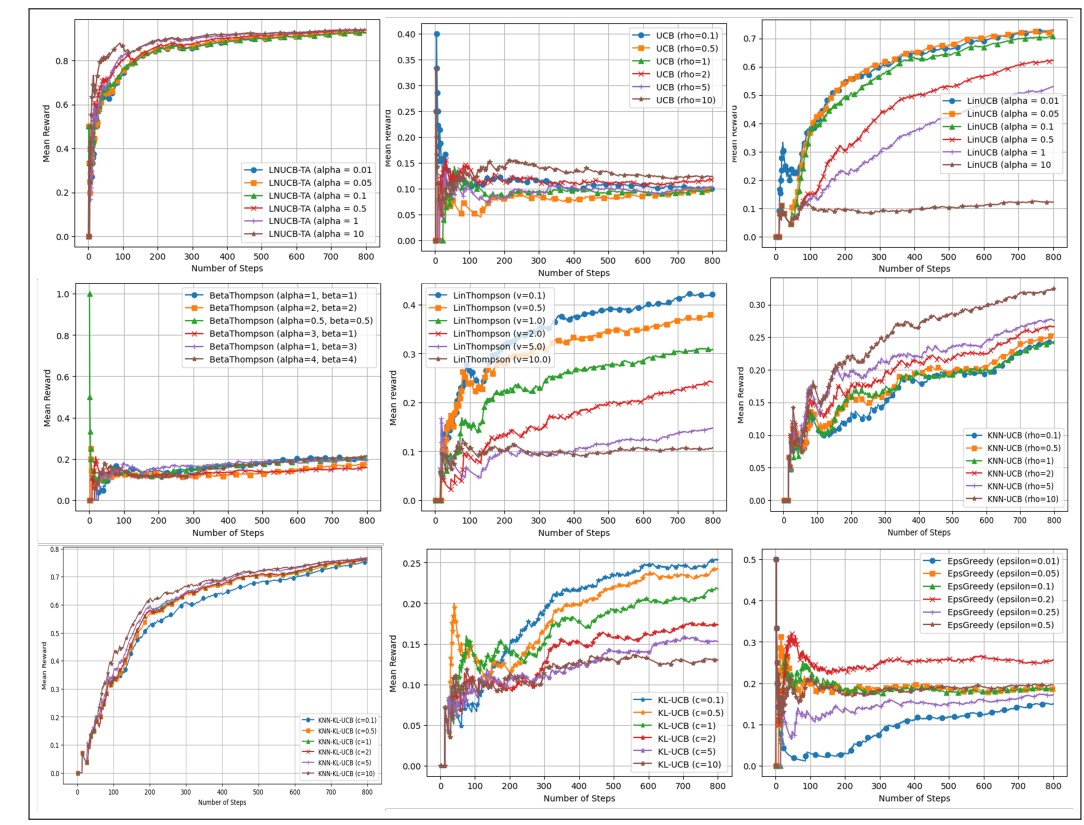

Figure 4: Comparison of model performance based on mean reward across six distinct parameter settings. The LNUCB-TA model achieves the highest mean rewards and exhibits stable performance.

icant enhancement and consistent performance underscore the robustness and effectiveness of the LNUCB-TA model, particularly its integration of linear and nonlinear estimations, adaptive $k$-Nearest Neighbors, and an attention-based exploration mechanism.

**Improvement over other models.** Figure 5 illustrates the performance enhancements achieved by integrating the $k$-NN adaptive strategy and an attention mechanism inspired by (Vaswani et al., 2017) (for each arm $a$ at time $t$, the exploration rate is weighted by an attention score as

$$\text{attention-score} = \frac{\exp(-\gamma \cdot N_t^a)}{\sum(\exp(-\gamma \cdot N_t^a))}, \quad (114)$$

where, $\gamma$ is a scaling parameter) into three traditional models namely BetaThompson, Epsilon Greedy, and LinThompson. Each enhanced model demonstrates a marked improvement in both cumulative and mean rewards over 800 steps. Specifically, the BetaThompson-enhanced model, with the best parameter combination $(\alpha, \beta) = (0.5, 0.5)$, achieves a mean reward of 0.79 and a cumulative reward of 632. Similarly, the Epsilon Greedy-enhanced model, optimized with $\epsilon = 0.25$, reaches a mean reward of 0.58 and a cumulative reward of 464. The LinThompson-enhanced model, with $v = 2$, shows a significant increase in performance, attaining a mean reward of 0.69 and a cumulative reward of 552.

Table 4 summarizes these results highlights the substantial improvements over their respective base models. The BetaThompson-enhanced model shows a 259.09% improvement over the base model, the Epsilon Greedy-enhanced model shows a 123.08% improvement, and the LinThompson-enhanced model demonstrates a 64.29% enhancement. Despite these significant gains, the comparison to the LNUCB-TA model reveals that while these enhancements are substantial, they still fall short of the performance of LNUCB-TA, which achieves a mean reward of 0.94. Specifically, the BetaThompson-enhanced model performs 16.08% worse than LNUCB-TA, Epsilon Greedy-enhanced 38.38% worse,

Table 3: Comparison of model parameters and performance: The table summarizes the various models (Model), the parameters tested (Param.), their values (Vals.), and the best-performing parameters (Best Param.). It also includes the best mean reward (BMR) and best cumulative reward (BCR) achieved by each model, as well as the percentage improvement of our model LNUCB-TA compared to others (Imp. by LNUCB-TA).

| Model | Param. | Vals. | Best Param. | BMR | BCR | Imp. by LNUCB-TA (%) |
|---|---|---|---|---|---|---|
| BetaThompson | $(\alpha, \beta)$ | (1, 1), (2, 2), (0.5, 0.5), (3, 1), (1, 3), (4, 4) | (4, 4) | 0.22 | 176 | 327.27 |
| Epsilon Greedy | $\epsilon$ | 0.01, 0.05, 0.1, 0.2, 0.25, 0.5 | 0.2 | 0.26 | 208 | 262.98 |
| KL-UCB | $c$ | 0.1, 0.5, 1, 2, 5, 10 | 0.1 | 0.25 | 200 | 276.50 |
| $k$-NN KL-UCB | $c$ | 0.1, 0.5, 1, 2, 5, 10 | 5 | 0.76 | 608 | 23.87 |
| $k$-NN UCB | $\rho$ | 0.1, 0.5, 1, 2, 5, 10 | 10 | 0.34 | 272 | 176.47 |
| LinThompson | $v$ | 0.1, 0.5, 1, 2, 5, 10 | 0.1 | 0.42 | 336 | 124.11 |
| LinUCB | $\alpha$ | 0.01, 0.05, 0.1, 0.5, 1, 10 | 0.01 | 0.73 | 584 | 28.91 |
| UCB | $\rho$ | 0.1, 0.5, 1, 2, 5, 10 | 10 | 0.14 | 112 | 572.32 |
| LNUCB-TA | $\alpha$ | 0.01, 0.05, 0.1, 0.5, 1, 10 | 1 | 0.94 | 753 | N/A |

and LinThompson-enhanced 26.69% worse. The superior performance of LNUCB-TA is attributed to its unique combination of both linear and nonlinear estimations. The results highlight the impact of the two key novelties—adaptive $k$-NN and attention mechanisms—setting a new framework for MAB algorithms through these innovative enhancements.

Table 4: Performance Comparison of Enhanced Models: The table presents the best parameters (Best Param.), best mean reward (BMR), and best cumulative reward (BCR), the improvement percentage over the base model, and the comparison percentage to LNUCB-TA (Comp. to LNUCB-TA (%)).

| Model | Best Param. | BMR | BCR | Imp. Over Base Model (%) | Comp. to LNUCB-TA (%) |
|---|---|---|---|---|---|
| BetaThompson-enhanced | (0.5, 0.5) | 0.79 | 632 | 259.09 | -16.08 |
| Epsilon Greedy-enhanced | 0.25 | 0.58 | 464 | 123.08 | -38.38 |
| LinThompson-enhanced | 2 | 0.69 | 552 | 64.29 | -26.69 |

**Error bars.** Based on the error bar plot in Figure 6, we can observe that the LNUCB-TA model demonstrates remarkable consistency in its performance across a variety of parameter settings. The plot shows the mean reward for different combinations of $\theta_{\min}$ and $\theta_{\max}$, and different values of $\kappa$, which is the weight of the global overall reward. Despite the changes in these parameters, the mean reward remains relatively stable, indicating that the model's performance is not heavily reliant on specific parameter choices. This consistency underscores the robustness of the LNUCB-TA model, making it a reliable choice for complex decision-making tasks where parameter tuning can be challenging.

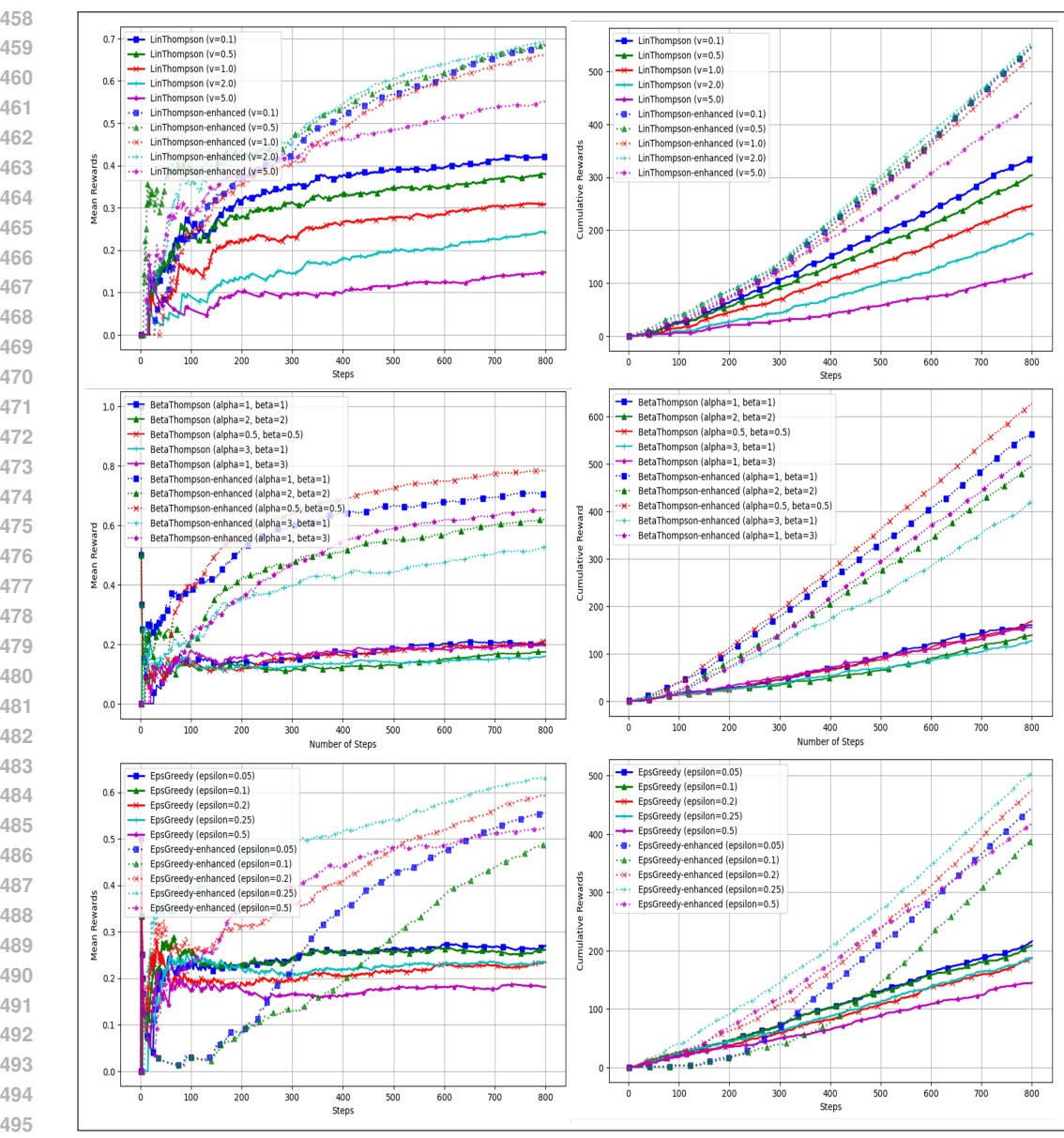

Figure 5: Performance enhancements achieved by integrating the $k$-NN adaptive strategy in Algorithm 2 and the attention mechanism in equation (114) into traditional models BetaThompson, Epsilon Greedy, and LinThompson.

**Additional datasets.** We extend our analysis of the LNUCB-TA model to additional real-world datasets to further validate its efficacy across diverse settings. One such dataset involves the AstroPh co-authorship network, initially observed at 5% (Madhawa & Murata, 2019a). Here, we focus on the cumulative reward comparison of our model against other state-of-the-art algorithms, demonstrating its capability in effectively expanding network visibility within a fixed query budget. Another dataset explored is an article matching dataset (Li et al., 2010; 2011), where the LNUCB-TA's performance is assessed in the context of matching relevant articles based on user preferences and interactions. These expanded evaluations provide a broader perspective on the model's versatility and its applicability to complex, real-world problems such as network exploration and content recommendation.

In Figure 7, the LNUCB-TA model, marked by the bold red line, outperforms other models with its superior performance as the evaluation progresses. This highlights the model's efficiency in adapting and optimizing its strategy over time, solidifying its effectiveness in dynamic settings. Additionally,

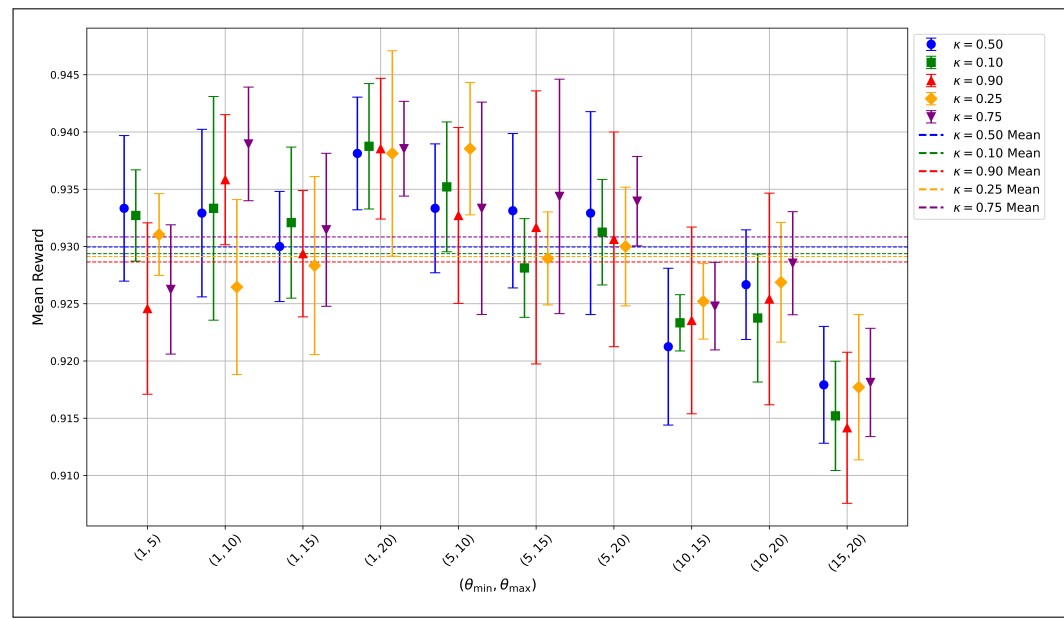

Figure 6: Performance stability of LNUCB-TA across various parameter settings: The plot illustrates the mean reward ranges for different combinations of $\theta_{\min}$ and $\theta_{\max}$, and different values of $\kappa$. Despite variations in these parameters, the model consistently maintains high performance, underscoring its robustness and the effectiveness of integrating adaptive $k$-NN and attention mechanisms.

our innovative approach that integrates $k$-NN with an attention mechanism into the $\epsilon$-Greedy strategy is represented by the bold green line. This combination shows significant improvements over the traditional KNN-$\epsilon$-Greedy model, underscoring the effectiveness of our proposed modifications in handling the exploration-exploitation balance more dynamically and efficiently.

Figure 8 presents the difference runtime between our proposed model against the vanilla combination of LinUCB and $k$-NN UCB model. The LNUCB-TA model, represented by the bold red line, consistently exhibits the lowest runtime, particularly as the maximum number of neighbors increases, underscoring its computational efficiency compared to the Lin+$k$-NN-UCB model (blue line) and other setups denoted by the dotted lines for varying NSteps. This demonstrates the LNUCB-TA model's capability to maintain lower computational costs even as the complexity of the task increases.

Additionally, the results presented in Table 5 shows that the LNUCB-TA model consistently outperforms purely linear models, purely nonlinear models, and the vanilla combination of linear and nonlinear approaches in terms of cumulative rewards across various exploration rates and operational steps. For instance, at an exploration rate of 0.1 and 7500 steps, it achieves the highest cumulative reward of 7261. The model is also substantially more efficient than the straightforward combination model (Lin+$k$-NN)-UCB, which takes 3381.71 seconds for a lower reward score, compared to the LNUCB-TA's 102.00 seconds.

## C  LIMITATION AND FUTURE DIRECTION

**Limitation.**  One limitation of our approach is the assumption of equal weights for the linear and nonlinear components in the model. While this simplifies the model, it may not fully capture the complexities of the underlying data. Future work could explore assigning different weights to these components, potentially enhancing performance by better capturing the data's structure. Additionally, the weights could be dynamically adjusted for each arm at each time step using attention mechanisms, further improving adaptability.

Also, our current implementation of the GALA mechanism employs a fixed weight ($\kappa$) to balance global and local attention in adjusting the exploration factor. While we have tried different fixed

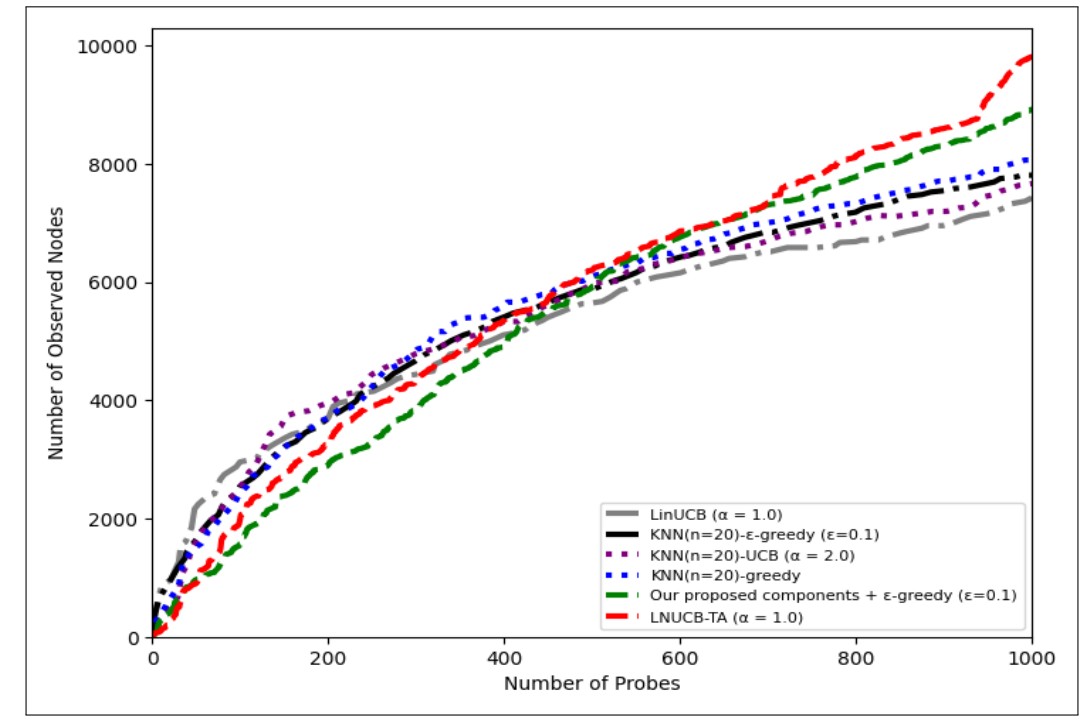

Figure 7: Cumulative reward (y-axis) comparison of models on AstroPh co-authorship network initially observed at (5%). Our **LNUCB-TA** model, represented by the **red** line, outperforms other models. Also, the **green** line, representing our novel $k$-NN approach with attention combined with $\epsilon$-Greedy, surpasses KNN-$\epsilon$-Greedy, showing the superiority of our proposed $k$-NN over existing $k$-NN bandit settings.

Table 5: Comparison of models on the article matching dataset, using a maximum of 5 neighbors based on cumulative reward (CR). We observe varying performance between the purely linear, purely nonlinear, and the vanilla combination model with neither of them demonstrating absolute dominance. However, the **LNUCB-TA** model consistently outperforms all three of them.

| $\alpha/\rho$ | Steps | LinUCB (CR) | LinUCB Run-time | $k$-NN UCB (CR) | $k$-NN UCB Run-time | (Lin+$k$-NN)-UCB (CR) | (Lin+$k$-NN)-UCB Run-time | **LNUCB-TA** (CR) | LNUCB-TA Run-time |
|---|---|---|---|---|---|---|---|---|---|
| 0.1 | 2500 | 2089 | 10.11 | 1618 | 14.79 | 2126 | 287.85 | **2262** | 26.21 |
| 0.1 | 5000 | 4570 | 12.36 | 3763 | 35.80 | 4604 | 1333.2 | **4762** | 92.01 |
| 0.1 | 7500 | 7063 | 19.79 | 6004 | 62.92 | 7099 | 3381.71 | **7261** | 102.00 |
| 1 | 2500 | 1349 | 5.80 | 1607 | 15.98 | 1401 | 295.59 | **1997** | 24.08 |
| 1 | 5000 | 3720 | 12.30 | 3739 | 36.17 | 3785 | 1331.09 | **4497** | 58.43 |
| 1 | 7500 | 6149 | 19.02 | 5996 | 62.03 | 6186 | 3226.52 | **6996** | 98.57 |
| 10 | 2500 | 410 | 6.53 | 1595 | 15.84 | 410 | 279.34 | **1601** | 21.65 |
| 10 | 5000 | 1197 | 13.37 | 3721 | 36.57 | 1311 | 1169.57 | **4019** | 55.50 |
| 10 | 7500 | 2282 | 18.07 | 5966 | 61.90 | 2536 | 3223.6 | **6519** | 95.48 |

values for $\kappa$, these weights might be not optimized. Determining the optimal value for $\kappa$ could further enhance the model's performance.

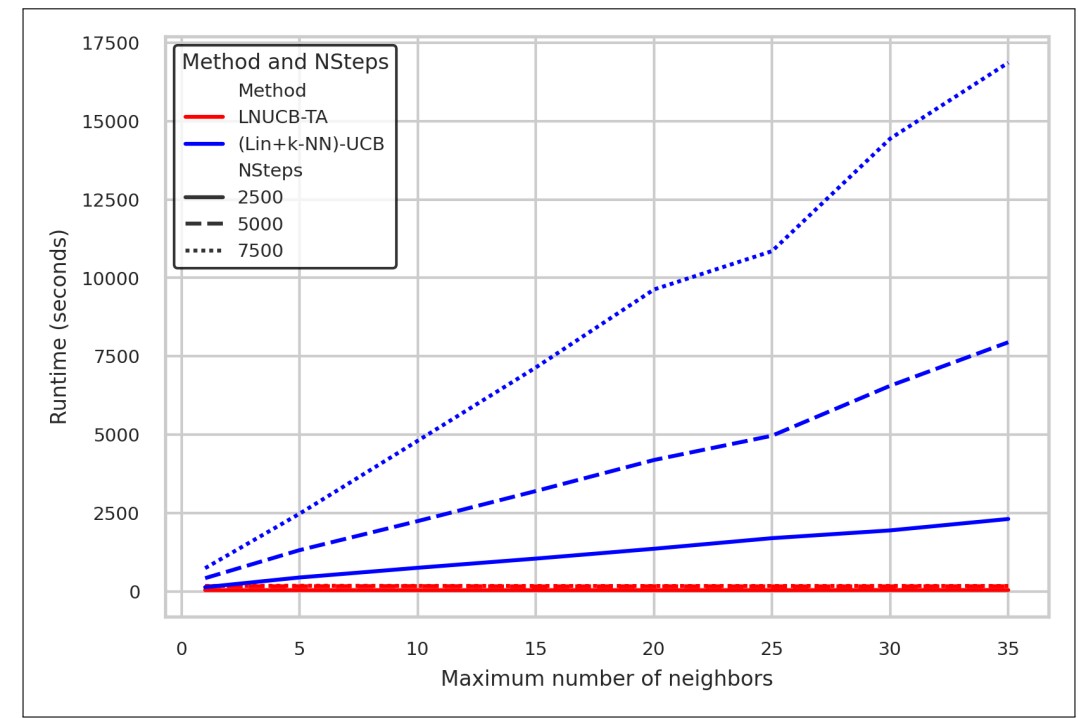

Figure 8: Runtime and scalability comparison of our model against the straightforward combination model on the article matching dataset. The **LNUCB-TA** model is more scalable, maintaining quite consistent processing times, even as $\max_k$ and the number of steps increase.

**Attention mechanisms in MAB frameworks.** The introduction of attention mechanisms in the MAB framework opens new avenues for enhancing decision-making processes in various domains. While our work applied attention to the exploration rate, there are numerous other areas within the MAB framework where attention mechanisms can be beneficial. For instance, attention could be used to dynamically prioritize contexts based on their significance or complexity, thereby improving overall efficiency and effectiveness. Additionally, attention mechanisms could be applied to weight the influence of historical rewards differently over time, allowing for more nuanced learning from past experiences. Another potential application could be the use of attention to identify and focus on emerging trends or shifts in the data, ensuring that the model adapts swiftly to new patterns

**Impact on industrial settings.** LNUCB-TA, as it dynamically adjusts its exploration rate, can be beneficial in various areas where initial parameters need to be optimized, such as in recommendation systems (Zhou et al., 2017; Bouneffouf et al., 2012; 2014) where initial user preferences are unknown, in finance (Shen et al., 2015; Huo & Fu, 2017) for portfolio optimization where initial risk preferences must be set, and in healthcare (Bastani & Bayati, 2020; Durand et al., 2018) for personalized treatment plans where patient-specific parameters need to be optimized. Our model can also be applied to areas not yet extensively covered by MAB approaches (Bouneffouf & Rish, 2019), such as manufacturing. In this context, each arm represents a different material or material property configuration, while the context includes features describing the manufacturing conditions and requirements. The reward corresponds to the performance or suitability of the material under these conditions. By leveraging both linear and nonlinear estimations along with attention-based mechanisms, LNUCB-TA can effectively balance exploration and exploitation, identifying optimal material properties under varying conditions. This ability to dynamically adapt and refine decisions based on historical data and contextual insights makes our model particularly well-suited for such applications.

**A new paradigm for MAB algorithms.** Moreover, the incorporation of adaptive $k$-NN discussed in Algorithm 2, and attention mechanisms discussed in Algorithm 3 and equation (114) not only

enhances the performance of LNUCB-TA but also improves the performance of other models. This sets a new framework for MAB algorithms by integrating these advanced modifications.

**Technical extensions in other areas.** The inspiration from how we used attention mechanisms to make our model independent of initial parameter choices can be applied in various technical fields. This approach can enhance meta-heuristic algorithms for combinatorial optimization problems (Agushaka & Ezugwu, 2022; Shadkam, 2022), evolutionary algorithms where initial population parameters must be set (Lobo et al., 2007; Qin, 2023), and machine and federated learning models where hyperparameters need to be tuned (Koskela & Kulkarni, 2024; Khodak et al., 2021; Turner et al., 2021). By reducing the dependency on the initial parameter settings, this concept can improve the robustness and efficiency of these techniques, ensuring consistent performance irrespective of the chosen initial parameters.

## D  IMPLEMENTATION GUIDELINE.

For implementing the LNUCB-TA algorithm and other models, the chosen parameters, detailed in Table 3 and illustrated in Figure 6, were selected based on a comprehensive review of the literature to cover a wide range for thorough analysis. The implementation was conducted using Google Colab, which provides an accessible and efficient environment for running Python code. The essential libraries required for this implementation include NumPy (version 1.25.2) for scientific computing, pandas (version 2.0.3) for data manipulation and analysis, Matplotlib for visualizations, scikit-learn for machine learning tasks (including KNeighborsRegressor), tqdm for progress bars, Requests (version 2.31.0) for handling HTTP requests, and the ABC module for defining abstract base classes.

