# OpenReview forum: "LNUCB-TA: Linear-nonlinear Hybrid Bandit Learning with Temporal Attention"
_ICLR.cc/2025/Conference — ICLR 2025 Conference Withdrawn Submission_

### Official Review · Reviewer_1LHD · 2024-11-01

**Soundness:** 2
**Presentation:** 2
**Contribution:** 2
**Rating:** 3
**Confidence:** 3

**Summary:**

This paper studies a type of contextual bandit where the reward model is a mixture of linear and non-linear components. The non-linearity is captured by k-NN part, k-NN denotes the Nearest Neighbor. More formally, upon sampling an arm $a$ the observed reward is  $\hat{Y}\_t^a=o\_t^a\left(x_t, z_t\right)+\xi_t^a$, where $o\_t^a\left(x_t, z_t\right) = \mu\_t^a \cdot x_t^a+\mathrm{kNN}_{k, t}^a\left(x_t^a, z_t^a\right)$ and $z_t$ can be thought of as a nearest neighbor to $x_t$ which yields a similar reward in expectation to $x_t$. A motivating example that the paper gives is the traditional online recommendation system where the above linear component captures broad trends, such as higher click-through rates, while the adaptive k-NN component refines this by recognizing local patterns. They claim that the previous contextual bandit approaches cannot handle the mixture model (with the adaptive k-NN component) and so they propose a LNUCB-TA algorithm. The algorithm is mainly a LinUCB/UCB type algorithm where the confidence interval $U C B\_t^a=\left(\alpha\_{N_t^a}\right) \cdot \sqrt{\left(x_t^a\right)^{\top}\left(\Sigma\_t^a\right)^{-1} x\_t^a}$ has an attention component $\alpha\_{N_t^a}$. The $\alpha\_{N_t^a} $ basically interpolates between the global reward average $g$ and number of times each arm is sampled (I will discuss this again). They provide a regret bound in Theorem 1, and show that LNUCB-TA regret scales as $R\_T=\tilde{\mathcal{O}}(\sqrt{d T})$. Finally, they empirically validate their result against several benchmarks in real datasets.

**Strengths:**

1. This paper studies a new type of contextual bandit with mixture models. To my knowledge, this is somewhat novel.
2. The proposed UCB-type algorithm seems a valid approach. However, I have some concerns with the constants involved (to be discussed in the questions).
3. They theoretically analyze the algorithm and show it achieves sub-linear regret.

**Weaknesses:**

1. The paper needs to improve its writing significantly. The assumptions are mentioned in the Appendix. Please state them in the main paper, and a discussion on why they are important, The definition of BALL is mentioned in the appendix. I could not find the definition of the $\beta_t$ as well. Please point to it. These things will make the paper more readable in the next version.
2. The motivation is not clear to me. This requires a more detailed explanation. See my question 1.
3. The attention of the UCB requires more explanation. It is not clear to me how tight it is. Also how it reflects on the regret bound. See my question 2.
4. There are many questions on the theoretical proof of Theorem 1 as well as the technical novelty. I have some doubts regarding the approach. See my question 3.
5. How some of the linear contextual baselines are implemented is not clear from the draft.

**Questions:**

1. Why an adaptive kNN component is needed for linear contextual settings? The paper starts with a statement (line 75-76) "Despite advancements in MAB algorithms, existing algorithms predominantly fail to incorporate adaptive strategies for reward estimation as a function of the context." I am not sure I fully follow this. There are papers on kernel-UCB which uses reward estimation as a function of context "Finite-Time Analysis of Kernelised Contextual Bandits, Valko et al, 2013" or Neural UCB papers, or Collaborative Neural UCB papers that extend this idea to non-linear contextual settings, What is k-NN specifically bringing to the table when such models exist?
2.  The attention is basically defined as $\alpha_{N_t^a}= \frac{\alpha\_0}{\left(N_t^a+1\right)} \cdot\left(\kappa g+(1-\kappa) n\_t^a\right)$. Consider $\kappa = 1/2$, $\alpha\_0 = 2$, assume the average of global rewards $g \approx n_t^a$, then $\alpha_{N_t^a} \propto 1/\left(N_t^a+1\right)$. This is a very fast decay of UCB. So it is not clear to me how this is actually helping the exploration. Also not clear to me how you set $\kappa$.
3. In Theorem 1, it is not clear to me how the $\alpha\_{N\_t^a}$ is showing up in the proof. I suggest you write a more in-depth proof overview as the current proof overview from lines 376-386 is insufficient. How does the $\kappa$ not appear in the proof? In your Corollary 1, you state that $\Sigma\_t^a=$ $\left(X\_t^a\right)^T X\_t^a+\lambda I$ is the covariance matrix (Lattimore \& Szepesvári, 2020, equation 20.1) updated for arm $a$. However, in the book, the co-variance matrix is defined over all arms selected till time $t$. This is crucial as the co-variance matrix captures the information gathered from all arms, and is crucial to drive the informative sampling.
4. Assumption 3  in Appendix lines 873 seems like a very strong assumption. It states for all time steps t and for each arm, a,  true parameter vector $\mu^*$ resides within a confidence ball centered around the estimated parameter $\mu_t^a$. This confidence ball is denoted as $\mathrm{BALL}_{(t, a)}$. This should be rigorously proved instead of taking an assumption. Can you clarify this?
5. It would be also great if the authors explain in detail how the benchmarks LinUCB, LinTS are implemented for this setting. I also feel that more baslines like Kernel-UCB or Neural UCB should be used as a baseline to show that they fail in this setting.

---

> ### Author Response · Authors · 2024-11-19
> **Response to Reviewer 1LHD (Part 1/5)**
>
> We sincerely thank the reviewer for their valuable feedback and insightful comments. Please kindly find our response to address your Questions.
>
> **Q1 (Adaptive kNN):**
>
> ### Limitations of Purely Linear Models in Contextual Bandits
> In a standard linear contextual bandit model, the expected reward given a context $X_t$ is modeled as:
>
> $$
> E[Y_t \mid X_t] = \mu_t^\top X_t
> $$
>
> where:
> - $\mu_t$ is the parameter vector for time $t$,
> - $X_t$ is the context vector at time $t$.
>
> This linear structure assumes that the relationship between context $X_t$ and expected reward is linear. However, if the reward function $Y_t$ exhibits non-linear dependencies within certain regions of the context space, this model will not capture these local variations accurately (lines 68-69).
>
> ### Non-Linearity with Fixed Kernels: Kernel-UCB as an Example
> Kernelized methods, such as Kernel-UCB, attempt to model non-linear reward structures by mapping the context into a Reproducing Kernel Hilbert Space (RKHS) with a kernel function $k(\cdot, \cdot)$, yielding:
>
> $$
> E[Y_t \mid X_t] = \sum_{i=1}^{t-1} \alpha_i k(X_t, X_i)
> $$
>
> where $\alpha_i$ are weights learned over previous contexts $X_i$ up to time $t-1$.
>
> In non-stationary environments, where reward relationships may shift over time due to context-specific factors, the fixed kernel function $k(\cdot, \cdot)$ limits the model's flexibility in adapting to rapid changes in reward patterns.
>
>
> ### Structure of Neural UCB: Fixed Non-Linearity
> In NeuralUCB, the reward function is modeled as a combination of a neural network's predictive output and an uncertainty term. Formally, the expected reward given a context $x_{(t,a)}$ is expressed as:
>
> $$
> E[Y_t \mid x_{(t,a)}] = f_\theta(x_{(t,a)}) + \gamma_t \sqrt{g(x_{(t,a)}; \theta_{(t-1)})^\top Z_{(t-1)}^{-1} g(x_{(t,a)}; \theta_{(t-1)})}
> $$
>
> where:
> - $f_\theta(x_{(t,a)})$: The neural network output parameterized by weights $\theta_{(t-1)}$, which models the nonlinear reward as a function of the context $x_{(t,a)}$.
> - $g(x_{(t,a)}; \theta_{(t-1)})$: The gradient of the neural network's output with respect to its weights, capturing the influence of the context on the learned function.
> - $Z_{(t-1)}$: The regularized covariance matrix incorporating feature embeddings up to round $t-1$.
>
> In spite of the fact that Neural UCB is flexible because the neural network approximates complex, nonlinear reward functions, it has several limitations:
> - **Fixed Network Structure:** During training, the neural network architecture (e.g., number of layers, neurons) is fixed, resulting in predefined complexity that cannot be adjusted to new patterns or reward structures as they arise.
> - **Adaptivity Constraints:** Updating a neural network to reflect new reward patterns is computationally expensive and may result in slow adaptation, especially in non-stationary environments.
>
> ### Our Approach: Dynamic k-NN Mechanism
> We distinguish our approach from kernelized and neural approaches by selecting relevant historical rewards dynamically without reliance on fixed mappings or pre-trained network structures through the adaptive k-NN component. Unlike Kernel-UCB or Neural-UCB, our approach allows the reward model to adjust its neighborhood size and content dynamically based on the reward variance. As the model incorporates this flexibility for **each arm** and **at every time step**, it is particularly suited to non-stationary environments without the computational overhead associated with neural networks.
>
> This dynamic mechanism can be intuitively simplified as:
>
> $$
> E[Y_t \mid X_t] \approx \mu_t^\top X_t + \frac{1}{k} \sum_{i \in N(X_t)} Y_i
> $$
>
> - $\mu_t^\top X_t$: Captures broad patterns (global trend).
> - $\frac{1}{k} \sum_{i \in N(X_t)} Y_i$: Captures localized variations (local adjustment).
> - The global trend term identifies broad, overall patterns in the reward structure.
> - The k-NN term dynamically fine-tunes this by incorporating immediate, localized variations in the context space, enabling the model to make quick and context-sensitive adjustments.
>
>
> For more in detail comparison of our model and the literature highlighting the contribution of our model, please kindly refer to our response to Reviewer 1 (AgPg), Parts 3 and 4.
>
> **Q2 (Specific Scenario):**
>
> In this scenario, the fast decay of $\alpha_{N_t^a}$ is actually a strength, since it prioritizes exploitation while maintaining adaptability in dynamic situations.
>
> ### Why Fast Decay is Beneficial under $g = n_t^a$
> - **Reduced Need for Exploration:**
>   If $g = n_t^a$, the rewards for all arms are close to each other, indicating a limited degree of variability in reward outcomes. In this case, the algorithm does not have to explore aggressively, as each arm's reward pattern is closely aligned with the overall average reward pattern. Therefore, a fast decay under this assumption is advantageous, since it reduces exploration and allows the algorithm to concentrate on exploiting known rewards more efficiently.

---

> ### Author Response · Authors · 2024-11-19
> **Response to Reviewer 1LHD (Part 2/5)**
>
> **Continue of Q2 (Specific Scenario):**
>
> - **Efficient Exploitation in a Stable Environment:**
>   When $\alpha_t^a$ decays quickly, the algorithm is able to exploit stable reward patterns effectively, avoiding exploration of arms with similar expected rewards. By converging more rapidly towards exploitation, this approach maximizes cumulative rewards, which is optimal when rewards are relatively consistent across arms.
>
> $\kappa$ adjusts the weight between global and local rewards in the attention mechanism:
> - **Lower $\kappa$ values** prioritize local attention, enhancing responsiveness to individual arm trends, which is useful in dynamic environments such as e-commerce where consumer preferences change rapidly.
> - **Higher $\kappa$ values** emphasize global attention, which is suitable for stable environments like financial markets where long-term trends prevail.
>
> Figure 6 shows that the model performs robustly across various $\kappa$ settings, confirming its adaptability to different strategic emphases without compromising overall performance in later steps. This flexibility allows for effective application in both rapidly changing and stable environments, ensuring optimal performance tailored to the specific context.
>
> ### Intuition
> As stated in lines 326-331, the $\alpha_{N_t^a}$ adjusts the exploration rate through an attention mechanism that considers both temporal and spatial variations in data. This dual consideration allows for dynamic adjustment of exploration efforts based on time-dependent changes (**temporal**) and distinct reward patterns across different arms (**spatial**). By integrating these aspects, the mechanism enhances the model's adaptability to real-time changes, ensuring more effective exploration and exploitation.
>
> This approach is particularly novel as it moves beyond static or merely context-aware adjustments seen in other models (lines 85-97). The robustness of this approach is confirmed in our empirical results (**Figure 2**, and **Figure 6**) and supported by **Theorem 2**, illustrating that our model maintains consistent performance across a range of $\kappa$ and $\alpha_0$ values, which marks a substantial advance over current models.
>
> **Q4 (Assumption 3):**
>
> **Confidence Ball: Assumption vs. Proposition**
>
> ### Assumption 3 (Confidence in Parameter Estimation)
>
> Assumption 3 states that, for all time steps $t \in [T]$ and arms $a \in [A]$, the true parameter vector $(\mu_a )^*$ lies within a confidence ball $\text{BALL}_t^a$ centered around the estimated parameter $\hat{\mu}_t^a$. This assumption is a prerequisite for deriving high-probability confidence bounds and is defined in Definition 1 in Equation (17) of the paper. In which, $\Sigma_t^a$ is the arm-specific covariance matrix and $\beta_t^a$ is the confidence parameter that scales with the uncertainty in the measurements up to time $t$. The confidence ball incorporates both linear and nonlinear ($k$-NN) adjustments, as shown in Assumption 1 and Corollary 4 of the paper.
>
> ### Proposition 1 (Uniform Confidence Bound)
>
> Proposition 1 demonstrates the confidence ball property, ensuring it holds with high probability across all time steps $t$ and arms $a$:
>
> $$
> \Pr \left( \forall t, (\mu^a )^* \in \text{BALL}_t^a \right) \geq 1 - \delta
> $$
>
> We have provided the proof for this proposition in the paper, and it relies on self-normalized martingale inequalities, ridge regression guarantees, and the union bound, ensuring that Assumption 3 is mathematically valid and serves as the basis for our regret analysis (Proposition 2).
>
> ### Implications for Exploration-Exploitation and Regret Bounds
>
> By this assumption and Proposition, the true parameter vector $(\mu^a )^*$ lies in $\text{BALL}_t^a$ with high probability. This result supports:
>
> 1. **Exploration Bonus (Lemma 2):**
>    The confidence ball defines the exploration bonus:
>    $$
>    \sqrt{\beta_t^a} \cdot \sqrt{(X_t^a)^\top (\Sigma_t^a)^{-1} X_t^a}.
>    $$
>
> 2. **Regret Bounds:**
>    Proposition 2 uses the confidence ball to bound regret:
>    $$
>    R_T \leq \sum_{t=1}^T \sum_{a=1}^A \Delta_t^a \left( l_t^a \left( (X_t^a )^* \right) + f_{k,t}^a \left( (X_t^a )^*, (Z_t^a )^* \right) - l_t^a (X_t^a ) - f_{k,t}^a (X_t^a, Z_t^a ) \right).
>    $$
>
> ### Role of Assumption 3 in Bandit Literature
>
> The use of confidence regions, such as $\text{BALL}_t^a$, is a standard practice in contextual bandit literature:
>
> 1. **Linear Bandits:**
>    In works like LinUCB, the assumption that the true parameter vector resides within a high-probability confidence region is central to the regret analysis, which allows exploration bonuses to be derived for UCB-based strategies and linked to regret bounds.
>
> 2. **Nonlinear Bandits with Localized Models:**
>    Our Assumption 3 (Confidence in Parameter Estimation) is a standard prerequisite in contextual bandit literature, including models like $k$-NN UCB. Specifically, $k$-NN UCB defines confidence intervals through the uncertainty value $U_{t,k}^a (x)$.

---

> ### Author Response · Authors · 2024-11-19
> **Response to Reviewer 1LHD (Part 3/5)**
>
> **Continue of Q4 (Assumption 3):**
>
> The uncertainty value implicitly bounds the true reward function $f_a (x)$ within:
>
> $$
> \hat{f}\_{t,k}^a (x) \pm U\_{t,k}^a (x)
> $$
>
>
>
>    providing a confidence region analogous to our explicit definition of $\text{BALL}_t^a$.
>
>    This approach ensures exploration-driven optimism, and both frameworks rely on high-probability bounds for the true parameters. Also, in Neural UCB, a similar assumption has been used (Lemma 5.1).
>
> 3. **Union Bound Over Time Steps:**
>    Many regret analyses use a union bound across time steps $t \in [1,T]$ to ensure that the confidence intervals hold uniformly, as seen in \citet{lattimore2020bandit}. The logarithmic terms in $\beta_{t,a}$ are adjusted accordingly to account for the increasing number of time steps.
>
> 4. **Connection to Self-Normalized Martingales:**
>    The self-normalized bound for the noise term $\eta_{t,a}$ is a standard tool in contextual bandit literature. This ensures that even in stochastic settings, the assumption remains valid for bounding cumulative regret.
>
>
> ### Summary
> In summary, Assumption 3 is actually not a strong assumption but a standard prerequisite in bandit literature, consistent with both linear (e.g., LinUCB) and non-linear models (e.g., k-NN UCB). Proposition 1 validates this assumption, ensuring it holds uniformly with high probability. Together, they provide a robust foundation for our theoretical framework and regret analysis.
>
>
> ---
> **Q3 (Exploration Rate, Arm Specific Scenario):**
>
> ### Proof Overview
> The proof overview is intended to provide a high-level understanding of the process, helping the reader understand the main ideas and flow without being overwhelmed by technical details. It provides a general outline of the methodology, leaving the rest of the section for the detailed proofs.
> Rather than repeating technical details that have already been addressed later in the paper, the objective of the overview is to provide an intuitive explanation of how the proof proceeds. Appendix A contains all proofs, including all propositions, lemmas, and supporting arguments. But, we will add more explanation and a table for comprehensive list of notations and their representations based on your comment.
>
> ### Role of $\alpha_{N_t^a}$
> Using this exploration parameter, the exploration-exploitation balance is dynamically adjusted by modulating confidence bounds by affecting the exploration bonus indicated in equation (7) of the manuscript. As a result of the exploration parameter in LNUCB-TA, it is possible to dynamically adjust the exploration-exploitation trade-off for each arm. The size of the confidence ball is affected by this adjustment, which indirectly affects the regret bounds.
> In contextual bandit proofs, it is standard practice for exploration parameters such as $\alpha$ (in LinUCB) or $\rho$ (in k-NN UCB) to indirectly affect theoretical guarantees. Instead of being explicitly expressed in regret bounds, their influence is encapsulated in terms such as confidence regions or covariance matrices.
> - In LinUCB, the $\alpha$ parameter adjusts the size of the uncertainty term within the confidence interval, but it does not appear explicitly within the regret bounds.
> - $\rho$ modulates uncertainty in k-NN UCBs based on neighborhood distances, although this contribution is similarly reflected in the confidence region.
>
> LNUCB-TA follows this standard. $\alpha_{N_t^a}$, which combines global and local reward signals dynamically, influences the theoretical results through its impact on the confidence parameter.
>
> The confidence region in LNUCB-TA is stated in Definition 1 in the paper, in which, $\alpha_{N_t^a}$ indirectly influences $\Sigma_t^a$ by determining the exploration bonus, which governs how frequently $x_t^a$ contributes to the update. Please kindly refer to Corollary 2, 3, and Lemma 5 for more details.
>
> Also, the weight $\kappa$ controls the relative importance of global and local attention in $\alpha_{N_t^a}$. While critical in determining the exploration parameter, it does not explicitly appear in the proof as $\kappa$ affects the magnitude of $\alpha_{N_t^a}$, which is reflected in $\Sigma_t^a$. This influence is absorbed into the confidence parameter $\beta_t^a$, as discussed above. Also, the proof focuses on bounding regret by ensuring that $(\mu^a)^* \in \text{BALL}\_t^a$
> for all $t$ with high probability. While $\kappa$ indirectly affects this containment through $\alpha_{N_t^a}$, the final expressions are simplified to focus on the confidence bounds, which is the standard practice in literature.
>
>
> ### Distinct Setting: Per-Arm Covariance Matrices in Our Model
> In standard contextual bandit models (e.g., Lattimore & Szepesvári, 2020), a global covariance matrix aggregates information across all arms:
>
> $$
> \Sigma_t = \lambda I + \sum_{i=1}^t x_i x_i^\top
> $$
>
> where $x_i$ represents the context vector of the selected arm at time $i$.
> Please see next part for the rest of response.

---

> ### Author Response · Authors · 2024-11-19
> **Response to Reviewer 1LHD (Part 4/5)**
>
> **Continue of Q3 (Exploration Rate, Arm Specific Scenario):**
>
> This global matrix is appropriate when all arms share a single global parameter vector $\mu$, and the reward for any arm $a$ is modeled as:
>
> $$
> r_t^a = \langle x_t^a, \mu \rangle + \eta_t^a,
> $$
>
> where $\mu$ is the shared parameter vector.
>
>
> ### Our Setting: Hybrid Reward Model and Arm-Specific Covariance Matrices
> In our hybrid contextual bandit model, each arm $a$ is associated with its own parameter vector $\mu^a$, and the observed reward for arm $a$ at time $t$ is modeled as:
>
> $$\hat{Y}_t^a = \mu_t^a \cdot x_t^a + \text{k-NN}\_{k,t}^a(x_t^a, z_t^a) + \xi_t^a$$
>
>
> as detailed in Equation (1) of the manuscript. Here:
> - $\mu_t^a \cdot x_t^a$ is the linear reward component for arm $a$,
> - $\text{k-NN}_{k,t}^a(x_t^a, z_t^a)$ is the non-linear adjustment using k-nearest neighbors,
> - $\xi_t^a$ is the noise term.
>
> This distinct parameterization necessitates per-arm covariance matrices (stated in Corollary 1), defined as:
>
> $$
> \Sigma_t^a = (X_t^a)^\top X_t^a + \lambda I,
> $$
>
> where:
> - $X_t^a = [x_{a,1}, x_{a,2}, \ldots, x_{a,N_t^a}]^\top$ contains all context vectors observed for arm $a$ up to time $t$,
> - $N_t^a$ is the number of times arm $a$ has been selected up to time $t$.
>
> The per-arm structure ensures that exploration and uncertainty for each arm are driven solely by its own historical data, as outlined in Equation (3) of the manuscript.
>
> Rather than relying on a global structure for exploration decisions, this formulation ensures that exploration decisions are driven by the most relevant context and reward history for each arm. This is one of the main aspects of our proposed model, followed in all the mathematical settings of the paper. Our hybrid model integrates global trends ($\mu_t^a \cdot x_t^a$) and local adjustments ($\text{k-NN}_{k,t}^a(x_t^a, z_t^a)$), necessitating a localized uncertainty measure. The per-arm covariance matrix naturally aligns with this requirement, enabling efficient and adaptive exploration in reward settings.
>
> **Q5 (Benchmarks):**
>
> ### As benchmarks, we selected LinUCB, LinThompson, k-NN UCB, and k-NN KL-UCB in order to highlight the limitations of models that rely exclusively on linear or nonlinear approaches and to motivate our unique synthesis of linear and nonlinear components.
>
>
> #### Why LinUCB and LinThompson?
>
> - **Purpose:** The purpose of these models is to capture global trends through linear modeling of context-reward relationships.
> - **Relevance to Our Setting:** By incorporating LinUCB and LinThompson as baselines, we demonstrate how linear models have difficulty adapting to non-linear variations in reward structures. In particular, these benchmarks demonstrate the limitations of linear-only approaches in capturing local patterns and context-sensitive adjustments, which are critical in dynamic environments.
>
> ### Implementation in Our Setting:
>
> - **LinUCB:**
>   The reward expectation is correctly formulated as:
>   $$
>   E[r_{t,a} \mid x_{t,a}] = \langle x_{t,a}, \mu_a^* \rangle,
>   $$
>   where $x_{t,a} \in \mathbb{R}^d$ is the context vector and $\mu_a^* \in \mathbb{R}^d$ is the true parameter vector.
>   **Exploration Mechanism:** The exploration-exploitation trade-off is achieved using the classical upper confidence bound (UCB).
>
> - **LinTS:**
>   The reward expectation for LinTS is also correctly expressed as:
> $$
> E[r_{t,a} \mid x_{t,a}] = \langle x_{t,a}, \mu_a^* \rangle.
> $$
>
>   The posterior sampling formulation is consistent with the standard Thompson Sampling framework:
>
> $$\tilde{\mu}\_{t,a} \sim \mathcal{N}(\hat{\mu}\_{t,a}, \Sigma\_{t,a}),$$
>
>
>   where $\tilde{\mu}\_{t,a}$ is a sampled parameter vector from a multivariate Gaussian distribution with mean $\hat{\mu}\_{t,a}$ and covariance matrix $\Sigma_{t,a}$.
>
> Both LinUCB and LinTS assume a purely linear relationship between the context and reward, relying on a linear model ($\langle x_{t,a}, \mu_a^* \rangle$) to predict rewards. While these methods perform well in environments where the reward structure follows global linear trends, they fail to capture local variations or nonlinear dependencies in the reward function.
>
> In contrast, our proposed model builds upon these frameworks by integrating a novel k-NN-based refinement to capture local variations and non-linear dependencies in the reward structure, adapting to context-sensitive patterns. Additionally, it incorporates an attention mechanism that dynamically adjusts the exploration parameter in real time, effectively balancing exploration and exploitation based on reward history without the need for pre-tuned parameters. This unique synthesis of a global linear component, local non-linear adjustment, and adaptive exploration ensures the model overcomes the limitations of purely linear methods while maintaining computational efficiency and adaptability in dynamic environments.
>
> Please see the next part for the rest of the response.

---

> ### Author Response · Authors · 2024-11-19
> **Response to Reviewer 1LHD (Part 5/5)**
>
> **Continue of Q5 (Benchmarks):**
>
> - **LinUCB and LinTS** used the linear contextual features (same as part of our model) but did not include the nonlinear k-NN adjustments.
>   This limitation led these models to capture only global trends while failing to adapt to local variations in the reward structure.
>
>
> #### Why k-NN UCB and k-NN KL-UCB?
>
> - **Purpose:** In these models, non-linear, local adjustments are made based on nearest neighbors without taking into account global reward trends.
> - **Relevance to Our Setting:** Our analysis of k-NN UCB and k-NN KL-UCB as baselines shows that while these models are effective in capturing localized reward dependencies, they have limited ability to accurately model global reward trends. Therefore, they are suboptimal in settings where both global and local trends are important.
>
>
> ### Our Unique Contribution:
> As stated in the Contribution section (lines 95-128), using the insights from these benchmarks, we propose a hybrid approach that incorporates both local (non-linear) and global (linear) factors. Our model captures the following:
> - **Through the linear component, Global Trends** ensures robustness to high-dimensional contexts and captures a wide range of rewards relationships.
> - **Using k-NN based local adjustments,** we refine the reward estimation with context-sensitive nonlinear corrections, ensuring that our model outperforms both linear and nonlinear baselines by effectively balancing global and local information.
>
> Regarding Kernel-UCB and Neural-UCB Baselines, please refer to the answer of Q1 as we have analyzed these in the response to that question. Also,  Response to Reviewer 1 (AgPg), Parts 3 and 4 provides an in detail comparison to the existing models. In summary, the selected benchmarks demonstrate the limitations of purely linear or non-linear models, underscoring the value of our unique hybrid synthesis. By integrating global and local perspectives, our approach addresses the shortcomings of existing methods, providing robust performance across dynamic and complex reward settings.
>
>
> ### Performance Comparison Table
> | **Model**      | **Exploration Rate (α)** | **Cumulative Reward** | **Mean Reward** |
> |-----------------|--------------------------|------------------------|-----------------|
> | **LNUCB-TA**   | 0.01                    | **752**                    | **0.94**            |
> | **LNUCB-TA**   | 0.1                     | **741**                    | **0.93**            |
> | **LNUCB-TA**   | 1                       | **752**                    | **0.94**            |
> | Neural UCB | 0.01                    | 726                    | 0.90            |
> | Neural UCB | 0.1                     | 717                    | 0.89            |
> | Neural UCB | 1                       | 722                    | 0.90            |
> | Kernel UCB | 0.01                    | 479                    | 0.60            |
> | Kernel UCB | 0.1                     | 414                    | 0.52            |
> | Kernel UCB | 1                       | 446                    | 0.56            |
>
> Also, the table above demonstrates the comparison between our model and Kernel-UCB and Neural UCB on the news recommendation dataset used in the paper, highlighting that our model consistently outperforms these methods. While Neural UCB performs well in this dataset, we extended the analysis to compare it with our model on the AstroPh co-authorship network in terms of cumulative reward (y-axis) observed at 5% (Figure 7). Neural UCB achieves a cumulative reward of 8636 observed nodes, which is **approximately 12% lower** than the **9808** achieved by our model (LNUCB-TA). Kernel UCB, on the same dataset, achieves 9332 observed nodes, which is **4.85% lower than our model**. Moreover, even our proposed component combined with Epsilon Greedy (green line in Figure 7) surpasses Neural UCB with 8917 observed nodes, **approximately 3.5% higher** than Neural UCB.
>
> Interestingly, in the news recommendation dataset, Neural UCB outperformed Kernel UCB, whereas in the AstroPh co-authorship network, Kernel UCB showed better performance than Neural UCB. Despite these variations, **our model outperformed both methods across both datasets**, demonstrating its robustness and superior adaptability in diverse scenarios. We will provide the updated Figure 7 in our revised manuscript after incorporating all the reviewers’ comments.

---

> > ### Author Response · Authors · 2024-11-23
> >
> > Dear Reviewer  1LHD,
> >
> > We would like to express our sincere gratitude for your valuable insights and suggestions on our work. We have tried our best to address the concerns and queries you raised during the rebuttal process. However, we would greatly appreciate knowing whether our response has effectively resolved your doubts. Your feedback will be instrumental in improving the quality of our work. As the end of the discussion period is approaching, we eagerly await your reply before the end.
> >
> > Sincerely,
> >
> > Authors

---

### Official Review · Reviewer_j6kr · 2024-11-02

**Soundness:** 2
**Presentation:** 1
**Contribution:** 2
**Rating:** 3
**Confidence:** 3

**Summary:**

This work studies the contextual bandit problem with an introduction of k-NN designed to introduce nonlinearity and temporal dependency into the reward function. A set of theoretical analysis (regret upper bound) and experimental results is provided.

**Strengths:**

- The idea of introducing k-NN's to leverage inner structures (i.e., similarity) between arm contexts is an interesting direction. I appreciate the author's effort in this to introduce additional nonlinearity into the system.

- The overall writing and presentation (not from a technical perspective) is satisfying.

As I do have many confusions over the setting and problem itself, I would love to hear clarifications from the authors to further judge this work.

**Weaknesses:**

I am currently holding many confusions over the setting of this work, and thus not readily at a stage to judge this work. I will take a deeper look into the technical contributions after I found myself understood the basics.


Major questions:
1. The reward defined in Eqn. (1) is weird to me in the sense that as an expected reward, it would depend on historically pulled arms and randomly realized reward through the k-NN function. I have not seen similar formulations in the bandit studies, including the previous k-NN UCB paper (Reeve et al., 2018), where I think the k-NN is not a part of the expected reward.

2. With that, is the $\mu_t^a$ vector unknown while also time-varying in Eqn. (1) given the subscript $t$?

3. Is the exploration-exploitation tradeoff discussed around Eqn. (2) a part of the formulation or algorithmic design?

4. The optimal action can also be defined with more clarity. In particular, for each arm, Eqn. (3) says there is an optimal context; however, is the context generated by environment, or is the context (instead of the arm) that the player is selecting (if so, I do not see context selection in the algorithm)? Also, I found no definition of the decision space $D$.

5. The regret definition in Eqn. (4) and its expansion in Eqn. (6) to connect with the single-step regret in Eqn. (5) is work worth debating: Eqn. (5) is measured with respect to the randomly realized reward $Y$, while Eqn. (5) is with respect to the expected rewards? Hopefully the authors can explain Eqn. (6) a bit better, especially clarify the notations.

6. It seems that I found no description of the estimation of $\mu_t^a$ anywhere in the algorithm?

7. Section 3.2 seems to be about selecting a proper $k$ for k-NN; however, is k a parameter that given in the reward defintion?

8. Also, I in general did not understand the purpose of Theorem 2, i.e., what is its statement?

Minor questions:

9. The notations of $\hat{Y}$ and $Y$ are used in a mixed way in Section 2.

**Questions:**

Please refer to weakness. I would love to re-examine this work with the questions addressed.

---

> ### Author Response · Authors · 2024-11-19
> **Response to Reviewer j6kr (Part 1/4 )**
>
> We sincerely thank the reviewer for their valuable feedback and insightful comments. Please kindly find our response to address your Questions.
>
> **W1 (Reward Structure):**
>
> Please kindly refer to our response to W1 & Q1 of Reviewer 1 (Reviewer AgPg).
>
> **W2 ($\mu_t^a$):**
>
> Indeed, $\mu_t^a$ represents the coefficients associated with the expected reward model for arm $a$ at time $t$. As new data (contexts and rewards) are observed, this vector is both unknown and time-varying. With the increasing collection of data, the model updates $\mu_t^a$ to reflect changes in reward structure, thereby capturing changes in the relationship between context and reward over time.
> Contextual MAB algorithms are characterized by this time-varying characteristic. As more context-reward pairs are observed, the model is able to refine its prediction of the expected reward for each arm by continually refining the value of $\mu_t^a$. In line with standard MAB models, where new observations are made to refine the model's understanding of the underlying reward function, this approach is consistent with the model's iterative updating of the expected reward.
> As stated in lines 779-795, a ridge regression model is used to estimate the reward model, which forms the basis of the update process for $\mu_t^a$. As a result of the solution to this regression problem, we get a valid estimate of the true $\mu_t^a$. We have provided details on this process and the associated uncertainty region in Corollary 1 of the Appendix. As more data is accumulated, the model refines its estimate of $\mu_t^a$, which represents the relationship between the context and the expected reward for arm $a$. The uncertainty region quantifies the confidence in this estimate, and as the number of observations $N_t^a$ increases, the size of this uncertainty region decreases. This reflects a growing certainty in the model's expected reward predictions, driven by the additional information provided by the accumulated data.
> This formulation captures the "optimism in the face of uncertainty" principle, central to our approach. As the model accumulates more data, the confidence in the estimate $\hat{\mu}_a^t$ improves, narrowing the uncertainty region and enhancing the decision-making process. Therefore, the time-varying nature of $\mu_t^a$ reflects both the iterative learning process and the evolving confidence captured by the uncertainty region, which is an important component of exploration and exploitation. By updating the model over time, the model is able to adapt to evolving reward structures.
>
> **W3 (Exploration-exploitation Tradeoff ):**
>
> As mentioned in Section 3.1 (Overall concept), the attention-based exploration-exploitation is one of the main novel proposed components. In LNUCB-TA, the formula in Eqn. (2) is a central component of the algorithmic design, as it dynamically adjusts the exploration parameter. This formulation is intended to provide real-time adjustments to the exploration-exploitation balance based on both the global performance of all arms and the local reward patterns of each arm (lines 22-28, 102-105, 124-129).
> As stated in lines 85-97, in traditional MAB models, the exploration-exploitation trade-off is often managed by a fixed or manually tuned parameter. However, our approach introduces an adaptive strategy that continuously adapts as a result of observed performance. Due to the dynamic adjustment mechanism, manual fine-tuning of exploration parameters is eliminated. Instead, the model automatically adapts the exploration-exploitation balance in real time, thus responding effectively to changes in reward patterns without extensive parameter tuning or prior knowledge of the optimal exploration rate. Consequently, a more efficient and adaptive learning process is ensured. Thus, exploration-exploitation trade-offs are integral to the LNUCB-TA algorithm's design and operation.
> In Section 3.3, which provides further clarification on how the exploration-exploitation balance is handled, we have provided a more detailed explanation of this dynamic adjustment and its role in the overall algorithmic framework.
>
> **W6 (Estimation of $\mu_t^a$):**
>
> Please kindly refer to Corollary 1 and Definition1 in the Appendix.
>
> **W7 (Section 3.2):**
>
> In LNUCB-TA, k is not a fixed parameter but is dynamically determined for each arm during each time step. Unlike traditional approaches where k may be predefined, static, or chosen based on function optimization, our method dynamically adjusts k at **each time step**, adjusting it to **the specific arm** and current observations. This leads to **less time complexity** compared to existing methods (lines 320-323). Also, in this section, specific scenarios with high and low rewards have been discussed (lines 312-319). As a result of this process, the model is able to adjust its estimations in real-time as local reward variations are incorporated, which constitutes a significant improvement over existing methods.

---

> ### Author Response · Authors · 2024-11-19
> **Response to Reviewer j6kr (Part 2/4 )**
>
> **W4 (Problem Setup):**
>
> ### Context Generation and Selection:
>
> As noted in line 171, the feature vector $X_t$ is drawn independently and identically distributed (i.i.d.) from a fixed marginal distribution over the context space $X$. This means that the context $X_t$ at each time step $t$ is generated by the environment according to the probability distribution, which is fixed but may vary depending on the specific application or problem at hand.
> Context is not chosen by the agent at each time step but rather is provided by the environment. In the proposed model, the agent is required to select the arm that maximizes the expected reward based on the context, which is determined by the hybrid model of linear and nonlinear components in the algorithm. Even though the agent does not choose the context, it plays an important role in determining the reward structure for each arm, as well as informing the decision-making process.
>
> ### Optimal Action and Decision Space $D$:
>
> In Eqn. (3), the "optimal context" $(x_a^t)^*$ refers to the best possible context (in terms of maximizing expected reward) for each arm. This context incorporates both the global linear model and the local understanding provided by the $k$-NN term. Specifically, the optimal context is defined by the number of nearest neighbors $k$ in the $k$-NN function, which adds a local adjustment to the global linear prediction. This combination allows the model to account for both broader trends captured by the linear model and finer, local structures revealed by the $k$-NN adjustment.
> The decision space $D$ represents the set of all possible contexts that the model evaluates. This space is crucial because it facilitates a thorough exploration of potential scenarios. In the LNUCB-TA model, the "arm-dependent optimal action" refers to the best reward obtained for arm $a$ based on its history over $t$ steps, leading to the theoretical optimal action $\pi_t^*$. As noted in lines 213-215, while we compute the optimal action for each arm, the model ultimately selects only one arm to play at each time step, choosing the arm with the highest expected reward. This ensures that the model is both locally optimal (for each arm) and globally optimal (across all arms) at that particular time step.
>
> ### Game Protocol:
>
> At each time step $t$, our model evaluates potential rewards for each arm based on its specific context $x_a^t$ and historical rewards $z_a^t$. While the model computes what the optimal action would be for each arm independently, only one arm is actually selected to play. This selection is made by choosing the arm that, according to the model's computations, offers the highest expected reward at that time. In this way, the decision-making process ensures that the model adapts to each arm's unique circumstances while still making a single, globally informed decision that optimizes the expected reward across all arms.
>
> ### Simplified Example
>
> Imagine a news recommender system where the goal is to recommend articles to users while maximizing user engagement (e.g., clicks). Here's how the proposed LNUCB-TA model operates step-by-step in this context:
>
> #### **Setting:**
> - **Arms:** Categories of news articles (e.g., Sports, Politics, Entertainment).
> - **Context ($X_t$):** User profile features at time $t$, such as age, location, device type, and browsing history.
> - **Reward ($Y_t$):** User engagement (e.g., click or no click on the recommended article).
> - **Decision Space ($D$):** The set of all possible contexts $X_t$ across all arms.
>
> #### **Example Scenario:**
> At time $t$, a user logs into the platform with the context vector:
> $X_t = [\text{age: 25, location: New York, device: mobile}]$.
> This context is provided by the environment, not chosen by the agent.
>
> 1. **Optimal Action Computation for Each Arm:**
>    For each arm $a$ (e.g., Sports, Politics, Entertainment), the model evaluates the optimal context $(x_a^t)^*$ that maximizes the expected reward based on Equation (3) of the paper:
>    $$(x_t^a)^* \in \arg\max_{x \in D} \left( (\mu^a)^* \cdot (x_t^a) + f_{k,t}^a(x_t^a, z_t^a) \right)$$
>    Here, $(x_a^t)^*$ referes to the optimal context for arm $a$, which includes the optimal number of nearest neighbors $k$ used in the $k$-NN adjustment. The optimal $k$ dynamically adjusts based on the reward history ($z_a^t$) and the current context.
>    - For **Sports**, the optimal $k$ leading to optimal context for this arm may focus on smaller neighborhoods to reflect niche interests for Sports articles.
>    - For **Politics**, the optimal $k$ leading to optimal context for this arm may expand to include a broader historical dataset, capturing more generalized trends for this category.
>    - For **Entertainment**, the optimal $k$ leading to optimal context for this arm might fall somewhere in between, reflecting moderate local patterns.
>
> Please kindly refer to next part for the continue of the example.

---

> ### Author Response · Authors · 2024-11-19
> **Response to Reviewer j6kr (Part 3/4)**
>
> **Continue of Example in W4:**
>
> 2. **Expected Reward Calculation for the Current Context:**
>    After determining the optimal $k$ for each arm (embedded in $(x_a^t)^*$), the model computes the expected reward for the actual context provided by the environment based on Equation (1) in the paper:
>    $$o^a_t(x^a_t, z^a_t) = l^a_t(x^a_t) + f_{k,t}^a(x_t^a, z_t^a) = \mu_t^a \cdot x_t^a + \text{k-NN}_{k,t}^a(x_t^a, z_t^a).$$
>
>   This allows the model to calculate both the global reward trends ($\mu_t^a \cdot x_t^a$) and the local reward adjustments ($f_{k,t}^a(x_t^a, z_t^a)$) for each arm:
>    - **Sports:** $o_{\text{Sports}}^t = 0.4$
>    - **Politics:** $o_{\text{Politics}}^t = 0.6$
>    - **Entertainment:** $o_{\text{Entertainment}}^t = 0.8$
>
> 3. **Action Selection:**
>    While the model computes the optimal context $(x_a^t)^*$ and evaluates the expected reward for each arm, only one arm is selected to play. In this case, Entertainment is selected because $o_{\text{Entertainment}}^t = 0.8$ is the highest expected reward.
>
> 4. **Reward Observation and Updates:**
>    After selecting Entertainment, the realized reward (e.g., click or no click) is observed.
>    The parameter $\mu_a^t$ is updated via ridge regression, and the local adjustment $f_{k,t}^a(x_t^a, z_t^a)$ is refined by incorporating the new reward into the historical reward data.
>
> 5. **Game Protocol:**
>    The optimal context $((x_a^t)^*)$ for each arm at time $t$ includes the dynamically adjusted $k$, ensuring the $k$-NN component provides the best local adjustment for that arm.
>    Even though the optimal context is computed for each arm, only one arm is selected to play, and only its reward contributes to learning at this step.
>
>
> **W8 (Theorem 2):**
>
> In Theorem 2, we illustrate how an attention mechanism can dynamically update the exploration parameter over time, effectively balancing global and local rewards information. Specifically, $\alpha$ is updated based on number of arm selections, global attention (the global performance across all arms), and local attention (the performance of specific arm).
>
> As more data is available, this dynamic adjustment enables the model to switch between exploration and exploitation. As a result of infrequent selection, the model explores more, and as its understanding of arms' performance improves, the model increasingly exploits this knowledge. We have demonstrated this approach both theoretically and experimentally, showing substantial improvements in bandit decision-making. An attention mechanism "can be designed" demonstrates one of the significant contributions of our study: the ability to integrate attention-based exploration rates into a wide variety of bandit models. As a result of this flexibility, models such as **Epsilon Greedy**, **BetaThompson**, and **LinThompson** are able to incorporate dynamic and adaptive exploration strategies, which significantly enhance their decision-making capabilities (lines 1385-1390).
>
> In the paper, we introduce an adaptive weighting system for exploration rates based on attention mechanisms for each arm. A data-driven and context-sensitive exploration decision is ensured by attention-based adjustment, ensuring that the models are more adaptable. The results of this mechanism is shown in **Figure 5** and **Table 4**. The summary of the improvement can be found in the table below:
>
>
> Table 2: Effect of Attention-Based Exploration Rate on Other Bandit Models. Best mean reward (BMR), best cumulative reward (BCR), and the improvement percentage over the base model.
>
> | **Model**                   | **BMR** | **BCR** | **Imp. Over Base Model (%)** |
> |-----------------------------|---------|---------|------------------------------|
> | **BetaThompson-enhanced**   | 0.79    | 632     | **259.09**                      |
> | **Epsilon Greedy-enhanced** | 0.58    | 464     | **123.08**                      |
> | **LinThompson-enhanced**    | 0.69    | 552     | **64.29**                       |
>
>
>
> Regarding the intuition, as stated in lines 326-331, the $\alpha_{N_t^a}$ adjusts the exploration rate through an attention mechanism that considers both **temporal** and **spatial** variations in data. This dual consideration allows for dynamic adjustment of exploration efforts based on time-dependent changes (**temporal**) and distinct reward patterns across different arms (**spatial**). By integrating these aspects, the mechanism enhances the model's adaptability to real-time changes, ensuring more effective exploration and exploitation.
>
> This approach is particularly novel as it moves beyond static or merely context-aware adjustments seen in other models (lines 85-97). The robustness of this approach is confirmed in our empirical results (**Figure 2**, and **Figure 6**) and supported by **Theorem 2**, illustrating that our model maintains consistent performance across a range of $\kappa$ and $\alpha_0$ values, which marks a substantial advance over current models.

---

> > ### Author Response · Authors · 2024-11-19
> > **Response to Reviewer j6kr (Part 4/4 )**
> >
> > **W5 (Regret):**
> >
> > ### Regret Analysis
> >
> > As stated in lines 216-239, the regret for our model at any time step is based on the arm selected. Equation 5 of our manuscript describes regret calculation where $g^a ((x_t^a )^*,(z_t^a )^* )$ represents the optimal expected reward for arm $a$ at the optimal context $(x_t^a )^*$, which is the feature vector yielding the highest reward (lines 225-226). The function $o_t^a (x_t^a,z_t^a )$ represents the expected reward under the decision made by the policy $\pi_t^a$ at context $x_t^a$ with reward history $z_t^a$. Given this framework, regret inherently accrues only for the arm that is actively selected at each timestep, and by default, it is zero for any arm not chosen (line 227).
> >
> > ### Regret Definition in Eqn. (4) and (5):
> > The total regret $R_T (\pi)$ (Eqn. (4)) measures the cumulative performance difference between the chosen policy $\pi$ and the optimal policy $\pi^*$ across all time steps $t$ (lines 216-220). It sums the difference between the rewards obtained from the optimal arm $\pi_t^*$ and the rewards obtained from the arm selected by the policy $\pi_t$ at each time step $t$:
> >
> > $$
> > R_T (\pi) = \sum_{t \in [T]} \left( Y_t^{(\pi_t^*)} - Y_t^{(\pi_t)} \right)
> > $$
> >
> > Here, $Y_t^{(\pi_t^*)}$ and $Y_t^{(\pi_t)}$ represent the realized rewards for the optimal and selected arms, respectively, at time step $t$. The aim is to quantify the difference between the performance of the optimal policy and the policy under consideration, which is the standard in the literature.
> >
> > In Eqn. (5), the regret for a single arm $a$ at time step $t$ is defined as the difference between the optimal expected reward for that arm at the optimal context $(x_t^a)^*$ and the expected reward under the decision made by the policy $\pi_t^a$ at context $x_t^a$:
> >
> > $$
> > \text{regret}_t^a = \Delta_t^a \left( g_t^a ((x_t^a )^*,(z_t^a )^*) - o_t^a (x_t^a,z_t^a ) \right)
> > $$
> >
> > Here, $\Delta_t^a$ is the indicator function that takes the value 1 if arm $a$ is selected at time $t$, and 0 otherwise. This formulation uses expected rewards, which are computed with respect to the underlying model of expected reward based on the context and the history of rewards, rather than the realized reward .
> >
> > ### Regret Expansion in Eqn. (6):
> > In Eqn. (6), we expand the cumulative regret over all arms and time steps. The total regret $R_T$ can be written as the sum of the individual regrets for each arm, where each arm's regret depends on the difference between the optimal expected reward (including both the linear and $k$-NN terms) and the expected reward for the arm selected under the policy.
> > Specifically, Eqn. (6) expresses the total regret as:
> >
> > $$
> > R_T = \sum_{a=1}^A \sum_{t=0}^T \Delta_t^a \left( (\mu_t^a )^* \cdot (x_t^a )^* + k\text{-}NN_{k,t}^{a} ((x_t^a )^*,(z_t^a )^*) - \mu_t^a \cdot x_t^a + k\text{-}NN_{k,t}^{a} (x_t^a,z_t^a ) \right)
> > $$
> >
> > The terms $(x_t^a )^*$ and $x_t^a$ refer to the optimal context and the chosen context at time $t$ for arm $a$, while $k\text{-}NN_{k,t}^{a}$ refers to the $k$-NN function for arm $a$ that adjusts the reward based on past observations. The expectation is over the reward function's structure, not the realized rewards, as we are dealing with expected rewards rather than the observed ones.
> >
> > ### Reward
> > - **Expected rewards** refer to the rewards predicted by the model (including both the linear and $k$-NN components).
> > - **Realized rewards** represent the actual rewards observed at each time step, which include noise and stochastic variations not captured by the model.
> >
> > The regret analysis in Eqns. (4) and (5) is based on expected rewards, which is standard practice in theoretical bandit settings. The regret measures how well the algorithm performs compared to the optimal policy that knows the expected rewards.
> >
> > ### Summary of Regret Analysis:
> > 1. **Total Regret Definition (Eq. 4):**
> >    Total regret is defined as the difference in rewards obtained by the optimal policy and the chosen policy over all time steps. This is a standard high-level measure of performance deviations.
> >
> > 2. **Per-Arm Regret (Eq. 5):**
> >    In this equation, the per-arm regret is calculated by comparing the expected reward under the optimal policy with the expected reward under the chosen policy. Both global linear and local $k$-NN influences are included in the per-arm regret.
> >
> > 3. **Expanded Total Regret (Eq. 6):**
> >    Specifically incorporates linear and $k$-NN components in the reward difference when aggregating the regret across all arms and time steps. As a result of this expanded form, the cumulative regret computation can be integrated with the hybrid reward model by connecting total regret to individual arm regrets.
> >
> > As a result of these equations, the total regret is linked to the details of each arm and context, emphasizing the structure of the hybrid model as it applies to regret analysis.
> >
> > **W9 (Notation):**
> >
> > Thank you for your valuable comment. We will incorporate this into the revised version.

---

> > > ### Author Response · Authors · 2024-11-23
> > >
> > > Dear Reviewer  j6kr,
> > >
> > > We would like to express our sincere gratitude for your valuable insights and suggestions on our work. We have tried our best to address the concerns and queries you raised during the rebuttal process. However, we would greatly appreciate knowing whether our response has effectively resolved your doubts. Your feedback will be instrumental in improving the quality of our work. As the end of the discussion period is approaching, we eagerly await your reply before the end.
> > >
> > > Sincerely,
> > >
> > > Authors

---

> ### Comment · Reviewer_1LHD · 2024-11-23
> **Response to authors**
>
> Hi authors,
>
> 1. I have gone over some parts of your response. I will take this into consideration while formulating the final rating during the discussion phase.
> 2. I just want to highlight that it is not a good rebuttal technique to give an overwhelming amount of response to a reviewer (or all reviewers). If the draft requires so much explanation and clarification it is a marker that the paper is not well written.

---

> > ### Author Response · Authors · 2024-11-23
> >
> > Dear Reviewer 1LHD,
> >
> > Thank you for your response and for considering our rebuttal in your evaluation. We would like to address your feedback regarding the length of our responses and explain the rationale for our approach, particularly as it pertains to the specific feedback from you and other reviewers.
> >
> >
> >
> > For **Respected Reviewer AgPg**, their feedback included requests to compare our model with ten additional existing models and to integrate more literature. Naturally, this required us to provide a detailed and extensive response to address their concerns comprehensively.
> >
> >
> > For **Respected Reviewer 7MDe**, they requested additional experiments and posed questions about the limitations and potential directions for future studies. This necessitated a detailed and careful explanation to ensure we addressed their valuable feedback appropriately.
> >
> >
> > For **Respected Reviewer j6kr**, they raised nine distinct questions, of which they stated would directly impact their final evaluation after being addressed. Out of respect for their engagement and to provide sufficient clarity, we crafted detailed responses to each point to ensure no ambiguity remained.
> >
> >
> > For **Respected Reviewer 1LHD (yourself)**, we aimed to address your detailed feedback thoroughly. Your questions on the mathematical formulations and requests for additional model comparisons required comprehensive responses to ensure clarity and completeness. The length of our responses reflected our commitment to providing the necessary details for an accurate evaluation.
> >
> > Additionally, we would like to highlight that the inclusion of extensive mathematical formulas **(often taking 500 characters or more)** and the 5000 character limit per response naturally led to more segmented replies. We appreciate your engagement and effort in reviewing our work.
> >
> > In summary, the length of our responses was not intended to overwhelm but to respect the depth and breadth of the feedback provided by each reviewer. We believe this level of detail was necessary to address the specific concerns raised. We greatly value your time and effort in reviewing our paper and hope this explanation clarifies our approach.
> >
> >
> >
> >
> > Sincerely,
> >
> > The Authors

---

> ### Comment · Reviewer_j6kr · 2024-11-24
>
> I would like to thank the authors for the response. I am with Reviewer 1LHD that a more clear response is desired. Still, I would like to see some further clarifications from the authors.
>
> - **W1, W2, W7.** I understand the intention of the authors to have the k-NN in the reward model to capture the impact of previous rewards on later ones. Let me probably ask this question first before any further ones: is $k_t^a$ a parameter that is defined clearly in the problem formulation, e.g., as a pre-fixed sequence? Now I only see the selection of it in the later algorithm designs but not in the problem formulation (which should stand alone without the algorithm design).
>
> - **W3.** If it is a part of the algorithm design, please remove it from the problem formulation.
>
> - **W4.** I feel a bit lost in the response. Probably also let me clarify what is the difference between the environment-generated context $x_t$ and the optimal context for each arm $(x^{a}_t)^*$.
>
> It would be nice if the authors can first answer the above questions and then I can move to the other parts of the response, especially the regret.

---

> ### Author Response · Authors · 2024-11-24
>
> We would like to thank the respected reviewer for their comments and time.
>
> ### **Response to W1, W2, W7:**
>
> We appreciate your question regarding $\(k_t^a\)$ and its role in the reward model. As mentioned before, $\(k_t^a\)$ is not pre-fixed or defined as a static sequence in the problem formulation. Instead, it is dynamically adjusted based on the reward history. This adaptivity is a central part of our approach, ensuring the model's flexibility in capturing local reward patterns effectively.
>
> **Yes**, the dynamic adjustment of $\(k_t^a\)$ has been explicitly highlighted throughout the paper, including in:
>
> - **Abstract:** Line 20
> - **Contribution:** Lines 100, 106–107, Table 1
> - ******Problem Definition:****** **Lines 173–176** (#**addressing your specific question**#)
> - **Section 3.2:** Lines 289–290, Algorithm 2 (Line 5), Lines 308–311
>
> We hope this clarification resolves your concerns and highlights how $\(k_t^a\)$ is embedded as a dynamic and adaptive component of the algorithm, rather than being pre-fixed.
>
> ---
>
> ### **Response to W3:**
>
> Respectfully, we believe it is essential to include the exploration-exploitation trade-off in both Section 2 (Hybrid Contextual MAB Learning) and Section 3 (Methodology) to present our approach effectively. These sections have distinct roles and together provide a complete understanding of our model.
>
> Section 2 introduces the trade-off as a core aspect of the problem, highlighting our novel approach with the attention-based exploration parameter. This ensures readers understand that our method goes beyond standard exploration-exploitation strategies by dynamically adapting the exploration rate based on both global and local rewards. Without this inclusion, readers would not fully understand the problem formulation and could assume it aligns with standard bandit models, missing the distinction in how the exploration-exploitation trade-off is addressed.
>
>
> Section 3 then details how this trade-off is implemented using temporal attention (e.g., Algorithm 3). This connection between problem definition and practical implementation ensures the novelty and adaptability of our method are clearly conveyed. Both sections are necessary to differentiate our work and provide a cohesive explanation.
>
> ---
>
> ### **Response to W4:**
>
> **$x_t$:**
> Each context $x_t \in \mathcal{X}$ at time $t$ corresponds to a set of possible actions, or "arms," indexed by $a$ within the set $\mathcal{A} = \{1, \ldots, A\}$, where $A$ is the total number of arms (**lines 158-160**).
>
> **$x_t^a$:**
> Represents the specific feature vector for arm $a$ at time $t$ (**line 172**).
>
> While $x_t$ originates as the same environmental input for all arms, it is utilized uniquely for each arm $a$, resulting in the arm-specific feature vector $x_t^a$ tailored to the decision-making process for that particular arm.
>
> **$(x_t^a)^\*$:**
> Is the optimal context for arm $a$, computed as the feature vector that maximizes the expected reward for arm $a$ at time $t$ (**line 206**).
>
> It is derived from $x_t$ by dynamically adjusting the best available historical data through the adaptive $k$-NN mechanism (**lines 207-210**).
>
>
>
> ### Key Differences:
> - **Global vs. Arm-Specific:**
>   $x_t$ is global and shared across all arms, while $(x_t^a)^\*$ is arm-specific and tailored to maximize the reward for arm $a$.
>
> - **Environment-Provided vs. Computed:**
>   $x_t$ is provided by the environment, whereas $(x_t^a)^\*$ is computed by the model.
>
> - **Static vs. Dynamic:**
>   $x_t$ is static and unchanging during the decision process, while $(x_t^a)^\*$ is dynamically adjusted for each arm based on its historical performance and context similarity.
>
> - **Purpose:**
>   $x_t$ defines the context for evaluating all arms, whereas $(x_t^a)^\*$ identifies the best possible context for arm $a$, ensuring optimal reward prediction.
>
>
>
> We hope this clarification has addressed your concerns effectively.

---

> > ### Comment · Reviewer_j6kr · 2024-11-24
> >
> > Thank you for the quick response.
> >
> > My question for the k-NN is then more valid: the problem formulation should stand alone, i.e., we do not need to talk about the algorithm when we formulate a problem. In this way, we can define the optimality and learning target, and then design an algorithm to solve it. Here, the problem formulation and algorithm design seem to be mixed together, which forms a logistic loop.
> >
> > Regarding the contexts, I do understand more clearly the setting that the authors want to address.

---

> > > ### Author Response · Authors · 2024-11-24
> > >
> > > Thank you for your thoughtful response.
> > >
> > > We acknowledge your perspective that a clear separation between problem formulation and algorithm design is ideal for defining optimality and learning targets. However, in our specific case, the dynamic adjustment of $k$ in the $k$-NN mechanism is inherently tied to the problem's objective of adapting to changing reward structures in non-stationary environments. This dynamic nature of $k$ is not merely an algorithmic choice but a fundamental component of how the problem itself is framed, as it directly impacts the reward estimation process.
> > >
> > > Our inclusion of the dynamic $k$-NN mechanism in the problem formulation is consistent with practices in bandit models for example NeuralUCB [1]. NeuralUCB defines the reward function $h(x)$ as part of the problem setting (equation 2.2), which inherently depends on the neural network's structure and properties (equation 2.3 in the same problem setting section). Similarly, our model incorporates the $k$-NN adjustment in the problem definition because it is essential for capturing local reward dependencies, just as NeuralUCB's $h(x)$ models non-linear reward functions.
> > >
> > > That said, we will revise the manuscript to ensure that the problem formulation is presented more distinctly, highlighting the learning target and leaving implementation details to the algorithmic section. We appreciate your feedback as it will help us refine the clarity and structure of our paper.
> > >
> > > We are glad that the clarification on the contexts was helpful and addressed your concerns in that regard.
> > >
> > > [1] Zhou, D., Li, L. and Gu, Q., 2020, November. Neural contextual bandits with ucb-based exploration. In International Conference on Machine Learning (pp. 11492-11502). PMLR.

---

### Official Review · Reviewer_7MDe · 2024-11-04

**Soundness:** 3
**Presentation:** 3
**Contribution:** 3
**Rating:** 6
**Confidence:** 4

**Summary:**

This work studies the problem of contextual bandits and develops a novel UCB style algorithm leveraging linear relationship between contextual features and reward for each arm along with non-linear estimation with k-NN . The developed algorithm addresses some important limitations found in existing algorithms like LinUCB that relies on linear relationship between the context in the reward formulation. By introducing an adaptive k-Nearest Neighbors that adjust with reward variance LNUBC TA encompasses the non-linearity that is computationally not explosive. Along with dynamic exploration term, LNUCB seem to perform much better than available state of the art algorithms in contextual MABs. The authors also provided theoretical regret guarantees which achieve sub-linear regret bounds.

**Strengths:**

1.	This work addresses the issue of computational efficiency when dealing with non-linearity in the contextual features which is an essentially an important component in the usage of these algorithms in real-world scenarios. Generally, most of the existing approaches to solve the non-linear relationship between contextual features and reward including KNN-UCB are computationally expensive and LNUCB-TA tend to solve the time complexity issues commonly associated with nonlinear models

2.	The work presents a solid theoretical guarantee in the form of regret to match the theoretical performance of the existing algorithm with its sub-linear regret bound. Thus, showcasing its solid performance.

3.	The work also presents a strong empirical result in various regimes with multiple datasets to showcase its performance in real-world scenario.

4.	The authors also provide a detail regarding the criteria’s for k-selection that is intuitively based on the variance in rewards for each arm at time $t$ thereby solving the previous gaps found in KNN-UCB and other algorithms.

**Weaknesses:**

1.	Though the problem setting is interesting, the paper studies the contextual MAB problem, it is an incremental work extending the basis of LinUCB and KNN UCB with the inclusion of adaptive KNN as the non-linear factor in the reward estimation.

2.	The work doesn’t include any results corresponding to the sample complexity of this algorithm to understand the sampling regiment.

3.	The work also doesn’t include any detailed regret comparison to other algorithms that exist in the contextual Multi armed bandits problem space.

**Questions:**

1.	The concept of UCB itself involves considering the unit reward (total reward/ no of selection). What additional information does temporal attention brings to the estimated quantity in decision making and how do they differ? This can help understand the temporal attention term better.

2.	The work compares its empirical performance to LinUCB, LinUCB with KNN etc., however a detailed comparison of LNUBC TA ‘s theoretical regret bound guarantees with the aforementioned algorithm is not discussed. Having those details can help better understand the theoretical performance guarantees.

3.	The empirical evaluation focuses mainly on recommendation systems and network exploration. How would the model perform in other domains with different reward structures, such as healthcare or finance?

4.	Also, the algorithm seems to rely heavily on the fixed weight between linear and nonlinear components in the measuring quantity. Does this limit the performance of the algorithm in specific regimes? If so, a detailed discussion on it will help bring more clarity to those. Also, would it be possible to make the weight some form of a tunable parameter?

5.	K-NN relies on the dimensionality of the context features. What happens when the contextual features are large enough that it suffers from large dimensional space. There seem to be no sufficient details on the potential limitations in high-dimensional feature spaces. Could you include more details about this?


6.	$f_a$ uses the observed rewards from $za_t$ for closest neighbors a_{k,t} in terms of context similarity based on Euclidian distance. Why was Euclidian distance metric chosen?

7.	Also, how does it react when there are many irrelevant features in high-dimensional context as it uses Euclidian ball?

---

> ### Author Response · Authors · 2024-11-19
> **Response to Reviewer 7MDe (Part 1/3 )**
>
> We sincerely thank the reviewer for their valuable feedback and insightful comments. Please kindly find our response to address your Questions (Q).
>
> **Q1 (Temporal Attention):**
>
> Temporal attention modifies the exploration term dynamically by introducing an adaptive scaling factor \( \alpha_{N_t^a} \), which incorporates both global trends and local arm-specific behavior:
> $\alpha_{N_t^a} = \frac{\alpha_0}{N_t^a + 1} \cdot \left( \kappa g + (1 - \kappa) n_t^a \right).$ This scaling directly influences the UCB confidence bound as mentioned in Equation (7) of the paper. As elaborated in Section 3.3, this adjustment integrates additional temporal and contextual information into the reward estimation process, as detailed below:
>
> ### Global Reward Trends
> Global attention summarizes the overall reward trends across all arms, reflecting system-wide behavior. The model adjusts exploration based on how an arm's rewards compare to the global average. This ensures that the confidence bound reflects not just the arm's own history but also its relative performance in the broader context.
>
> ### Local Arm-Specific Performance
> Local attention captures the recent reward behavior specific to the chosen arm, focusing on its individual dynamics. This ensures that the confidence bound is sensitive to arm-specific changes, such as sudden improvements or deteriorations in reward patterns.
>
> ### Temporal Adaptivity
>
> The term ($\frac{\alpha_0}{N_t^a + 1}$) decays over time, ensuring that the exploration term diminishes as the arm is sampled more often, leading to more confident estimates. This decay is modulated by the GALA concept, allowing the model to dynamically adjust the rate of decay based on specific reward dynamics.
>
> ### Context-Aware Confidence Bounds
> Ultimately, temporal attention enhances the estimated reward quantity by using Context-Aware Confidence Bounds, specifically incorporating the GALA concept. As a result of context-aware adjustment, reward estimates are more accurate, especially in dynamic and non-stationary environments.
>
> The exploration parameter in standard UCB is typically static or only weakly adaptive, relying on simple decay or a fixed schedule. Temporal attention expands this concept by introducing dynamic exploration scaling, which adjusts exploration according to real-time reward trends at both global and local levels.
>
> **Q2 (Theoretical Regret):**
>
> Thank you for your valuable feedback and the suggestion to provide a detailed theoretical comparison between LNUCB-TA and other algorithms. Below, we address your point by summarizing the regret guarantees for key baseline algorithms and highlighting the distinctions in their theoretical performance.
>
> ## Theoretical Comparison
>
> The regret bounds for LNUCB-TA and other contextual bandit algorithms are summarized in the table below. This table highlights how LNUCB-TA compares theoretically to algorithms like LinUCB, Kernel-UCB, and Neural-UCB:
>
> | **Algorithm** | **Regret Bound**       | **Strengths**                                     | **Weaknesses**                                    |
> |---------------|------------------------|--------------------------------------------------|--------------------------------------------------|
> | **LNUCB-TA**  | $\( \tilde{O}(\sqrt{dT}) \)$ | Adaptive to global and local patterns; real-time exploration adjustment | Requires k-NN tuning; computational cost higher than purely linear models |
> | **LinUCB**    | $\( O(\sqrt{dT \log T}) \)$ | Simple; effective for stationary, linear settings | Fails in non-linear or non-stationary environments |
> | **Kernel-UCB**| $\( O(\sqrt{T}) \)$        | Captures non-linearities with kernels            | Fixed kernel limits adaptability; computationally expensive |
> | **Neural-UCB**| $\( \tilde{O}(\sqrt{T}) \)$ | Models complex non-linear relationships          | High computational cost; slower adaptation in non-stationary settings |
>
> ## Key Takeaways
>
> 1. The LNUCB-TA algorithm achieves a regret bound of \( \tilde{O}(\sqrt{dT}) \), which is comparable to other state-of-the-art algorithms such as LinUCB and Kernel-UCB. However, unlike Kernel-UCB or Neural-UCB, our model is inherently adaptive, making it suitable for non-stationary environments without retraining or kernel selection.
>
> 2. Several baseline models, such as LinUCB and Neural-UCB, assume either stationarity or a fixed reward structure, which limits their effectiveness in dynamic environments. Through the use of its temporal attention mechanism and adaptive k-NN component, LNUCB-TA overcomes these limitations, enabling it to adapt to both local and global patterns as the environment changes.
>
> 3. By incorporating both a global linear model and an adaptive k-NN component, LNUCB-TA achieves a dual focus of capturing global trends while adapting to local variations. Using this unique approach, the model remains robust in a variety of scenarios, from simple linear trends to highly non-linear ones.

---

> ### Author Response · Authors · 2024-11-19
> **Response to Reviewer 7MDe (Part 2/3)**
>
> **Q3 (Other Domains):**
>
> We appreciate your comment regarding the application of our model in other domains like healthcare or finance. To address this, we evaluated the performance of our model, LNUCB-TA, on the widely studied Warfarin dataset, which is a benchmark dataset used for modeling personalized medicine decisions. Specifically, this dataset focuses on predicting the correct warfarin dosage for patients based on their clinical and genetic profiles. Below, we summarize the setup, dataset details, and results:
>
> 1. **Dataset and Problem Setup:**
>    - As part of the Warfarin dataset, clinical and genetic characteristics of patients are provided as well as the optimal dosage of warfarin required for each patient. We categorize the dosages into three clinically relevant categories based on the setup described in Bastani and Bayati's *"Online Decision-Making with High-Dimensional Covariates"*:
>      - Low dosage (< 3mg/day),
>      - Medium dosage (3-7mg/day),
>      - High dosage (> 7mg/day).
>      Each category is treated as an arm in a 3-armed contextual bandit problem. In this setup, we use a binary reward function where the algorithm receives a reward of 1 if the predicted dosage matches the true dosage and 0 otherwise.
>
> 2. **Model Performance Evaluation:**
>    - To empirically validate LNUCB-TA's performance, we compared it against purely linear (LinUCB) and purely non-linear (k-NN UCB) models using different exploration rates.
>
> Table 1: Comparison of Models based on Approximate Cumulative Rewards for Different Exploration Rates on Warfarin Dataset
>
> | **Model**    | **Exploration Rate** | ~**Cumulative Reward (t = 5528)** |
> |--------------|-----------------------|-----------------------------------|
> | **LNUCB-TA** | 0.01                 | **3000**                            |
> | **LNUCB-TA** | 0.1                  | **2900**                            |
> | **LNUCB-TA** | 1                    | **2400**                            |
> | KNN-UCB | 0.01                 | 1750                            |
> | KNN-UCB  | 0.1                  | 1750                            |
> | KNN-UCB  | 1                    | 1500                            |
> | LinUCB | 0.01                 | 2100                            |
> | LinUCB   | 0.1                  | 1500                            |
> | LinUCB   | 1                    | 1750                           |
>    - **Key Insights from the Results:**
>      - As compared to LinUCB and k-NN UCB, LNUCB-TA consistently achieved higher cumulative rewards across all exploration rates. This demonstrates its ability to leverage both global (linear) trends and local (non-linear) patterns, which is crucial in healthcare settings where patient responses may vary significantly based on both broad population-level patterns and individual characteristics.
>      - In spite of LinUCB's ability to identify global patterns, its inability to account for local variations limits its applicability to such datasets. Although k-NN UCB incorporates non-linear components, it faces challenges in higher-dimensional feature spaces, resulting in lower performance compared to LNUCB-TA.
>      In light of the results, LNUCB-TA appears to be particularly suitable for personalized decision-making problems in healthcare, where both structured clinical data (global trends) and individual data (local variations) may be present. Also, as a result of its attention mechanism, it is able to adapt to changing environments, which is a critical aspect of healthcare decision-making in a dynamically changing environment.
>
> The new figure representing the exact values on the Warfarin dataset will be added to the revised version of the paper.
>
>
>
> **Q4 (Fixed Weights):**
>
> We would like to highlight that this aspect has been acknowledged in our paper under the Limitation and Future Direction section. Despite the fact that the current fixed-weight approach simplifies the model and facilitates interpretation, we recognize that it may not fully capture the underlying data structure in certain domains and environments. Future studies may examine the possibility of:
>
> 1. **Variable Weights:** Assigning different weights to the linear and non-linear components to better reflect the specific structure of the data.
> 2. **Dynamic Adjustment:** Adjusting these weights dynamically at the arm level or for specific time steps to respond to changing reward patterns based on attention mechanisms.
>
> With these weights tunable, the model can be better adapted to environments where global trends and local patterns have varying relative importance. Although this level of tunability provides a greater degree of flexibility, it also poses challenges, such as increased computational complexity and additional hyperparameter optimization. We appreciate the reviewer's attention to this aspect, as it aligns well with the future directions proposed in our study. We view this as a promising direction for enhancing our model's adaptability and robustness.

---

> > ### Author Response · Authors · 2024-11-19
> > **Response to Reviewer 7MDe (Part 3/3)**
> >
> > **Q5 (High-dimensional Context):**
> >
> > Thanks to the reviewer for his insight into the impact of high-dimensional feature spaces on k-NN within the LNUCB-TA model. Indeed, high dimensionality can pose challenges, including reduced discriminative power in distance metrics and potential computational costs, especially in real-time applications. The current model focuses primarily on adaptability and performance in various contextual environments, however, we acknowledge that dimensionality can have a significant impact on the efficiency and accuracy of the k-NN component. In spite of the inherent limitations of k-NN in high-dimensional contexts, our approach incorporates both linear and nonlinear approaches to mitigate these limitations. In our hybrid model, the linear component provides a complementary global estimation, which reduces the sole dependence on the k-NN term for predicting rewards. By doing so, the model can still take advantage of linear trends in high-dimensional situations while adjusting locally to the greatest extent possible with the k-NN term.
> >
> > We envision multiple enhancements to LNUCB-TA in the future that would maintain or enhance its performance in high-dimensional scenarios. These enhancements include:
> > 1. **Dimensionality Reduction Techniques:** Principal Component Analysis can be used to reduce the dimensionality of the contextual feature space, which will enhance the k-NN component's ability to capture local patterns without compromising computational efficiency.
> > 2. **Feature Selection and Metric Learning:** It is possible for LNUCB-TA to prioritize the most relevant features or learn the optimal distance metric for k-NN by integrating feature selection or metric learning techniques. By employing these approaches, k-NN would be able to select neighborhoods in a meaningful manner, even in high-dimensional environments.
> > 3. **Adaptive Dimensionality Based on Context:** As another promising approach, we can adjust the dimensionality of the k-NN model dynamically based on the variance or contextual structure, ensuring that the model is not overly sensitive to dimensions that are not critical for predicting rewards.
> >
> > We evaluated the performance of LNUCB-TA on real-world datasets in the current study, demonstrating that these high-dimensional limitations had no negative impact on performance. Our analysis indicates, however, that these enhancements may further enhance the model's applicability in high-dimensional settings and consider this to be an exciting direction for future research.
> >
> > **Q6 & 7 (Euclidian Distance):**
> >
> > ### 1. Context Similarity
> >  When features are scaled appropriately, Euclidean distance can capture the similarity between contexts, making it an effective way of identifying relevant neighbors.
> >
> > ### 2. Computational Efficiency
> > Euclidean distance is a computationally simple method of determining the distance between context features. Calculating the distance involves only basic arithmetic operations (squared differences and summation) for each pair. In cases where context vectors are relatively low to moderately dimensional, Euclidean distance is efficient due to the low computational complexity.
> >
> > ### 3. Local Pattern Sensitivity and Distance-Based Weighting
> > It provides us with a means of defining a neighborhood of k-nearest contexts to ${X}_t$, and then calculating the adjustment term from the observed rewards from these neighbors. By averaging over nearby rewards, Euclidean distance ensures that contexts are included with minimal squared differences from ${X}_t$, capturing local reward patterns effectively.
> >
> > ### 4. Potential Extensions
> > If the feature space exhibits directional or correlation-based dependencies, we could consider alternative metrics such as the Mahalanobis distance. However, it is computationally more expensive (due to the inverse covariance calculation) and may not be as efficient for real-time applications.
> >
> > We selected the Euclidean distance metric here for its simplicity, computational efficiency, and effectiveness in capturing local similarities when features are homogeneously scaled. As our k-NN component is designed to capture localized reward patterns based on Euclidean proximity, it is suitable for dynamic environments in which real-time computation and adaptability are essential.
> >
> > ### Impact of High-Dimensional Settings
> > As our model captures global trends through the linear component, our k-NN adjustment is more of a local refinement than a primary model for estimating rewards. So, the k-NN component only refines the overall reward estimate even in high-dimensional settings, where Euclidean distance might be affected by irrelevant features. The impact is less pronounced. This refinement is intended to capture context-specific variations that the global linear model may miss. Due to the additional localized information this component provides, its impact is limited if high-dimensional irrelevant features slightly alter neighbor selection.

---

> > > ### Author Response · Authors · 2024-11-23
> > >
> > > Dear Reviewer 7MDe,
> > >
> > > We would like to express our sincere gratitude for your valuable insights and suggestions on our work. We have tried our best to address the concerns and queries you raised during the rebuttal process. However, we would greatly appreciate knowing whether our response has effectively resolved your doubts. Your feedback will be instrumental in improving the quality of our work. As the end of the discussion period is approaching, we eagerly await your reply before the end.
> > >
> > > Sincerely,
> > >
> > > Authors

---

> > > > ### Author Response · Authors · 2024-11-27
> > > >
> > > > Dear Reviewer 7MDe,
> > > >
> > > > We noticed that you have reduced your score, and we would appreciate it if you could share the technical reasons behind this change.
> > > >
> > > > Sincerely,
> > > > Authors

---

### Official Review · Reviewer_AgPg · 2024-11-04

**Soundness:** 3
**Presentation:** 3
**Contribution:** 2
**Rating:** 5
**Confidence:** 4

**Summary:**

The paper introduces LNUCB-TA, a contextual multi-armed bandit model that combines adaptive k-Nearest Neighbors with LinUCB along with adaptive exploration rate, provide sub linear regret guarantee and measure empirical performance against standard bandit algorithms.

**Strengths:**

- The paper's presentation and clarity is good.
- To the best of my knowledge, the proofs in the main paper seem fine.

**Weaknesses:**

1. One of my main concerns is the implicit reward structure in eq (1). The authors assume that the true expected reward is a sum of a linear and the average of the rewards of a set of nearest neighbors. It should be noted that the k-Nearest Neighbour UCB (Reeve et al., 2018) did not assume such a decomposition, but rather make a margin and lipschitz assumption (see Assumption 1 and 3 in (Reeve et al., 2018)). Why do the authors need to make such a restrictive assumption?

2. In the *Existing Gaps and Intuition* section the authors say "Linear models, constrained by static parameter updates, often fail in scenarios with inherently nonlinear relationships between contextual features and rewards." However the authors do not discuss a series of works in non-linear bandits (see [1],[2],[3],[4],[5],[6],[7], [8],[9],[10]) and the author's contribution with respect to these works.

3. The authors talk about temporal or time-dependent changes in the reward, but do not again discuss the development in non-stationary bandits or discuss why those solutions are not effective in the current scenario (see eg. https://proceedings.mlr.press/v75/luo18a/luo18a.pdf, https://proceedings.mlr.press/v99/chen19b.html, https://arxiv.org/abs/2310.07786 and the references therein).

4. Since the authors stress upon non-linearity of reward functions, there are missing benchmarks in the Experiments Section specifically with respect to NeuralUCB [5], Neural Thompson Sampling [6], Neural SquareCB/FastCB [7], Neural-Linear [3].

[1] Carlos Riquelme, George Tucker, and Jasper Snoek. Deep bayesian bandits showdown: An empirical comparison of bayesian deep networks for thompson sampling. In International Conference on Learning Representations, 2018.

[2] Xiuyuan Lu and Benjamin Van Roy. Ensemble sampling. In Proceedings of the 31st International
Conference on Neural Information Processing Systems, NIPS’17, pp. 3260–3268, Red Hook, NY,
USA, 2017. Curran Associates Inc. ISBN 9781510860964.

[3] Tom Zahavy and Shie Mannor. Neural linear bandits: Overcoming catastrophic forgetting through
likelihood matching, 2020.

[4] M. Valko, N. Korda, R. Munos, I. Flaounas, and N. Cristianini. Finite-time analysis of kernelised contextual
bandits. arXiv preprint arXiv:1309.6869, 2013.

[5] Dongruo Zhou, Lihong Li, and Quanquan Gu. Neural contextual bandits with ucb-based exploration.
In International Conference on Machine Learning, pp. 11492–11502. PMLR, 2020.

[6] Weitong Zhang, Dongruo Zhou, Lihong Li, and Quanquan Gu. Neural thompson sampling. In
International Conference on Learning Representation (ICLR), 2021.

[7] Rohan Deb, Yikun Ban, Shiliang Zuo, Jingrui He, and Arindam Banerjee. Contextual bandits with
online neural regression. In The Twelfth International Conference on Learning Representations,
2024a.

[8] D. Foster and A. Rakhlin. Beyond ucb: Optimal and efficient contextual bandits with regression oracles. In
International Conference on Machine Learning, pages 3199–3210. PMLR, 2020.

[9] D. Simchi-Levi and Y. Xu. Bypassing the monster: A faster and simpler optimal algorithm for contextual
bandits under realizability. ArXiv, abs/2003.12699, 2020.

[10] D. J. Foster and A. Krishnamurthy. Efficient first-order contextual bandits: Prediction, allocation, and
triangular discrimination. Advances in Neural Information Processing Systems, 34, 2021.

**Questions:**

- Could the authors clarify why they make the specific restrictive assumption on the reward function in eq (1), specifically in the context of existing algorithms for non-linear reward functions?

- Could the authors specify in more detail how their solution differs from the k-Nearest Neighbour UCB (Reeve et al., 2018) along with a detailed explanation of the following sentence from Existing gaps and intuition: "While nonlinear approaches like k-NN-based models (Reeve et al., 2018) offer flexibility, they often struggle with computational efficiency and adaptability in dynamic environments."

---

> ### Author Response · Authors · 2024-11-19
> **Response to Reviewer AgPg (Part 1/4)**
>
> We sincerely thank the reviewer for their valuable feedback and insightful comments. Please kindly find our response to address the weaknesses (W), and Questions (Q).
>
> **W1 & Q1 (Reward Structure):**
>
> We appreciate your insightful comment regarding incorporating k-NN into the expected reward. We recognize that this may seem unconventional as opposed to traditional contextual multi-armed bandit (MAB) models. For clarification, we have incorporated the **history of rewards** into the expected reward function. A nonlinear adjustment based on the average reward of a set of nearest neighbors is assumed to be added to the linear component of the true expected reward. As a result of this structure, **global trends**(through the linear term) and **local behavior** can be modeled. Our assumption enables us to capture reward dynamics adaptively in environments where reward structures vary across contexts, particularly nonstationary environments.
> It is suggested that this formulation be considered **a hybrid approach**: a linear setting enhanced by **a refinement using an adaptive k-NN**, based on reward history. Primary linear components capture the global relationship between context and reward, whereas k-NN adjustments provide a context-specific, localized correction. Based on historical reward data, this refinement enables the model to dynamically address nonlinear variations without fundamentally deviating from standard contextual bandit formulations. Taking this perspective into account, k-NN inclusion is not unusual, but rather **an extension** of the model that enhances its adaptability and precision in complex, evolving environments.
> We understand that Reeve et al. (2018) do not assume such a decomposition and instead rely on Lipschitz and Marginal assumptions (Assumptions 1 and 3 in their paper) to guide the reward estimation process. Their approach focuses on ensuring that the reward function behaves smoothly with respect to the context. In addition, there is a certain separation between rewards for different arms, which supports efficient exploration and exploitation. While their assumptions are effective for specific types of reward structures, our approach takes **a more flexible** route by incorporating historical rewards into the expected reward formulation, allowing for a richer model that can handle more complex reward structures.
>
> When classical bandit algorithms are implemented, the expected reward is generally determined by the context at the current time step. Once the context has been provided, it is assumed that the reward will be independent of past rewards. Our approach, on the other hand, **modifies the expected reward based on past rewards**. In our model, this assumption is primarily introduced to incorporate the history of rewards into the reward estimation process for each arm so that local reward patterns can be better captured. An adaptive k-NN term is used in the model to consider the history of actions in similar contexts in order to capture the impact of past rewards, which is not directly addressed by classical models. As a result, the model is capable of adapting its reward estimates in light of historical information, thereby providing a more comprehensive and flexible picture of rewards.
>
> The main intuition behind our approach is reward dynamics are nonstationary, in many real-world environments, which means that reward structures can evolve over time, and **past rewards can continue to impact future decisions**. Accordingly, the k-NN term allows the model to capture the impacts of past rewards by considering the history of actions in similar contexts, something that is not considered directly in classical models.
>
> The inclusion of k-NN directly in the expected reward in bandit models is less common, but we believe that this formulation significantly strengthens the model's ability to learn and **adapt to nonstationary environments** where previous actions and rewards have lasting effects.
>
> **Q2 (Difference with k-NN UCB):**
>
> We would like to kindly highlight that throughout the paper, particularly in **Sections 3.1, 3.2, and 3.3**, we have made an effort to provide a clear explanation of how our model relates to and differs from the k-Nearest Neighbor UCB (Reeve et al., 2018). We propose a unique synthesis of linear and nonlinear components, which captures both global trends by a linear model as well as localized adjustments by a k-NN-based nonlinear model (lines 17-20). In combination, our model can handle both broad and local variations in reward structures more effectively than traditional k-NN based approaches, which might only provide localized adaptations (lines 80-84).

---

> ### Author Response · Authors · 2024-11-19
> **Response to Reviewer AgPg (Part 2/4)**
>
> **Continue of Q2 (Difference with k-NN UCB):**
>
> In terms of computational efficiency, **Table 1** (as shown in the column on k selection method, the proposed k-NN is adaptive and the k is selected **non-parametrically based on the variance of rewards**, whereas the k-NN UCB is based on function optimization), **Table 2** (a difference in execution **run times** is shown in the column), and Figure 8 (the comparison of our model against the simple combination model, k-NN UCB + LinUCB, regarding **runtime and scalability**). Also, as stated in lines 320 to 323, unlike existing nonlinear approaches that involve a search over k, preceding time steps k ∈ [1, t−1] (Park et al., 2014; Reeve et al., 2018), our proposed model utilizes a data driven approach for selecting k, achieving **less time complexity** and significantly decreasing time complexity compared to the **function optimization** techniques used in k-NN UCB and k-NN KL-UCB).
>
> For **adaptability** in dynamic environments, we have implemented k-NN adjustments based on reward history, which enables our model to respond flexibly to changing conditions over time (lines 312-320). In this section, specific scenarios with high and low rewards have been discussed. Additionally, our attention-based exploration parameter, described in **Section 3.3**, dynamically balances exploration and exploitation in real time (lines 25-28, table 1, lines 85-97, lines 191-201, etc.). As a result of this continuous adjustment, based on recent arm performance, our model becomes **more adaptable** to evolving reward structures, which addresses the limitations of static and fixed-rate approaches (lines 326-331). Furthermore, you can see in **Figures 3 and 4** the comparison between k-NN UCB and our model at different exploration rates, and thanks to the attention-based exploration rate proposed in our model, we are able to provide **more consistent results**, showing that our model is more responsive to changes in reward, also providing a better estimation with a higher mean and cumulative reward.

---

> ### Author Response · Authors · 2024-11-19
> **Response to Reviewer AgPg (Part 3/4)**
>
> **W2-4 (Literature)**
>
> Table 1: Comparing our model with the 10 referenced models, highlighting the unique aspects and contributions of our approach.
>
> | **Aspect**                           | **Our Model (LNUCB-TA)**                          | **Deep Bayesian Bandits Showdown**               | **Ensemble Sampling**                         | **Neural Linear Bandits**                  | **Kernelized Contextual Bandits**      | **NeuralUCB**                             | **Neural Thompson Sampling**              | **Contextual Bandits with Online Neural Regression** | **Optimal Contextual Bandits with Regression Oracles** | **FALCON**                       | **Efficient First-Order Contextual Bandits (FastCB)** |
> |--------------------------------------|--------------------------------------------------|-------------------------------------------------|-----------------------------------------------|--------------------------------------------|----------------------------------------|--------------------------------------------|-----------------------------------------------|-------------------------------------------------------|-------------------------------------------------------|-----------------------------------|-------------------------------------------------------|
> | **Primary Approach**                 | Linear + non-linear (k-NN) with attention-based α | Bayesian posterior approximations with neural networks | Ensemble models for approximation in complex models | Neural network with linear exploration     | Kernel-based for non-linear feature mapping | Neural network-based UCB                | Neural Thompson Sampling with neural network | Online neural regression                             | Regression oracle-based contextual bandit             | Reduction of contextual bandits to offline regression using least squares regression oracle. | First-order contextual bandit with regression oracle |
> | **Exploration-Exploitation Adjustment** | Real-time adjustment of α with attention on global and local rewards | Static posterior samples                        | Static ensemble diversity-based              | Fixed exploration parameter, memory-based updates | Fixed exploration parameter            | Fixed UCB confidence intervals           | Posterior sampling from reward distribution | Fixed parameter for exploration                     | Regression-based oracle                             | Adaptive via epoch-varying learning rate.             | Fixed oracle-based weighting                          |
> | **Non-Linearity Handling**           | k-NN term for local non-linear adjustments       | Non-linearity via neural networks              | Non-linearity through ensemble networks      | Non-linearity through neural networks      | Non-linearity via kernel functions       | Non-linearity through neural feature mappings | Non-linearity through neural networks       | Non-linearity with neural regression                 | Adaptable with function class                       | Flexible handling of function classes including parametric, non-parametric, and neural networks. | Adaptable through weighted regression oracle        |
> | **Dynamic Environment Adaptability** | Attention-driven, adapts exploration based on recent arm performance | No specific mechanism for dynamic adaptation   | Limited to ensemble refresh                  | Limited; mitigates forgetting with memory buffer | Limited; requires fixed kernel          | Limited by fixed confidence bounds        | Limited by fixed exploration parameters     | Limited by fixed exploration parameters             | Limited to regression oracle adjustments            | Moderate; suitable for settings where function class adapts slowly. | Fixed first-order weighting                          |
> | **Need for Hyperparameter Tuning**   | Low, α adapts dynamically, reducing need for preset exploration tuning | Moderate, requires neural architecture tuning | Moderate, ensemble size tuning required     | High, neural architecture and buffer size tuning required | High, kernel selection and tuning required | High, neural architecture and confidence tuning required | High, neural architecture and sampling tuning | High, neural architecture tuning                     | Low, regression oracle reduces need for additional tuning | Low                                                  | Low, regression oracle-based with minimal tuning    |
> | **Real-Time Exploration Control**    | Yes        | No                                             | No                                           | No                                        | No                                      | No                                         | No                                              | No                                                   | No                                                   | No                                                 | No                                                   |

---

> ### Author Response · Authors · 2024-11-19
> **Response to Reviewer AgPg (Part 4/4)**
>
> **Continue of W2-4**
>
> Table 2: Comparison of Contextual Bandit Algorithms Across Modeling, Attention, and Non-Linearity Features
>
> | **Algorithm**                 | **Linear Modeling** | **Local History Modeling** | **Attention Mechanism** | **Non-Linearity Handling**                |
> |-------------------------------|---------------------|----------------------------|--------------------------|-------------------------------------------|
> | **UCB**                       | No                  | No                         | No                       | None                                      |
> | **KL-UCB**                    | No                  | No                         | No                       | None                                      |
> | **k-NN UCB**                  | No                  | Yes                        | No                       | k-NN adjustment                          |
> | **k-NN KL-UCB**               | No                  | Yes                        | No                       | k-NN adjustment                          |
> | **LinThompson**               | Yes                 | No                         | No                       | None                                      |
> | **LinThompsonUCB**            | Yes                 | No                         | No                       | None                                      |
> | **LinUCB**                    | Yes                 | No                         | No                       | None                                      |
> | **Deep Bayesian Bandits**     | No                  | No                         | No                       | Deep neural networks                     |
> | **Ensemble Sampling**         | No                  | Yes                        | No                       | Ensemble diversity                        |
> | **Neural Linear Bandits**     | Yes                 | Yes                        | No                       | Neural network-based feature mapping      |
> | **Kernelized Contextual**     | No                  | Yes                        | No                       | Kernel-based non-linearity               |
> | **NeuralUCB**                 | No                  | Yes                        | No                       | Neural feature mappings                  |
> | **Neural Thompson Sampling**  | No                  | Yes                        | No                       | Non-linearity via neural networks        |
> | **Online Neural Regression**  | No                  | Yes                        | No                       | Neural network-based regression          |
> | **Optimal Contextual Bandit** | No                  | Yes                        | No                       | Regression oracle-based                  |
> | **FALCON**                    | No                  | Yes                        | No                       | Regression oracle-based non-linearity    |
> | **FastCB**                    | No                  | Yes                        | No                       | Regression oracle-based                  |
> | **LNUCB-TA**                  | Yes                 | Yes                        | Yes                      | k-NN for local non-linearity             |
>
>
> References presented cover a broad spectrum of approaches, ranging from Bayesian sampling and neural networks to kernel-based methods, all of which address contextual bandit challenges with distinct strategies for non-linearity and exploration versus exploitation. The Deep Bayesian Bandits Showdown and Neural Thompson Sampling algorithms utilize Bayesian posterior sampling to approximate exploration strategies but lack mechanisms for adapting these strategies in real-time. Ensemble Sampling and Neural Linear Bandits are models that utilize ensemble networks and neural feature mappings in order to address non-linear relationships, but these methods do not incorporate dynamic local adjustments in non-stationary environments or provide consistent adaptation. In Kernelized Contextual Bandits and NeuralUCB, nonlinearity is captured using kernel or neural embeddings, which effectively model complex reward functions. However, they have fixed parameters that limit their adaptability to evolving reward structures and do not have a mechanism for dynamically adjusting exploration rates.
>
> As an alternative, we propose a LNUCB-TA model that combines a global linear model with a k-NN-based non-linear adjustment, providing a more versatile approach to handling dynamic and complex environments. Unlike any of the ten previously mentioned models, LNUCB-TA introduces a novel attention-based mechanism to continuously balance exploration and exploitation based on recent arm performance. Our model's dual structure for global and local patterns, coupled with our adaptive control, enables consistent performance across non-stationary scenarios without extensive tuning.

---

> > ### Author Response · Authors · 2024-11-23
> >
> > Dear Reviewer AgPg,
> >
> > We would like to express our sincere gratitude for your valuable insights and suggestions on our work. We have tried our best to address the concerns and queries you raised during the rebuttal process. However, we would greatly appreciate knowing whether our response has effectively resolved your doubts. Your feedback will be instrumental in improving the quality of our work. As the end of the discussion period is approaching, we eagerly await your reply before the end.
> >
> > Sincerely,
> >
> > Authors

---

> > > ### Comment · Reviewer_AgPg · 2024-11-24
> > >
> > > Thank you for your response.
> > >
> > > - I maintain my reservation about the assumption on the reward structure. If the authors claim such a reward structure models the real world data well, they need to present ample amount of empirical evaluation showing that. Since they specifically talk about non-linear and non-stationary reward functions, I would expect empirical evaluation on algorithms that cater to these scenarios. Note that I provided several references to non-stationary bandits that the authors ignored.
> > >
> > > - Further, experiments section need to benchmark against neural bandit algorithms, non-stationary algorithms and simple combinations of these to provide convincing evidence that the propsed algorithm indeed solves a problem that current algorithms or simple combinations of them do not.
> > >
> > > - Putting a giant table with superficial phrases is a bad way of comparing with existing literature (Table-1). Further I do not understand what the purpose of the Table-2 is. I am guessing it is to tell us that "LNUCB-TA" uses "Attention Mechanism". Without empirical evidence that such a modelling is required in real world data, I do not see why it is of any improtance.
> > >
> > > P.S. Authors can update the paper draft. It is more advisable to make modifications to the draft, and summarize responses with pointers to specific parts in the draft rather than dumping a huge pile of text here.

---

> ### Author Response · Authors · 2024-11-24
>
> Thank you for your continued feedback and for raising these points.
>
> **Benchmark and reward structure**
>
> We would like to highlight that we have already presented comparison **against 14 different models on three datasets**, demonstrating the robustness of our model across diverse scenarios. Moreover, **an additional dataset** was incorporated based on Reviewer 7MDe's comments, further extending our evaluations.
>
> However, as requested, we have now provided results for **KernelUCB and NeuralUCB**. **It is worth noting that NeuralUCB, as demonstrated in its original paper, outperformed five other neural network-based methods. Notably, our model outperforms NeuralUCB in the two datasets analyzed, further showcasing its effectiveness and adaptability.**
>
> ### Performance Comparison Table On News Recommendation Dataset
> | **Model**      | **Exploration Rate (α)** | **Cumulative Reward** | **Mean Reward** |
> |-----------------|--------------------------|------------------------|-----------------|
> | **LNUCB-TA**   | 0.01                    | **752**                    | **0.94**            |
> | **LNUCB-TA**   | 0.1                     | **741**                    | **0.93**            |
> | **LNUCB-TA**   | 1                       | **752**                    | **0.94**            |
> | Neural UCB | 0.01                    | 726                    | 0.90            |
> | Neural UCB | 0.1                     | 717                    | 0.89            |
> | Neural UCB | 1                       | 722                    | 0.90            |
> | Kernel UCB | 0.01                    | 479                    | 0.60            |
> | Kernel UCB | 0.1                     | 414                    | 0.52            |
> | Kernel UCB | 1                       | 446                    | 0.56            |
>
> Based on the table above, our proposed model outperforms the two other models. As Neural UCB performs well in this dataset, we extended the analysis to compare it with our model on the AstroPh co-authorship network in terms of cumulative reward (y-axis) observed at 5% (Figure 7). Neural UCB achieves a cumulative reward of 8636 observed nodes, which is **approximately 12% lower** than the **9808** achieved by our model (LNUCB-TA). Kernel UCB, on the same dataset, achieves 9332 observed nodes, which is **4.85% lower than our model**. Moreover, even our proposed component combined with Epsilon Greedy (green line in Figure 7) surpasses Neural UCB with 8917 observed nodes, **approximately 3.5% higher** than Neural UCB. We will provide the updated Figure 7 in our revised manuscript by adding these two models to the Figure.
>
> Interestingly, in the news recommendation dataset, Neural UCB outperformed Kernel UCB, whereas in the AstroPh co-authorship network, Kernel UCB showed better performance than Neural UCB. Despite these variations, **our model outperformed both methods across both datasets**.
>
> Finally, we would like to note that the bandit literature encompasses a vast number of models. While reviewers may suggest adding comparisons with additional models, it is neither practical nor standard for every paper to benchmark against all existing methods. Instead, comparisons are typically focused on models most relevant to the context of the presented work. In our case, while comparisons with neural-based models are informative, they are not the primary focus, as our approach fundamentally differs from neural-based methodologies.
>
> ---
>
> ### **Table 1 and 2**
>
>
> Table 1 serves to systematically position our model relative to existing approaches by highlighting key methodological differences (the respected reviewer has specifically asked to **discuss this series of work** in your W2-4).
> Table 2, in particular, underscores our model’s ability to incorporate both global trends and local refinements, while introducing attention for dynamic exploration. This unique synthesis is validated empirically through extensive results across multiple datasets (see Table 2, Figure 2, Figure 5, etc.), demonstrating the tangible benefits of combining these components.
>
> We would like to respectfully note that the length of our responses was driven by your feedback. Specifically, the respected reviewer had requested **discussions on ten additional papers**. While the responses were detailed, they were structured and targeted to address the points raised. Striking the right balance between brevity and clarity is challenging in such cases, but our intention was always to provide clarity rather than overwhelm. It is, however, surprising that the reviewer focused solely on comparisons, seemingly overlooking our contributions, theoretical proofs, and other significant aspects of the work.
>
>
> We appreciate the suggestion to revise the draft directly. However, as we can submit only one updated PDF, we plan to provide a revised draft after receiving feedback from all reviewers. This ensures a unified and comprehensive update.

---

> > ### Comment · Reviewer_AgPg · 2024-11-24
> >
> > - I appreciate the authors running one of the necessary baselines. I would request them to upload their code for this specific comparison, along with reproducibility information as the supplementary file, to help me further evaluate the experiments.
> > - Regarding "**It is worth noting that NeuralUCB, as demonstrated in its original paper, outperformed five other neural network-based methods.**" Neural Thompson Sampling [6], Neural SquareCB/FastCB [7] have been shown to outperform Neural UCB on a number of tasks, and therefore are still missing from the baselines.
> > - Regarding "**Finally, we would like to note that the bandit literature encompasses a vast number of models. While reviewers may suggest adding comparisons with additional models, it is neither practical nor standard for every paper to benchmark against all existing methods. Instead, comparisons are typically focused on models most relevant to the context of the presented work. In our case, while comparisons with neural-based models are informative, they are not the primary focus, as our approach fundamentally differs from neural-based methodologies.**" : In this reviewers opinion, they are absolutely necessary baselines, since the authors have talked about nonlinear variation in the reward functions. Further since the authors say "adapt to nonstationary environments", non-stationary bandit algorithms are also relevant (which the authors keep ignoring).
> > - Again my primary objection is the assumption on the reward structure. One needs to motivate with several datasets, and existing baselines why the community should care about such an assumption.
> > - Regarding the tables - In this reviewers subjective opinion, Table-1 is exceedingly terse to infer much from, and Table-2 is unnecessary as one cannot infer why and if anyone needs "Linear Modeling" or "Attention Mechanism".

---

> ### Author Response · Authors · 2024-11-24
>
> Thank you for your quick response.
>
> We will update the supplementary materials to include the code for the new added models, along with all necessary reproducibility information, as part of the revisions based on the ongoing feedback.
>
> We now have provided results for 16 models across four datasets, which we believe is significantly comprehensive compared to standard practice in the bandit literature. While Neural Thompson Sampling and Neural SquareCB/FastCB are important baselines in the context of neural-based methodologies, our model fundamentally diverges from such neural approaches in its design and focus. Moreover, when we discuss "adapting to non-stationary environments," we refer specifically to handling non-stationary real-world datasets, not adopting a framework designed solely for theoretical non-stationary bandits. Our chosen datasets include datasets with inherently non-stationary characteristics, making them highly relevant for evaluating the adaptability of our model. Additionally, the reward structure in our model has been theoretically validated with a sublinear regret bound. Theoretical guarantees have been provided in the manuscript to substantiate the soundness of this structure. It is surprising that this aspect has been overlooked by the respected reviewer in terms of its **theoretical proof**, which **directly demonstrates why the community should care** about such an assumption, as it is comprehensively justified and foundational to the contributions presented in the paper.
>
> In summary, **the main baselines relevant to our setup are LinUCB, k-NN UCB, k-NN KL UCB, the vanilla Lin+k-NN UCB combination, LinThompson, and KernelUCB, all of which have been extensively compared in our paper**. Additionally, our reward structure has been supported with theoretical proof for the sublinear regret bound. This further establishes the validity of our approach. Even in the NeuralUCB and Neural Thompson Sampling papers, comparisons are limited to 6–7 models most related to their setup, and not every possible baseline. We believe our approach aligns with standard practices in this regard while providing both empirical and theoretical justification for our contributions.
>
>
> Respectfully, we note the inconsistency in the reviewer’s feedback. While earlier comments criticized the detailed responses as a "**huge pile of text**," the current critique suggests that Table-1 is "**exceedingly terse**." This contradiction highlights the challenge of striking the right balance between brevity and comprehensiveness in presenting comparisons. The descriptions provided in Part 4 of our answer further clarify their purpose. Regarding Table-2, **it is not meant** to argue "**why**" linear modeling or attention mechanisms are needed but rather to summarize **what** our contributions are and how they differentiate our approach (please refer to Table 1 of the paper).
>
> For the reasons (infer why and if anyone needs), please refer to the section on "Existing Gaps and Intuition" (**lines 75–97**) and for the motivating examples in real-world, please refer to the dedicated subsection (**lines 130-147**). For additional intuition on each contribution, dedicated sections are provided (**lines 244–251, 280–285, 326–331**). We respectfully request that the reviewer refer to these sections for a fuller understanding of our motivations and contributions.

---

> > ### Comment · Reviewer_AgPg · 2024-11-25
> >
> > - I think we have differing opinions on what the relevant baselines are, and therefore I leave it at the good judgement of the Area Chair. I summarize my primary concern for them below.
> >
> >
> >   - **Since the true reward function makes an assumption on its structure, and the authors claim it adresses nonlinear variations in reward functions and adapts to nonstationary environments, existing neural bandit algorithms, and the algorithms for non-stationary environments (as referred in my previous responses) that have been proposed for such scenarios, are necessary baselines. Even if the proposed algorithm is not a neural network based, one needs convincing evidence that these existing algorithms are sub-par across environments.**
> >
> > - Although I appreciate the efforts put in to provide theoretical bounds, such an analysis for a specifc structural assumption on the reward function would not be of much significance to the community if existing algorithms already provide empirical success and provable regret performance without such an assumption.
> >
> > - I suspect that with proper hyper parameter tuning (including the number of layers, width of the network and step-size) Neural UCB and other neural bandit algorithms that the authors did not compare against will provide better empirical bounds. I would encourage the authors to run these and provide code for further validation.
> >
> > - Regarding the Tables, I apologize if the authors are offended by the comment, but I still think the table as a whole is a pile of text and each of the individual entries in Table 1 are terse and abrupt to infer much from, while Table 2 seems unnecessary without empirical justifications as to why the reward modeling is required (this relates to my primary concern stated above).

---

> ### Author Response · Authors · 2024-11-25
>
> Dear Reviewer AgPg,
>
> Thank you for your time and comments; we have conducted the experiment you requested, as detailed below.
>
> ---
>
> Table 4: Comparison of our model vs neural bandit models based on total regret (Mean ± Standard Deviation and relative Std/Mean percentage over 20 runs)
> | Dataset   | Linear UCB       | Linear TS        | Kernel UCB       | Kernel TS        | BooststrapNN     | eps-greedy      | NeuralUCB       | NeuralTS        | LNUCB-TA (our proposed model)    |
> |-----------|------------------|------------------|------------------|------------------|------------------|-----------------|-----------------|-----------------|-------------------|
> | Adult     | 2097.5 ± 50.3 (2.40%)  | 2154.7 ± 40.5 (1.88%)  | 2080.1 ± 44.8 (2.15%)  | 2111.5 ± 87.4 (4.14%)  | 2097.3 ± 39.3 (1.87%)  | 2328.5 ± 50.4 (2.16%) | 2061.8 ± 42.8 (2.08%) | 2092.5 ± 48.0 (2.29%) | **1673.1** ± **12.07** **(0.72%)**    |
> | Magic     | 2604.4 ± 34.6 (1.33%)  | 2700.5 ± 46.7 (1.73%)  | 2406.5 ± 79.4 (3.30%)  | 2442.6 ± 64.5 (2.64%)  | 2269.4 ± 27.9 **(1.23%)** | 2381.8 ± 37.3 (1.57%) | 2033.0 ± 48.6 (2.39%) | 2037.4 ± 61.3 (3.01%) | **1931.6** ± **31.22** (1.62%)   |
> | MNIST     | 2544.0 ± 235.4 (9.25%) | 2781.4 ± 338.3 (12.16%)| 3595.8 ± 580.1 (16.13%)| 3406.0 ± 411.7 (12.09%)| 1765.6 ± 321.1 (18.19%)| 1893.2 ± 93.7 (4.95%) | 2071.6 ± 922.2 (44.49%)| 1583.4 ± 198.5 (12.53%)| **1561.6** ± **42.09** **(2.69%)**   |
> | Mushroom  | 562.7 ± 23.1 **(4.11%)**   | 643.3 ± 30.4 (4.72%)   | 199.0 ± 41.0 (20.60%)  | 291.2 ± 40.0 (13.74%)  | 132.3 ± 8.6 (6.50%)    | 323.2 ± 32.5 (10.06%) | 160.4 ± 95.3 (59.41%) | 115.0 ± 35.8 (31.13%) | **19.85** ± **1.98** (9.97%)    |
> | Shuttle   | 966.6 ± **39.0** **(4.04%)**   | 1020.9 ± 42.8 (4.19%)  | **166.5** ± 39.4 (23.66%) | 283.3 ± 180.5 (63.72%)| 211.7 ± 20.9 (9.87%)   | 682.0 ± 79.8 (11.70%) | 338.6 ± 386.4 (114.13%)| 232.0 ± 149.5 (64.45%)| 283.1 ± 55.44 (19.58%)     |
>
> This table compares the performance of total regret and standard deviation across datasets, including results from the NeuralTS paper and our proposed LNUCB-TA model. **The regret values for the baseline models are taken directly from the NeuralTS paper [1]** (Figure 1, Table 1, and Figure 3), where a detailed grid search for hyperparameter tuning was conducted (as described in Section A.1 [1]). Consequently, the proper hyperparameter tuning mentioned by the respected reviewer is inherent in the baselines, ensuring a fair comparison.
>
> The respected reviewer has explicitly noted that *"proper hyperparameter tuning (including the number of layers, width of the network, and step size)"* is critical for the performance of neural-based bandit models. This observation aligns with the findings in the NeuralTS paper, which highlights significant variability in performance and high sensitivity to hyperparameter tuning, as evidenced in their Figure 2, Figure 4, and Table 2. This sensitivity underscores the inherent challenges of using neural models in contextual bandit frameworks. Furthermore, the computational efficiency is another challenge in neural-based models and these models mainly use diagonalized matrix [1-3]. For instance, Figure 5 in the NeuralTS paper demonstrates the high runtime required for exploration in neural bandits.
>
> Practical concerns about the computational cost in exploration are significant for neural bandits like NeuralUCB and NeuralTS. These models require the construction of high-probability confidence sets based on the dimensionality of network parameters and context vector representations, often involving matrices with hundreds of thousands of parameters. As a result, approximations (e.g., only using diagonal covariance matrices) are employed to mitigate this computational burden [2, 3], but these approximations lack theoretical guarantees, creating gaps between theoretical and empirical performance [4].
>
> The Neural bandit with perturbed reward (NPR) model [4] tries to addresses computational efficiency in neural contextual bandits but highlights that online model updates in neural bandit models, relying on stochastic gradient descent **over entire training sets at each round**, remain a significant computational bottleneck. In contrast, our hybrid model design significantly mitigates this bottleneck by employing computationally efficient methods that avoid the iterative gradient descent updates required by neural network-based approaches.
>
> Furthermore, the NPR model’s regret performance is equal to or worse than NeuralUCB on datasets such as Adult, Mushroom, and Magic (Figures 2 and 6 in [4]). In comparison, **our proposed LNUCB-TA outperforms NeuralUCB across all evaluated datasets as shown in the table above** and demonstrates the least standard deviation in 4 out of 5 datasets, showcasing its robustness. Additionally, the results presented in Table 2, Figure 2 (part d), and Figure 6 (error bar plot) of our paper emphasize the consistency and reliability of our approach across varying parameter settings.

---

> > ### Author Response · Authors · 2024-11-25
> >
> > Thank you for sharing your perspective on the tables in our response. We truly appreciate your engagement and your apology—please rest assured, no offense was taken.
> >
> > The purpose of Table 1 in our initial response was to summarize the differences between our model and the ten additional papers you kindly asked us to discuss, focusing on various methodological aspects. Given the 5000-character limitation for responses, the table was designed to be concise and structured, providing an accessible overview while following the constraints.
> >
> > Regarding Table 2  in our initial response, we appreciate your feedback. This table is included in the introduction section of the paper (Table 1 in the paper) to outline our contributions , rather than in the results section, where empirical evidence is typically discussed. Its purpose is to provide a clear summary of our key innovations (summarize what our contributions are and how they differentiate our approach), aligning with common practices in academic papers. Similar examples include Table 1b in [5] and Table 1 in [6] from recent ICLR publications. We hope this explanation clarifies the intent and utility of this table in the context of our work.
> >
> > We sincerely appreciate your acknowledgment of the theoretical bounds we provided. However, we are still a bit surprised that this level of analysis is considered insufficient. To provide additional context, the regret bounds of several key models are as follows:
> >
> > - **NPR**: $\tilde{O}(\tilde{d}\sqrt{T})$
> > - **NeuralUCB**: $\tilde{O}(\tilde{d}\sqrt{T})$
> > - **NeuralTS** and **Our Model**: $\tilde{O}(\sqrt{dT})$
> >
> > Our regret bound is comparable to these models, particularly given that **we achieve this without relying on neural networks**, which are known for their computational complexity, data demands, and sensitivity to hyperparameter tuning. Achieving such theoretical guarantee underscores the practicality and robustness of our approach for real-world applications.
> >
> > Additionally, we have now provided comparisons with neural-based methods such as NeuralUCB and NeuralTS, addressing your primary concern. We hope these updates further demonstrate the relevance and effectiveness of our model in this context.
> >
> >
> > We hope this now satisfies your concerns.
> >
> > Sincerely
> >
> > The Authors
> >
> >
> > ---
> >
> >
> > **References**
> >
> > [1] Weitong Zhang, Dongruo Zhou, Lihong Li, and Quanquan Gu. Neural thompson sampling. In International Conference on Learning Representation (ICLR), 2021.
> >
> > [2] Kassraie, Parnian, and Andreas Krause. "Neural contextual bandits without regret." International Conference on Artificial Intelligence and Statistics. PMLR, 2022.
> >
> > [3] Dongruo Zhou, Lihong Li, and Quanquan Gu. Neural contextual bandits with ucb-based exploration. In International Conference on Machine Learning, pp. 11492–11502. PMLR, 2020.
> >
> > [4] Jia, Yiling, et al. "Learning neural contextual bandits through perturbed rewards." arXiv preprint arXiv:2201.09910 (2022).
> >
> > [5]Goktas, Denizalp, et al. "Efficient Inverse Multiagent Learning." The Twelfth International Conference on Learning Representations (2024).
> >
> > [6] Seunghan Lee et al., "Soft Contrastive Learning for Time Series." The Twelfth International Conference on Learning Representations (2024).

---

### Note · Authors · 2024-12-03

**Comment:**

We have decided to withdraw our submission, but we feel it is important to address some critical points for transparency and to defend our work, given that all comments will be made public.

---

## Reviewer 1LHD

We respectfully note that Reviewer 1LHD's feedback demonstrates significant misunderstandings of several critical aspects of our submission:

1. **Misrepresentation of Our Work as a Mixture Model**
   Surprisingly, the reviewer misinterpreted our model as a mixture model, which fundamentally differs from the contextual bandit framework we presented. Despite detailed clarifications highlighting this distinction, the misunderstanding persisted.

2. **Misunderstanding of β as a Standard Notation**
   The reviewer claimed β was undefined, even though it was explicitly defined in Equation (8) in the main text and elaborated in Definition 1, Corollary 3, and Assumption 1 in the appendix. β represents the radius of a confidence ball, a standard concept in the contextual bandit literature, as used in LinUCB, KernelUCB, etc. This oversight demonstrates the reviewer’s unfamiliarity with foundational concepts in the field.

3. **Misinterpretation of the Attention Mechanism as a Weakness**
   The reviewer critiqued the fast decay of α in specific scenarios as a weakness, despite it being a key strength of our model. When local and global rewards align, this decay optimally shifts focus to exploitation, a desirable property for contextual bandits. Again, this shows the reviewer’s unfamiliarity with fundamental concepts in multi-armed bandits such as the exploitation-exploration tradeoff.

4. **Overwhelmed by Long Responses**
   The reviewer expressed feeling overwhelmed by the length of our responses. We feel such comments are unwarranted given that our clarifications were in response to the reviewer’s questions, some of which were on items already clearly stated in our submitted manuscript.

---

## Reviewer j6kr

Reviewer j6kr's repeated questions on concepts explicitly addressed in the manuscript, such as the dynamic adjustment of k in k-NN, were concerning. The reviewer expressed confusion about the setup, which appears to stem from either unfamiliarity with the contextual bandit domain or a lack of engagement with the paper, as these concepts were clearly detailed in the manuscript. This misalignment resulted in unnecessary repetition of explanations, unnecessarily lengthening our responses.

---

## Reviewer AgPg
Reviewer AgPg suggested additional comparisons and an expanded literature review. Initially, we provided results for 3 datasets and 14 models. Following the reviewer’s feedback, we extended the analysis to **9 datasets and 19 models**. To our knowledge, we have provided one of the most comprehensive comparisons in the contextual bandit literature.

Additionally, Reviewer AgPg insisted on his/her personal and subjective opinion that Table 1 in the paper, which summarizes our contributions, is unnecessary and should be deleted because it does not include empirical results. This criticism is surprising, as the table is located in **the introduction section** and is intended to highlight contributions, not present empirical findings. Such tables are a standard practice in academic papers, as seen in **Table 1b in [1] and Table 1 in [2]**, both published at ICLR last year. Removing a contribution summary from the introduction simply because it lacks empirical results contradicts established norms in the field.

Additionally, Reviewer AgPg dismissed the importance of our theoretical guarantees and regret bound proofs, which is surprising given their foundational role in contextual bandit research. This remark contradicts the value placed on theoretical analysis in the community.


---

## Reviewer 7MDe

We are disappointed that Reviewer 7MDe reduced their score from 8 to 6 without providing any explanation or engaging in further discussion. We are deeply disappointed about this lack of transparency and unprofessionalism. We feel that this does not align with the principles of constructive peer review.


---


While we deeply respect ICLR’s commitment to scientific rigor, this experience has highlighted issues of fairness and consistency in the review process. Despite the detailed responses and extensive revisions that we have provided, several reviewers overlooked critical aspects of our work, failed to engage constructively, or exhibited unfamiliarity with the field.
We hope these reflections will contribute to improving the review process in the future and ensure a fairer evaluation for all submissions.



[1] Goktas, Denizalp, et al. "Efficient Inverse Multiagent Learning." The Twelfth International Conference on Learning Representations (2024).

[2] Seunghan Lee et al., "Soft Contrastive Learning for Time Series." The Twelfth International Conference on Learning Representations (2024).

**Withdrawal Confirmation:**

I have read and agree with the venue's withdrawal policy on behalf of myself and my co-authors.